# Trophoblast stem cell-based organoid models of the human placental barrier

Takeshi Hori [1], Hiroaki Okae[2,3], Shun Shibata [2], Norio Kobayashi[2,4], Eri H. Kobayashi[2], Akira Oike[2,3], Asato Sekiya[3], Takahiro Arima [2] & Hirokazu Kaji [1] ✉

Human placental villi have essential roles in producing hormones, mediating nutrient and waste exchange, and protecting the fetus from exposure to xenobiotics. Human trophoblast organoids that recapitulate the structure of villi could provide an important in vitro tool to understand placental development and the transplacental passage of xenobiotics. However, such organoids do not currently exist. Here we describe the generation of trophoblast organoids using human trophoblast stem (TS) cells. Following treatment with three kinds of culture medium, TS cells form spherical organoids with a single outer layer of syncytiotrophoblast (ST) cells that display a barrier function. Furthermore, we develop a column-type ST barrier model based on the culture condition of the trophoblast organoids. The bottom membrane of the column is almost entirely covered with syndecan 1-positive ST cells. The barrier integrity and maturation levels of the model are confirmed by measuring transepithelial/transendothelial electrical resistance (TEER) and the amount of human chorionic gonadotropin. Further analysis reveals that the model can be used to derive the apparent permeability coefficients of model compounds. In addition to providing a suite of tools for the study of placental development, our trophoblast models allow the evaluation of compound transfer and toxicity, which will facilitate drug development.

In its role as essential organ for fetus development, the placenta produces hormones and mediates nutrients and waste exchange[1]. It also acts as a barrier against xenobiotics including multiple drugs. However, some drugs can still cross the placental barrier and reach the fetus, eliciting adverse effects. Thus, testing the placental permeability of chemicals is critical in drug development[2]. Ex vivo perfusion systems with human placental explants are the gold standard for measuring the transplacental passage of drugs[3–6]. An advantage of perfusion systems is that they can be used to obtain accurate information on the concentration rate of chemicals in both the maternal and fetal compartments. However, there are also disadvantages associated with using these systems, which include the complexity of system setup, access to placental resources, inter-individual heterogeneity, and the ability to expand usage to chemical screening and testing. Moreover, the perfusion systems are constituted with term placentas and therefore cannot produce information on the kinetics of chemicals in the first trimester when the embryo or fetus is generally most sensitive to chemicals[2]. Animal experiments are suboptimal for the assessment of chemical translocation because of interspecies variability in placental barrier structure. The development of a barrier model using human

[1]Department of Diagnostic and Therapeutic Systems Engineering, Institute of Biomaterials and Bioengineering (IBB), Tokyo Medical and Dental University (TMDU), 2-3-10 Kanda-Surugadai, Chiyoda-ku, Tokyo 101-0062, Japan. [2]Department of Informative Genetics, Environment and Genome Research Center, Tohoku University Graduate School of Medicine, 2-1 Seiryo-cho, Aoba-ku, Sendai 980-8575, Japan. [3]Department of Trophoblast Research, Institute of Molecular Embryology and Genetics, Kumamoto University, Kumamoto 862-0973, Japan. [4]Department of Mechanical Engineering, University of Michigan, Ann Arbor, MI 48109, USA. ✉e-mail: kaji.bmc@tmd.ac.jp

placental cells would, therefore, provide a solution to many of these problems.

In the human placenta, the chorionic villus functions as a barrier. The villus is encased in a bilayer of trophoblast, with an inner cytotrophoblast (CT) layer, and an outer syncytiotrophoblast (ST) layer that is in direct contact with maternal blood[7]. CT cells have the capacity to fuse and give rise to multinucleated ST cells, which serve as the main barrier at the interface between mother and fetus[8]. This structural information spurred research into the development of in vitro placental barrier models. Barrier models, in which human choriocarcinoma cell lines such as BeWo and Jeg-3 are cultured on a permeable membrane (such as the microporous membrane of a Transwell), are simple and easy-to-use systems[9,10]. However, these cell lines are proven to have different functions from in vivo trophoblasts, and the cell fusion rates are not high[11]. Microfabrication technology has been exploited to enhance the functions of cell lines, giving rise to cultures that partially mimic the in vivo milieu[12–18]. Although microfabricated devices can be exploited to elucidate some elements of placental physiology and pathology, the enhancement of cell function is somewhat limited. In vitro models using human primary CT cells are often the model of choice to study the human placenta. However, limitations on the access to sufficient quantities of primary CT cells have prevented the development of placental barrier models and therefore limited our understanding of human placental physiology and function. Additionally, primary CT cells cannot be maintained for an extended period of time. Furthermore, although barrier models with primary CT cells have been reported[19,20], ST barrier models derived from primary cells that can be used to assess the translocation of chemicals have not been established so far.

To address this shortage, we first established human trophoblast stem (TS) cells from primary CT cells or blastocysts[21]. The TS cells have similar functions to CT cells and proliferate with a normal karyotype for at least 5 months. The TS cells can efficiently differentiate into ST cells in the presence of forskolin (an adenylyl cyclase activator), and gene expression patterns of the ST cells were similar to those of ST cells in vivo. Until the establishment of TS cells, many studies relied on the conversion of primed-pluripotent stem cells (PSCs) to provide a source of 'TS-like' cells. Examples of stimuli used to convert PSCs to TS-like cells include bone morphogenic protein and mechanical stress[22,23]. However, these TS-like cells have been problematic, and most of them do not meet the criteria of TS cells[21,24]. For example, the ST cells present in barrier models developed using TS-like cells exhibit a suboptimal low fusion rate[25]. Although using human TS cells we established would obviate some of the above issues, it has been challenging to use them to develop barrier models[26]. For example, when TS cells are induced to differentiate with forskolin, the resultant ST cells cannot survive until the base substrate is covered entirely with cells. Under these conditions, the ST cells do not proliferate but fuse and shrink slightly, resulting in a part of the dish surface not being covered by cells (Supplementary Fig. S1). Recently, TS cells were cultured in a microfluidic device and differentiated into ST cells under medium perfusion, but the ST differentiation ratio was around 50% in the device[27], limiting its use.

To address this issue, our group has focused on the process of ST turnover in the villus. During turnover, the ST cell layer forms small knots and releases them into the maternal blood, and progenitor CT cells underlying the ST cell layer fuse to replenish ST cells. Inspired by this phenomenon, we aimed to generate trophoblast organoid barrier models in which an ST cell layer is placed over undifferentiated TS cells. We hypothesized that an ST cell layer constantly supplied with cells from the underlying undifferentiated cell layer can serve as a placental barrier model that holds a stable barrier function without being affected much by ST morphological changes. The generation of trophoblast organoids using primary trophoblasts cultured in extracellular matrix (ECM) has been reported previously[28,29]. However, ST cells only formed inside organoids under these conditions and therefore the interface between mother and fetus was not recapitulated in this model system.

In this study, based on the hypothesis that TS cells recognize ECM as the basal scaffold and do not form an ST layer outside of cell constructs in ECM, we seeded cells to agarose plates having micro-scale size wells instead of seeding cells in ECM (Supplementary Fig. S2). The microwell plates of agarose was used to avoid attaching of cells to the substrate and enhancing cell-cell adhesion under the efficient supply of nutrients and oxygen through agarose. We first identified the cell culture conditions required to generate apical-out spherical trophoblast organoids in which an ST cell layer formed on undifferentiated TS cells. Subsequently, we constructed a placental ST barrier model based on the culture conditions of the apical-out spherical trophoblast organoids for quantifying the amount of drug that crosses the placental barrier. This model was designed with a column insert and a basal collagen membrane to facilitate the culture of ST cells, to allow the measurement of transepithelial/transendothelial electrical resistance (TEER), and to permit microscopical observation. On this column device, ST cells fused and formed three-dimensional structures, and the ST cells covered an area of nearly 100% on the undifferentiated cell layer. Barrier model function and integrity were characterized by measuring TEER, secretion of human chorionic gonadotropin (hCG), and drug permeability. Finally, we show an ST barrier model co-cultured with endothelial cells as a promising model recapitulating the human placental villus structure.

## Results

### Spherical trophoblast organoids covered with an ST cell layer

We first examined culture conditions for generating spherical cell aggregate-based trophoblast organoids, which we refer to as spherical trophoblast organoids here, that TS cells underlie the outer layer of ST cells. EGF and CHIR99021 (a Wnt activator) have been used to form trophoblast organoids and other types of human epithelial organoids[28,30,31]. CHIR99021 stimulates the proliferation of stem cells such as intestinal stem cells[32]. Y-27632 (a ROCK inhibitor) increases cell viability and the growth of trophoblast organoids[28]. With this information, we prepared a pre-culture medium and weak differentiation medium (PreM and W-DM, respectively; see Methods for full details). PreM enabled TS cells to form cell spheroids. W-DM, which contains CHIR99021 at a lower concentration than PreM, enhanced the differentiation of TS cells into ST cells. To identify the ST cells in the cell constructs, expression of syndecan-1 (SDC1) and the beta-subunit of human chorionic gonadotropin (hCG) were analyzed. SDC1, a cell surface heparan sulfate proteoglycan, is apically localized in the placental villi and is not detected in other cells in the placenta[33]. hCG comprises α- and β-subunits and has multiple functions such as hormone production and differentiation of ST cells[34]. SDC1 and the beta-subunit of hCG (CGB) can be used as ST maker proteins. The ST cells formed a single layer on the surface of spheroids. However, differentiation to ST cells was inefficient in the PreM and W-DM culture conditions, and some organoids were not covered with ST cells totally (Supplementary Fig. S3A, B). We prepared a third, 'strong differentiation medium (S-DM)'. S-DM contains 10% fetal bovine serum (FBS), which is a potent inducer of ST cells when used to treat primary trophoblasts in 2D cultures[20]. Although the combination of PreM and S-DM induced an ST cell layer, the surface of the organoids was not smooth compared with organoids made with PreM and W-DM (Supplementary Fig. S3C(a), (b)), which did not resemble the surface of the human placental villus.

A 3-step treatment with PreM, W-DM, and S-DM efficiently enabled TS cells to form spherical trophoblast organoids with an outer ST layer (Fig. 1A, B), whose average diameter was $461 \pm 45.6\ \mu m$ (mean ± SD) (Supplementary Fig. S4A–C). The single layer of ST cells was confirmed by staining frozen sections with antibodies for ST makers SDC1 or CGB (Fig. 1C, D), and the outer layer was double-stained with the antibodies (Fig. 1E). Some spherical organoids

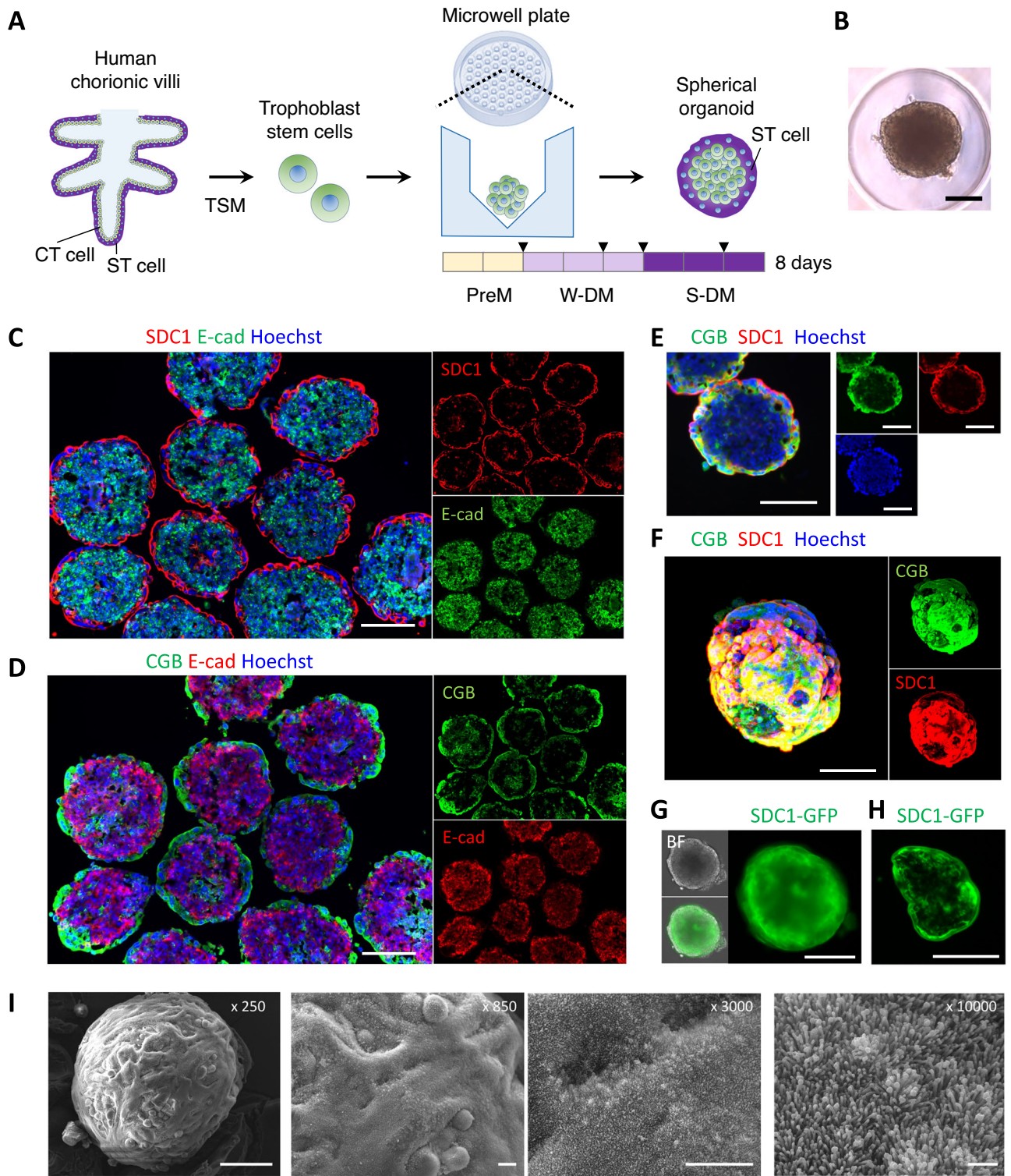

**Fig. 1 | Generation of apical-out spherical trophoblast organoids. A** The procedure for generation of the apical-out spherical trophoblast organoids. Triangular marks indicate medium exchange. A microwell plate was designed using SolidWorks 2019 (Dassault Systèmes SolidWorks Corporation) and Materialise MiniMagics23.5 (Materialise). **B** An image of a spherical trophoblast organoid in a microwell. **C**–**E** Immunostaining of syndecan 1 (SDC1), the beta subunit of the human chorionic gonadotropin (CGB), or E-cadherin (E-cad) in cross-sections of spherical trophoblast organoids. Images were obtained using BZ-X800/810 (Keyence). **F** Confocal images of a 3D structure of the organoid. The organoids of day 8 were fixed, permeabilized, and stained with antibodies for SDC1 and CGB. A series of confocal z-stack images were obtained using a Zeiss LSM 700 confocal microscope (Carl Zeiss), and a 3D image was reconstructed. **G** Epifluorescence and bright-field (BF) images of the organoid derived from SDC1-GFP (green fluorescent protein) TS cells that will express 7xGFP11 when the SDC1 promoter is activated and express GFP1-10 continuously (refer to Supplementary Fig. S7A, B). **H** A fluorescence image of a cross-section of the organoids derived from SDC1-GFP TS cells. **I** Scanning electron microscopy images of a trophoblast organoid. The scale bars indicate 200 μm (**B**–**H**), 100 μm (**I**, ×250), 10 μm (**I**, ×850 and ×3000), or 1 μm (**I**, ×10000). CT cytotrophoblast, ST syncytiotrophoblast, TSM trophoblast stem cell medium, PreM pre-culture medium, W-DM weak differentiation medium, S-DM strong differentiation medium. **B**–**I** Images from the organoids of day 8. Section samples were stained with antibodies in more than triplicates in over three independent experiments.

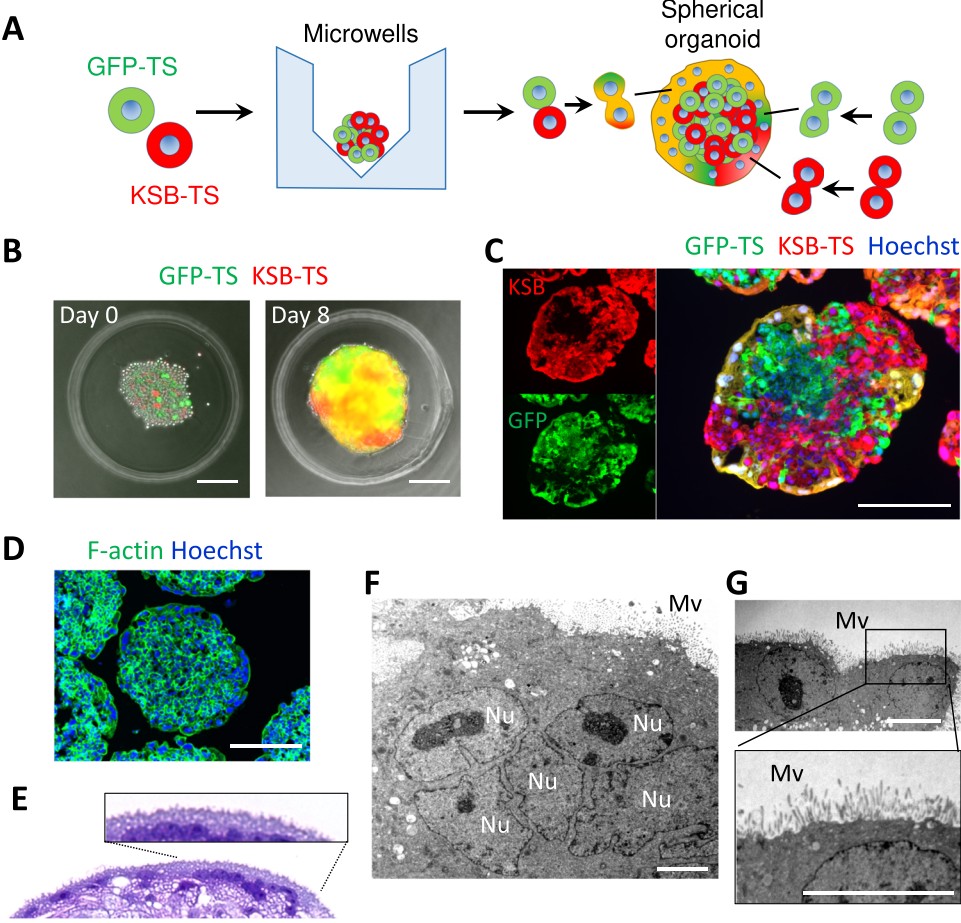

**Fig. 2 | Analysis of the fusion of ST cells in apical-out spherical trophoblast organoids. A** The procedure for generation of the trophoblast organoids using a mixture of GFP-TS cells and kusabira-orange (KSB)-TS cells. **B** An image of a trophoblast organoid in a microwell. **C** Fluorescence images of a cross-section of the organoids generated from a mixture of GFP-TS cells and KSB-TS cells. **D** Staining with phalloidin-FITC and Hoechst. **E** Staining with hematoxylin and eosin (H&E).

**F** Transmission electron microscopy images of the spherical organoids. The scale bars indicate 200 μm (**B**–**D**), 15 μm (**E**), 5 μm (**F**), or 10 μm (**G**). Images were taken using BZ-X800/810 (**B**–**E**) and JEM-1400Flash (**F** and **G**). ST syncytiotrophoblast, Nu nucleus, Mv microvilli on the surface of the organoids. **B**–**G** Images from the organoids of day 8.

included ST cells inside. Undifferentiated cells were stained with an antibody for E-cadherin and TP63, which were in accordance with the previous studies[28] (Fig. 1C, D and Supplementary Fig. S5A). Confocal microscopy showed that the ST cell layer covered almost the whole surface (Fig. 1F). In the human villus, GATA3 is expressed in CT and some ST cells in the first-trimester placentas, and positive cells especially are significantly decreased in the third-trimester placentas[35]. GATA3 was positive in both ST and undifferentiated cells in the spherical trophoblast organoids (Supplementary Fig. S5A), which characterize ST cells covering the organoids as more similar to those in the first-trimester placentas rather than those in the third-trimester placentas. Undifferentiated cells appeared to express Ki67 at higher levels than ST cells, but staining levels varied depending on individual cells (Supplementary Fig. S5A).

To confirm that this differentiation protocol can be applied to other TS cell lines, we used CT29 and CT30. Both cell lines also express SDC1 and E-cadherin as CT27 showed (Supplementary Fig. S5B). Moreover, we examined whether the spherical organoids can be passaged. The dissociated cells did not form the spherical organoids in the microwell plates probably because of the damage of some cells in the dissociation process (Supplementary Fig. S6A). Once the dissociated cells were re-cultured in 6-well plates (Supplementary Fig. S6A, B), the cells were able to form the spherical organoids (Supplementary Fig. S6C, D).

To further demonstrate the existence of ST cells in the outer layer of the spherical trophoblast organoids, we created SDC1-GFP TS cells that will have a fusion protein of SDC1-GFP when the *SDC1* promoter is activated (Supplementary Fig. S7A, B). Since it was difficult to introduce quite large size of DNA constructs into TS cells, we applied split fluorescent protein systems (Supplementary Fig. S7A, B). Spherical trophoblast organoids derived from SDC1-GFP TS cells showed green fluorescence around the surface of them (Fig. 1G, H), indicating that the *SDC1* promoter was activated on the organoid's surface in this culture schedule. Moreover, we analyzed the organoid's surface by scanning electron microscopy. We found that microvilli arose from the ST cell layer (Fig. 1I) and that those resembled the microvilli on the placental villus in vivo[36]. Taken together, these observations show that our 3-step treatment induces the formation of apical-out spherical trophoblast organoids.

### Fusion of ST cells at the surface of the spherical trophoblast organoids

To further demonstrate the ST formation and its fusion, we generated two kinds of genetically-modified TS cells, i.e., GFP-expressing TS cells and KSB (kusabira-orange)-expressing TS cells. If the two kinds of cells are fused, they should be seen with yellow color under fluorescence microscopy (Fig. 2A). As we expected, we were able to see a line of the ST cell layer showing yellow color, demonstrating that GFP-TS cells

and KSB TS cells were differentiated into ST cells and fused to make a layer and providing direct evidence of ST fusion (Fig. 2B, C). Results of F-actin staining with phalloidin-FITC (Fig. 2D) also indicated the fusion of the ST cells. H&E staining indicated that the surface of the ST cell layer of the organoids resembled that of human chorionic villi (Fig. 2E and Supplementary Fig. S9A). Results of transmission electron microscopy provided further evidence showing existence of multi-nucleated regions (Fig. 2F and Supplementary Fig. S8) and microvilli (Fig. 2G) in the ST layer.

## Consideration for developing trophoblast barrier models with column-type culture devices

Spherical models can be applied for the assessment of drug permeability. For example, blood-brain barrier (BBB) spheroidal models were reported as an in vitro screening platform for brain-penetrating substances[37,38]. However, this screening approach basically relies on a simple labeling of compounds with a fluorescence dye for measuring permeability, and is therefore only semi-quantitative. In contrast, models with a column insert are more appropriate for quantitative analysis of penetration. Therefore, we attempted to make column-type barrier models by applying our 3-step culture conditions protocol. First, a column insert device was fabricated (Fig. 3A, B). This device was designed for culturing TS cells on the outer surface of the collagen membrane to ensure maximal exposure of cells to medium (Fig. 3C). The device also facilitate the removal of dead cells. If some cells die on the membrane during cell cultures, the dead cells can fall into the bottom of wells by gravity. Moreover, the device allows the measurement of TEER without having to touch TS cells. TEER measurements in this study were conduced mainly to check robustness of the cell sheet and to obtain information about ST cell coverage on the circle aria of the collagen membrane, differentiation levels of TS cells, or thickness of cell sheets.

Four culture condition protocols were assessed with regard to cell integrity and differentiation into ST cells (Fig. 3D). Cells that were treated by the protocol (a) with PreM for 2 days, W-DM for 2 days, and S-DM for 3 days showed the highest TEER values (Fig. 3E). Treatment protocol (a) was the most efficient for producing ST cells (Fig. 3F), which formed 3D structures (Fig. 3G, H), that differed from the planar structures observed in 2D ST cultures with our previous protocol[21] (Supplementary Fig. S9B). Therefore protocol (a) was selected for generation of the ST cell barrier model. Confocal microscopy revealed that the barrier model has a single layer of ST cells on the undifferentiated cell layer, resembling the ST layer of the human chorionic villus (Fig. 3I). The single layer of ST cells was also confirmed by analysis of frozen sections (Fig. 3J).

In terms of medium composition, PreM and W-DM contains SB202190. Removal of SB202190 from these media rapidly induced ST differentiation, and a large number of cystic ST cells were observed (Supplementary Fig. S10A). Supplementation of SB202190 significantly reduced the number of cystic ST cells and enabled us to make a stable ST barrier. The results with FBS also indicated that rapid differentiation prevents the generation of ST barrier models. FBS strongly induces ST differentiation, and treatment with S-DM alone did not produce a robust ST layer (Supplementary Fig. S10B).

## Characterization of the placental barrier models

We next compared the ST barrier model and a TS cell sheet model with regard to barrier integrity and hCG secretion. Unlike ST cells, TS cells tended to pile up toward the collagen membrane and aggregated even though the polyimide film (yellow film) was coated with type I collagen (Supplementary Fig. S10C).

TEER values indicated that TS cells did not form an integrated barrier in TS medium (TSM). TEER values of ST barrier models were higher than those of TS barrier models (Fig. 3K). The TEER values were

markedly increased between day 4 and 6, indicating that the ST barrier matured over this time period. We observed that hCG secretion significantly increased between days 2 and 4 (Fig. 3L). The amount of hCG in the well was similar to that observed in cystic ST[21]. Since some TS cells can spontaneously differentiate into ST cells in TSM after the cells become a high density, we detected hCG in control samples of TS cells (Day 2–4) and the levels were lower than ST models (Day 2–4). Figure 3L shows the amount of hCG in columns or wells (mIU/h), which were normalized by the amount of culture medium. The amount of culture medium in the wells was 800 μL and that in the columns was 200 μL. Therefore, if concentrations of hCG were exactly the same in the wells and columns, the amount of hCG in wells should be 4-fold higher than that in columns. In control samples of TS cells (Day 2–4), the amount of hCG in the wells was 5.11-fold higher than that in the columns, i.e. the differences were 1.28-fold in concentrations, indicating that the TS cell sheet model barely functions as a barrier for hCG. Thus, these results demonstrate that the ST barrier model and TS cell sheet model are significantly different in barrier integrity and hCG secretion.

Whereas maternal IgG is selectively transported to the fetus, most IgA is considered not to be transferred to the fetus[39]. FcRn has been suggested as the mediator of the transplacental passage of IgG[40]. However, our previous results of RNA seq[21] interestingly show that both directly isolated primary ST cells and TS cell-derived ST cells express the *FCGRT* gene encoding for FcRn at very low levels (Supplementary Fig. S12A). Moreover, data in the Human Protein Atlas public repository (proteinatlas.org) also indicate that human ST cells express very little FcRn. These findings prompted us to examine the incorporation of IgG and IgA in our ST barrier model. Following exposure to FITC-labeled IgG or IgA, cells were washed and fixed, and fluorescence levels were analyzed. The levels of FITC-IgG and IgA were almost the same in the ST barrier models (Supplementary Fig. S12B). There was no significant difference in the fluorescence levels between controls (only collagen membrane) and TS cells cultured 1 day. Confocal microscopy confirmed that both IgG-FITC and IgA-FITC were internalized by ST cells (Supplementary Fig. S12C). These results raised the possibility that ST cells can incorporate IgA as well as IgG. The mechanisms by which IgA hardly translocates to fetal blood circulation may be attributed to processes after incorporation into ST cells.

## Trophoblast barrier models whose membranes are almost entirely covered with ST cells

The ST layer shown in Fig. 3F was not a continuous layer and may limit the applications. To efficiently find the culture conditions to make a barrier model entirely covered with ST cells, we generated SDC1-GFP TS cells that express SDC1-GFP11 under the control of the *SDC1* promoter and constantly express GFP1-10. Utilizing the SDC1-GFP TS cells, we do not need to confirm SDC1-positive cells by immunostaining in every culture condition tested. We modified the culture condition by changing S-DM to S-DM2 containing 2 μM forskolin, an activator of adenylyl cyclase, and by mixing fibronectin to a coating solution of Matrigel to increase the adhesion of cells to the collagen membrane (Fig. 4A, B). As minor modifications, the amount of medium in wells was changed from 800 μL to 1300 μL to equalize the water level in the inner column and outer well, and the culture schedule was shortened to 6 days from 7 days. The basal collagen membrane was almost entirely covered with SDC1-GFP TS cells expressing SDC1-GFP when cells were cultured in the abovementioned condition (Fig. 4C). The SDC1-GFP was specifically reacted with an antibody for SDC1, confirming the specificity of the antibody and localization of SDC1-GFP (Fig. 4D). Using a genetically normal TS cell line CT27, we eventually confirmed that the modified differentiation protocol works to generate placental barrier models totally being coved with ST cells (Fig. 4E). Other TS cell lines CT29 and CT30 showed the same ST

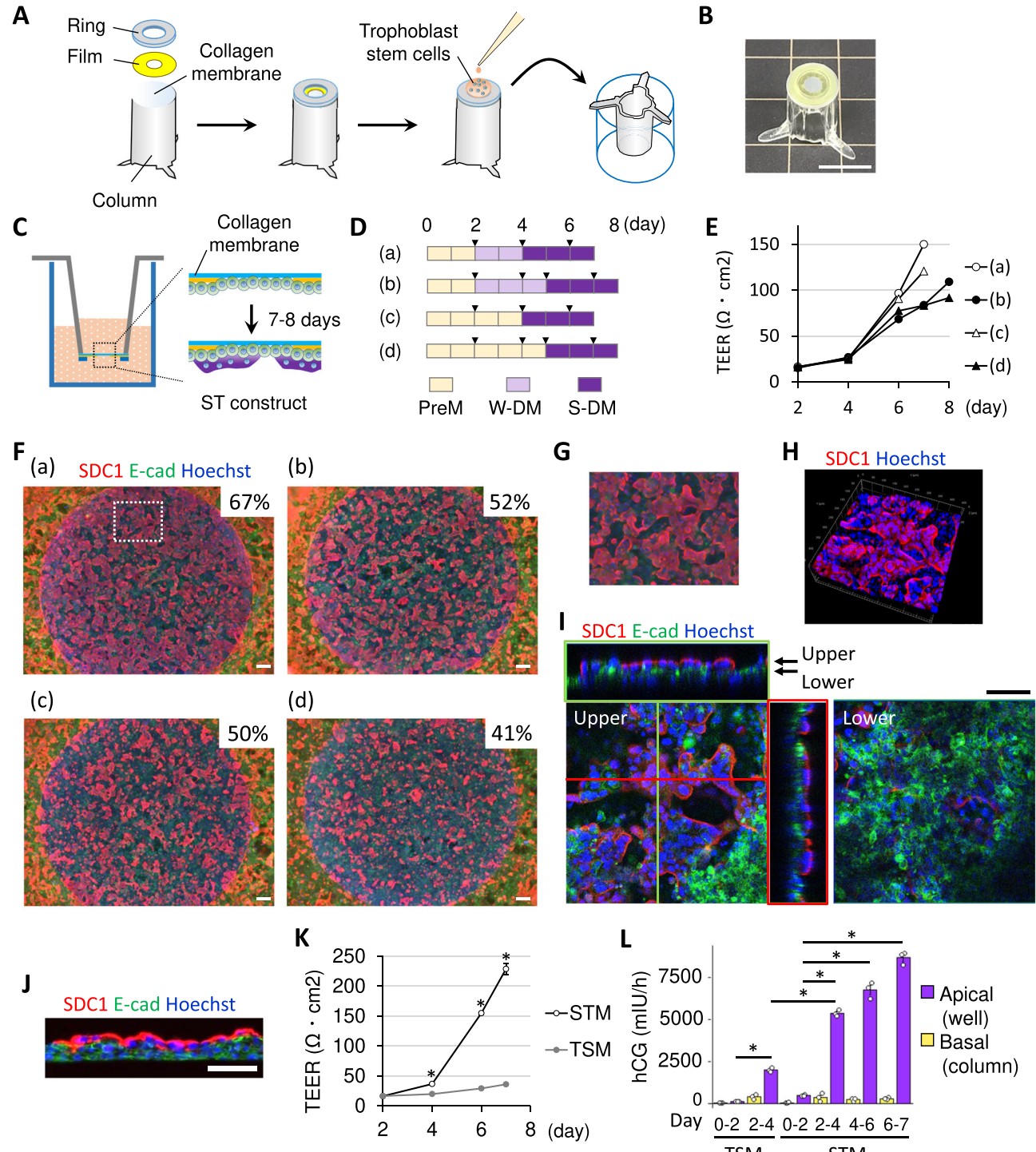

**Fig. 3 | Consideration for modeling the ST barrier. A** A schematic diagram of fabrication of a column insert and cell seeding. **B** An image of the column insert. **C** Schematic of ST barrier formation. **D** Culture schedules for ST barrier models. Triangular marks indicate medium exchange. **E** Transepithelial/transendothelial electrical resistance (TEER) values for the barrier models. **F–J** Immunofluorescence staining images of SDC1 and E-cadherin. **F** Percentage of the area covered with SDC1-expressing cells in the circular collagen membrane in each sample was determined using ImageJ version 1.47t (National Institutes of Health) (*N* = 1). **G** A partially magnified image of Fig. 3F(a). TS cells on the column device were cultured in PreM for 2 days, W-DM for 2 days, and S-DM for 3 days. **H** A 3D structure and (**I**)

cross-sections in ST barrier models analyzed using a Zeiss LSM 700 confocal microscope (Carl Zeiss). A sample that is shown in Fig. 3F(a) was analyzed. **J** Immunostaining of a frozen section of the ST barrier models. **K** TEER levels for the ST or TS models. TS cells were seeded at 1.2 ×10⁶ cells/mL ×40 μL/column on day 0 in this experiment (*N* = 7, STM; *N* = 3, TSM). *P* < 0.05, Student's *t* test (STM vs. TSM). STM, The three kinds of medium shown in Fig. 3D(a). TSM, trophoblast stem cell medium. Data are shown as mean ± SE. **L** Levels of secreted hCG (*N* = 3). *P* < 0.05, Tukey's multiple comparison test. Data are shown as mean ± SE. The scale bars indicate 1 cm (**B**), 200 μm (**F**), or 100 μm (**I** and **J**). **F, G, J** Images were taken using BZ-X800/810. N refers to replicates (biologically independent samples).

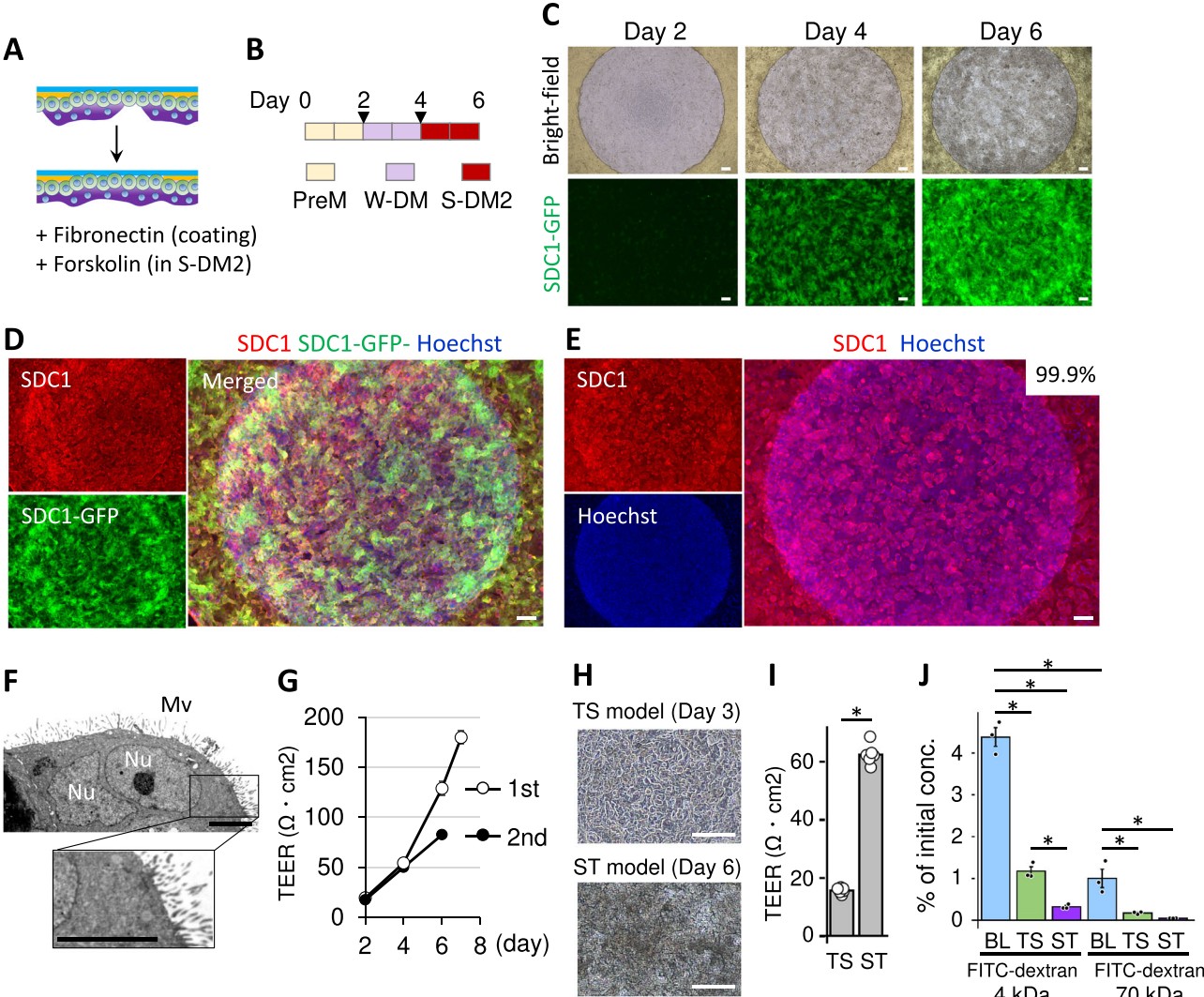

**Fig. 4 | Enhancement of ST coverage in the barrier models. A** Schematic of additional factors for ST barrier formation. **B** A culture schedule for the improved ST barrier model. Triangular marks indicate medium exchange. **C** SDC1 expression levels visualized using SDC1-GFP TS cells in the improved culture condition. **D** Immunostaining for SDC1 and fluorescence of SDC1-GFP in SDC1-GFP TS cells. **E** Immunostaining for SDC1 and fluorescence of Hoechst in TS cells (CT27 line). **F** Transmission electron microscopy images of the barrier models. **G** TEER levels for the previous 7-day culture model of Fig. 3D(a) (1st) and the improved 6-day culture model (2nd). Data are shown as mean ± SE (*N* = 4). **H–J** FITC-dextran permeability assay. Images of TS models and ST models (**H**), and TEER levels of both models (**I**, *N* = 6). **J** Percentage of transfer of FITC-dextran (4 kDa and 70 kDa) from the apical (well) to the basal (column). TS trophoblast stem cells, ST syncytiotrophoblast, BL blank (only collagen membrane). Data are shown as mean ± SE, which are percentages of concentrations against initial concentrations of FITC-dextran (*N* = 3). *$P < 0.05$, Tukey's multiple comparison test. The scale bars indicate 200 μm (**C–E**, and **H**),or 5 μm (**F**). Images were taken using BZ-X800/810 (**C–E**), JEM-1400Flash (**F**), or Olympus CKX53 (**H**). **D**, **E** Images from the barrier models of day 6. Nu nucleus, Mv microvilli on the surface of the organoids. **G**, **I**, **J** N refers to replicates (biologically independent samples).

coverage on the undifferentiated layer/the membrane, i.e. ca.100% (Supplementary Fig. S13A, B).

Next, we have examined the characterization of the improved barrier models. The ST cells derived from each of the three cell lines had microvilli, resembling ones in vivo, on their surface (Supplementary Fig. S13C). In section samples from these cell lines, multinucleated regions in ST cells were also observed (Fig. 4F and Supplementary Fig. S13D). We checked the barrier integrity of the improved models. Although TEER values of the improved 6-day culture models (2nd) were lower than those of the previous 7-day culture models (1st), the increase in TEER values was confirmed during the culture period (Fig. 4G). The modification of the culture protocol for the improved models, which was the use of fewer cells and more differentiative S-DM2, may have affected the thickness of cell sheets, resulting in lower TEER values in the improved models. To examine size-dependent paracellular transport, the ST barrier models and the control TS

models were exposed to two kinds of fluorescein isothiocyanate (FITC)-conjugated dextran (4 kDa or 70 kDa) after checking the integrity of cell sheets by microscopy and TEER measurements (Fig. 4H, I). FITC-dextran (70 kDa) has a molecular weight similar to human serum albumin, the most abundant protein in plasma, and it is often used to check cell sheet integrity[27,41]. FITC-dextran (70 kDa) hardly crossed ST barrier models, indicating the function of the cellular barrier (Fig. 4J).

## A co-culture barrier model with HUVEC

In the first trimester, the placental barrier consists of four layers, i.e., the ST layer, the CT layer, a stroma layer, and a fetal endothelial cell layer (EC)[42,43]. ST cells fuse and form a barrier without an intercellular gap, and endothelial cells have junctions consisting of wide zones of tight junctional regions discontinuously[44–47] and restrict translocation of chemicals to the fetal bloodstream. Our basic ST barrier model is

comprised of ST, TS, and Matrigel/collagen layers, but lacks the endothelial component. We therefore wanted to increase the physiological relevance of our ST barrier models by adding endothelial cells to mimic the human placental barrier. On day 2 of the culture schedule for setup of the ST cell barrier, GFP-HUVECs were seeded on the surface opposite to the one on which TS cells were cultured (Fig. 5A). GFP-expressing HUVECs were used to visually confirm the integrity of the cell sheet. TS cells differentiated into ST cells even in the presence of HUVECs and the EGM-2 medium (Fig. 5B). Formation of a single layer of ST cell barrier was confirmed by confocal microscopy (Fig. 5C, D). Scanning electron microscopy confirmed the microvilli arising from the ST cell layer (Fig. 5E). The TEER values in this ST/HUVEC co-culture were significantly higher than those in barrier models comprised of an ST/TS cell layer only (Fig. 5F). Similar tendencies were observed also using the ST barrier models created before improving ST coverage (Supplementary Fig. S14A–E). In the revised barrier models, TEER increased to 2.38-fold on day 6 by adding HUVECs to ST barrier models (Fig. 5F). In the previous models, the HUVEC addition increased TEER to 2.29-fold on day 6 and 1.40-fold on day 7 (Supplementary Fig. S14E).The mechanism of synergistic increases in TEER values after the co-culture is still unclear, but culture media or factors produced by HUVECs perhaps affected the tightness of the ST/TS barrier. Some factors in the ST/TS culture media may affect the integrity of HUVEC layers since treatment with PreM and W-DM induced a more planar morphology in HUVECs (Supplementary Fig. S15).

### Transportation of antipyrine, caffeine, and glyphosate

Antipyrine is an analgesic and nonsteroidal anti-inflammatory drug (NSAID), and caffeine is a central nervous system stimulant. They are widely used as reference drugs in ex vivo placental perfusion systems and cross the placental barrier by passive diffusion[48]. Glyphosate, a commonly used herbicide, has a significantly lower permeation rate in these systems. We used the improved ST barrier models to estimate apparent permeability coefficient (Papp) values of these compounds. The Papp values of antipyrine and caffeine were around 47.4 ($\times 10^{-6}$ cm/s) and 93.3 ($\times 10^{-6}$ cm/s), respectively (Fig. 5G). The Papp value of antipyrine was similar in a BeWo monolayer system [58.1 ($\times 10^{-6}$ cm/s)], but the value for caffeine [320 ($\times 10^{-6}$ cm/s)] was very different in this system when compared to our ST barrier models[49]. Barrier models with BeWo cells usually consist of a monolayer of cells, which are structurally different from our ST barrier models, and such difference may contribute to the difference of Papp values in both barrier models. Given that, in ex vivo placental perfusion, fetal side concentrations (% of substances added to the maternal side) were similar in antipyrine and caffeine[49], our ST barrier models showed trends more similar to ex vivo systems than BeWo systems. The Papp values of ST barrier models were significantly different from those of HUVEC barrier models or collagen membranes (Fig. 5G). Co-culture with HUVECs slightly increased the Papp values of antipyrine and caffeine obtained in ST cell models ($P < 0.05$, Student's $t$ test). EGM-2 for culturing HUVEC contains FBS at 2% like as S-DM/S-DM2 containing FBS at 10% for differentiating TS cells into ST cells. This FBS may affect the thickness of undifferentiated cell layers and the differentiation of ST cells in ST/HUVEC models, resulting in a slight change in Papp values. Proteins secreted from HUVEC may also affect ST barrier integrity. On the other hand, translocation of glyphosate was low in the ST barrier models [31.3 ($\times 10^{-6}$ cm/s)] or ST/HUVEC barrier models [29.2 ($\times 10^{-6}$ cm/s)] (Fig. 5G), which is in accordance with results of ex vivo perfusion studies[48]. The Papp value of glyphosate was 5.9 ($\times 10^{-6}$ cm/s) in the BeWo system[49], indicating that its passage is more slow in BeWo system than in our ST barrier models. Taken together, these results indicate that our placental barrier models offer significant advantages over ex vivo perfusion systems with regard to the measurement of Papp values and chemical screening.

We also examined Papp values of antipyrine, caffeine, and glyphosate for the ST models with S-DM and compared them to the above-mentioned results of the improved ST barrier models with S-DM2. Although Papp values for antipyrine and caffeine were not much different from the ST models with S-DM2 (Supplementary Fig. S14F–H), Papp values for glyphosate were estimated lower than 0.41 ($\times 10^{-6}$ cm/s) by calculation and measurement for the limit of detection (Supplementary Fig. S14I). The number of cells seeded and culture time (6 days or 7 days) aside from ratios of ST coverage probably affected the translocation of glyphosate critically. On the other hand, the transportation of chemicals in each model did not completely correspond to their TEER values. The discordance may come from the different nature between electricity and chemicals. The TEER value varies depending on factors such as the pathways in which electricity flows, i.e., intracellular, intercellular, and gaps between cells and substrates. It is thought that the thicker the cell layer in total, the higher the TEER value, and the more the ST is fused, the higher the TEER value. Small compounds cross the placental barrier depending on the property of the chemicals. Some chemicals cross the barrier through, for example, passive diffusion, facilitated diffusion, or active transport, and this translocation is different from how electricity crosses the cell layers.

## Discussion

The placenta is arguably the least understood organ in the human body[50], and establishment of robust trophoblast organoid models has great potential to further our understanding of human placental physiology. Although trophoblast organoids with internal ST cells have been created in ECM, organoids mimicking the outer layer of ST cells in the human chorionic villus have not been generated[28,29]. We focused on the observation that ECM provides a scaffold for cells but may affect the formation of external layers of ST cells. In fact, the removal of Matrigel does invert apical-basal polarity in models such as intestinal and gastric organoids[51,52]. These results suggest that cells recognize the 'Matrigel side' as a basal scaffold and that epithelial polarity can be changed reversibly by controlling the nature of the scaffold. In our current study, instead of culturing TS cells in Matrigel, we seeded them on agarose microwells where cells cannot attach to the substance and cultured them in three kinds of medium. This engendered the formation of spherical aggregates. These structures have a layer of ST cells that express SDC1 and CGB, and exhibit microvilli similar to human chorionic villi. The fusion of ST cells on the surface of the cell construct was demonstrated using a mixture of GFP-expressing TS cells and KSB-expressing TS cells. Although the mechanism of single layer formation remains unclear, we have identified culture conditions for the robust and reproducible generation of trophoblast organoids with an outer layer of ST cells.

The achievement of reversing the apicobasal polarity of trophoblast organoids will open up a new avenue for understanding the events at the fetal-maternal interface. During pregnancy, the fetal ST cells of villi contact with the maternal decidua derived from maternal endometrium. Cells originating from villous cytotrophoblasts in the anchoring villus give rise to extravillous trophoblasts (EVT)[53], which remodel the maternal spiral artery to enlarge it for placental development. Although such a dynamic event has yet to be recapitulated in vitro, it could be partially reconstructed using apical-out trophoblast organoids and decidua cells. The fetal ST cells interact with not only decidua but also maternal blood that potentially contains a variety of xenobiotics including pathogens. For example, infection of the placenta with Zika virus is reported to increase the incidence of fetal microcephaly and brain malformations[54–56], and infection with SARS-CoV-2 virus that leads to coronavirus disease 2019 (COVID-19) was reported to associate with a higher rate of pregnancy complications[57]. Recently, stem-cell-derived trophoblast organoids having ST cells inside were exposed to each virus and were shown to

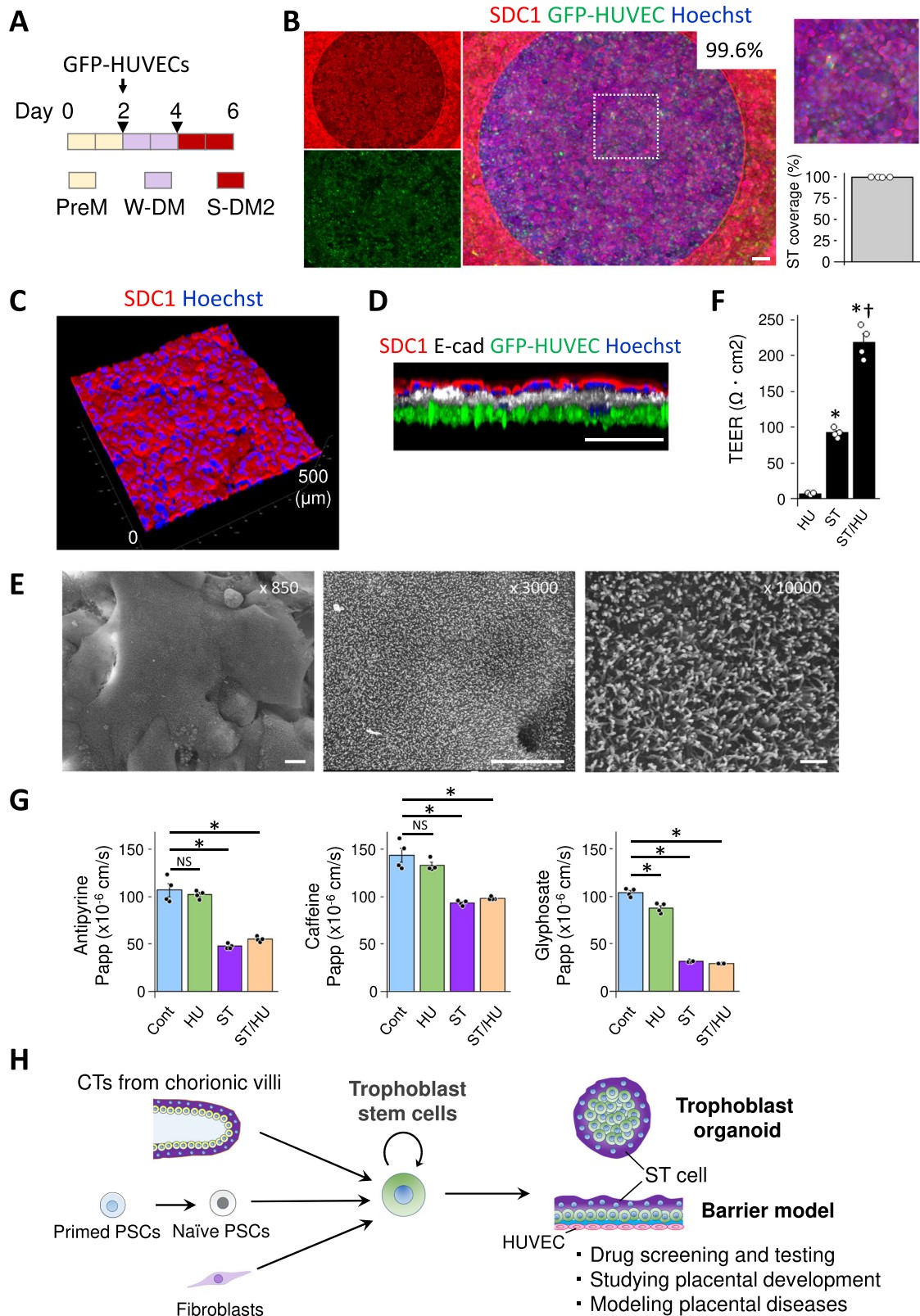

**Fig. 5 | ST/HUVEC co-culture barrier models and translocation of drugs. A** The strategy of co-culture for ST/HUVEC models. Triangular marks indicate medium exchange. **B** ST/HUVEC models were generated according to the strategy shown in Fig. 5A. Immunostaining of SDC1. The right panel shows a partially magnified image taken using BZ-X800/810. The average ST coverage was 99.6 ± 0.0973% (mean ± SE, $N = 4$). ST/HUVEC models in a perspective view (**C**) and in a side view (**D**) by confocal microscopy with TCS SP8 (Leica). **E** Scanning electron microscopy images of a ST/HUVEC barrier model. **F** TEER measurements in each barrier model ($N = 4$).

$*P < 0.05$, Dunnett's test. $†P < 0.05$, Student's $t$ test (ST vs. ST/HUVEC). Data are shown as mean ± SE. **G** The permeability of reference compounds in control (Cont, collagen membrane), HUVEC (HU), ST, and ST/HUVEC (ST/HU) models ($N = 4$). $*P < 0.05$, Dunnett's test. Data are shown as mean ± SE. **H** Potential applications of human trophoblast organoids and ST barrier models. The scale bars indicate 200 μm (**B**), 200 μm (**D**), 10 μm (**E**, ×850 and ×3000), or 1 μm (**E**, ×10,000). Data of ST or ST/HUVEC models were obtained from the models of day 6. **F**, **G** N refers to replicates (biologically independent samples).

have limited susceptibility to SARS-CoV-2 compared with Zika virus[58], but detail mechanisms should be revealed using apical-out trophoblast organoids. In addition to such viruses, infections with a protozoan parasite *Toxoplasma gondii*, which is a major source of congenital disease, can be transmitted to the fetus, and studies using primary trophoblast have shown that susceptibilities of ST cells were lower than CT cells[59,60]. As for *Porphyromonas gingivalis* associated with periodontal disease, animal experiments and meta-analysis of medical databases for human observational studies show that periodontitis increased the risk of preterm birth and low birth weight and caused inflammation in the placenta and umbilical cord[61,62]. The infection mechanisms of these parasites and bacteria remain poorly understood, and our accessible and reproducible 3D models would help to understand those things.

The quantitative analysis of transplacental passages of chemicals is crucial in drug development since maternally circulating drugs can result in severe adverse effects on the fetus. Ex vivo perfusion systems with human placental explants can provide information about it but have challenges such as the preparation of fresh placentas with written informed consent and procedures with a high leakage or failure rate and are not appropriate for screening[3,6]. Current in vitro barrier models with cancer cell lines have different nature from in vivo trophoblasts, and the cell fusion rates are not high[11]. Although ST cells derived from human TS cells are an appropriate resource for creating barrier models, but it has been challenging to control differentiation into ST cells for the models (Supplementary Fig. S1). We here have attempted to develop column-type barrier models with TS cells to predict the transplacental passage of chemicals. The culture conditions we used to generate apical-out spherical trophoblast organoids were compatible with the generation of a placental barrier model. We used PreM, W-DM, and S-DM. PreM and W-DM contain SB202190, a potent p38 MAPK inhibitor. p38 MAPK signaling enhances ST differentiation in isolated primary trophoblasts[63] and a cell line[64]. Supplementation of SB202190 was essential to reduce arising 3D cystic ST cells by repressing fast differentiation into ST cells (Supplementary Fig. S10A). CHIR99021 enhanced cell proliferation at a concentration of 2 μM, but lower concentrations induced the differentiation of the outer cells into ST cells, albeit at a rate slower than that associated with FBS. FBS worked as a potent inducer for ST cells but was not able to be used to produce a stable ST cell sheet in the case without a TS cell bed on the collagen membrane (Supplementary Fig. S10B). Based on these findings, we gradually differentiated TS cells into ST cells with three kinds of medium, i.e. PreM, W-DM, and S-DM, and obtained an ST barrier model. The ST model engendered partial but sustained release of ST cell aggregates, which simulated cell turnover and a behavior partially resembling the release of ST knots that is observed in human placental villi (Supplementary Fig. S11A, B). To improve the ST coverage up to about 100%, we modified the medium S-DM to S-DM2 containing forskolin. Although the treatment of cells with forskolin alone cannot be appropriate to make a stable cell layer of ST cells that can be used as a barrier model (Supplementary Fig. S1), forskolin worked for our column having an undifferentiated cell bed. In that situation, the cell bed may function as a source of undifferentiated TS cells for replenishing ST cells and as a substrate more sticky than artificial collagen membrane coated.

Our barrier models were validated using model chemicals, i.e. antipyrine, caffeine, and glyphosate, that have been used for ex vivo perfusion systems. In previous ex vivo studies, results are shown as percentages of the fetal-side amount against the amount of substances added to the maternal side at time zero. The fetal percentage of glyphosate was about 3.6% at 30 min after initiation and the percentage of caffeine or antipyrine was about 6-fold higher (about 23%) than that. In our ST barrier models, Papp values of caffeine were about 3-fold higher than those of glyphosate, although those of antipyrine were only 1.5-fold increase than that of glyphosate. Thus, our barrier models appear

to indicate a similar tendency in transportation but still have some differences from ex vivo perfusion systems. These gaps could be improved by adding other cellular components such as fibroblast and also by optimizing addition of HUVECs. Since additional HUVECs increased TEER values, the co-culture of the ST barrier layer with HUVECs should have strengthened the ST barrier. However, a part of the HUVEC layer appeared not to form a continuous layer, which was thought by results in Fig. 5B. To promote the formation of the HUVEC layer, we tried to use a conditioned medium that was obtained from normal human lung fibroblasts and contained abundant VEGF, but the promotion effect was quite limited in ST/HUVEC models (Supplementary Fig. S16A−C). To address these limitations of this study and to create a continuous complete layer of HUVECs along with the ST barrier, optimization of the seeding number and timing of HUVECs need to be conducted in future studies.

We have established both trophoblast spherical organoids and barrier models using TS cells. The barrier models facilitated the measurement of drug permeability and also enabled the analysis of interactions between the ST barrier and maternal molecules such as IgG and IgA. These organoids and barrier models should be valuable tools for understanding intricate placental physiology. Our use of TS cells to generate ST cells may be applicable to (and extend the utility of) future 'placenta-on-a-chip' studies. Here, we used TS cells from CT cells; we previously demonstrated that TS cells can be established from blastocysts[21] and complete hydatidiform moles[65]. Recent advances have made it possible to generate TS cells[66] or trophectoderm[67] from naïve PSCs. In addition, TS cells can be generated by direct reprogramming from fibroblasts[68]. Together, these data show that human TS cells can be generated from many sources, including patient-derived cells. Our organoid technology may lead to the generation of patient-derived trophoblast organoids, which may further elucidate the mechanisms underlying placental pathology, and facilitate the development of new drugs (Fig. 5H).

# Methods

## Ethical approval
TS cells and human placentas were used based on the approval the Ethics Committee of Tohoku University School of Medicine (Research license 2014-1-879).

## Fabrication of agarose microwell plates
Agarose microwell plates (61 wells) were fabricated using a 3D-printed mold and a polydimethylsiloxane (PDMS) mold, as illustrated in Supplementary Fig. S2. First, a microwell plate was designed using Solid-Works 2019 (Dassault Systèmes SolidWorks Corporation). A 3D-printed mold of a microwell plate was made with a printer (QIDI TECH Shadow 5.5S printer), poly(ethylene glycol) diacrylate (PEGDA, Mn: 250) as a resin, 1%(w/w) photoinitiator (Omnirad 819), and 1% (w/w) photosensitizer (2-Isopropylthioxanthone). The mold was immersed in ca.100 % ethanol for more than 1 min and was exposed to UV light for 4 min, then heated at 80 °C overnight. The mold was coated with Parylene (DPXC, CAS No. 28804-46-8; Parylane Japan) using a PDS-2010 Labcoter 2 (Specialty Coating Systems Inc.). Parylene was used for making PDMS (Silpot184; Toray-Dow Corning) easy to peel off. A PDMS mold was fabricated by conventional soft lithography. PDMS (elastomer: curing agent = 10:1) was cast into the parylene-coated molds and heated to solidify them. The solidified object was taken off from the molds. Agarose (2%) (SeaKem GTG Agarose, Lonza) was poured into the PDMS mold. After solidifying the agarose, the agarose microwell plate was removed from the mold and then immersed in TS basal medium until use.

## Fabrication of a column-type device for barrier models
Column inserts for culturing TS cells were fabricated by modifying columns that contained a vitrified collagen membrane (ad-MED

Vitrigel 2 for 24-well plates, Kanto chemical co., Inc.) as reported by Takezawa et al.[69]. The collagen vitrigel membrane functions as a permeable substrate. As shown in Fig. 3A, a ring of polyimide film (t, 5 μm; outer diameter, 8 mm; inner diameter, 3 mm) was bonded to the collagen membrane. To place cell suspension on the collagen membrane, a silicone ring (t, 0.5 mm; outer diameter, 8 mm; inner diameter, 5 mm) was bound onto the polyimide ring using PDMS solution. The column was heated at 60 °C overnight to solidify the PDSM solution and exposed to UV for about 20 min.

## Cells and culture conditions

Human TS cells were established in our previous studies[21], which were derived from CT cells isolated and purified from first-trimester placentas (6–9 weeks gestation). Experiments using human placentas were conducted according to protocols approved by the Ethics Committee of Tohoku University School of Medicine (Research license 2014-1-879). TS cells were cultured based on a previous method[21]. Briefly, TS cells were maintained in TS medium (TSM; DMEM/F12 supplemented with 0.15% BSA, 50 units/mL penicillin, 50 μg/mL streptomycin, 1% ITS-X, 1% KSR, 0.2 mM L-ascorbic acid, 2.5 μM Y-27632, 25 ng/mL EGF, 0.8 mM VPA, 5 μM A83-01, and 2 μM CHIR99021) containing 0.25 μg/mL iMatrix-511. Experiments were conducted using a cell line CT27, except for experiments with CT29 and CT30 to confirm reproducibility. Green fluorescent protein-expressing human umbilical vein endothelial cells (GFP-HUVECs) were purchased from Angio-Proteomie (cAP-0001GFP) and cultured in endothelial cell growth medium 2 (EGM-2, Lonza). All cells were cultured at 37 °C with 5% $CO_2$.

## Insertion of 7xGFP11 into the endogenous *SDC1* locus

The puromycin-resistant gene (Puro[R]) and *Cas9* were PCR amplified from pGIPZ vector (Open Biosystems) and Alt-R S.p. Cas9 Expression Plasmid (IDT), respectively. Puro[R], a T2A element, and *Cas9* were cloned into the CS-CA-MCS plasmid (kindly provided by H. Miyoshi, RIKEN BioResource Center, Ibaraki, Japan) using the In-Fusion HD Cloning kit (Takara). The resulting vector was designated as pCS-CA-Puro[R]-T2A-Cas9. The kanamycin/neomycin resistance gene and the pUC origin were amplified from pCAG-HIVgp (kindly provided by H. Miyoshi), and the human U6 (hU6) promoter was amplified from pCS-hU6 vector[65]. The kanamycin/neomycin resistance gene, the pUC origin, the hU6 promoter, 7xGFP11[70], and a loxP sequence were assembled using the NEBuilder HiFi DNA Assembly Cloning Kit (NEB). A sgRNA targeting the last exon of *SDC1* (target sequence: 5′- GAG CCG AAA CAA GCC AAC GG −3′), a loxP sequence, and a gRNA target sequence (5′-CCC CCG TTG GCT TGT TTC GGC TC −3′; the PAM sequence is underlined), were inserted between the hU6 promoter and 7xGFP11 to generate pU6-sgSDC1-loxP-7xGFP11-loxP. The Cre recombinase gene was amplified from the Cre Vector (Santa Cruz) and cloned into the pCS-3G vector[65] to generate pCS-3G-Cre. *GFP10*[70] was chemically synthesized and cloned into the CS-CA-MCS plasmid to generate pCS-CA-GFP10. Sequences of the primers used for the vector construction are shown in Supplementary Data 1.

pCS-CA-Puro[R]-T2A-Cas9 was cotransfected with pCMV-VSV-G-RSV-Rev and pCAG-HIVgp (kindly provided by H. Miyoshi) into 293T cells and a lentivirus expressing Puro[R]-T2A-Cas9 was prepared. A lentivirus expressing GFP10 was also prepared from pCS-CA-GFP10 in the same way. These lentiviruses were transduced into CT27 cells, and Cas9-expressing cells were selected with 2 mg/mL puromycin for two days. The pU6-sgSDC1-loxP-7xGFP11-loxP vector was transfected into the Cas9-expressing CT27 cells using an electroporator (CUY21Vitro-EX; BEX) with the following setting: Poration pulse [125 V, 10 ms]; Driving pulse [20 V, 50 ms, 5 pulses]. After 200 mg/ml G418 selection for 5 days, lentivirus expressing Cre was transduced into the cells to delete the kanamycin/neomycin resistant gene, the hU6 promoter, and the sgRNA sequence. Then, single-cell cloning was performed, and

clones with an in-frame insertion of 7xGFP11 were identified by Sanger sequencing. One of the obtained clones was used for the experiments. This knock-in strategy is based on homology-independent targeted integration[71].

## Isolation of TS cell clones expressing EGFP and Kusabira Orange (KSB)

*EGFP* was PCR amplified from pEGFP-N1 (Clontech) and cloned into the CS-CA-MCS plasmid to generate pCS-CA-EGFP. *KSB* was PCR amplified from Kusabira Orange-N1[72] (a gift from Michael Davidson & Atsushi Miyawaki; Addgene plasmid # 54793; http://n2t.net/addgene:54793; RRID: Addgene_54793) and cloned into the CS-CA-MCS plasmid to generate pCS-CA-KSB. Lentiviruses expressing EGFP or KSB were prepared as described above and transduced into CT27 cells. Then, single-cell cloning was performed to isolate EGFP- or KSB-expressing CT27 cell clones.

## Generation of spherical trophoblast organoids

TS cells or SDC1-GFP TS cells were seeded at 1.1 ×10[5] cells/mL ×200 μL on the agarose microwell plate and centrifuged at 160 × *g* for 20 sec. Then 2 mL of a pre-culture medium (PreM; DMEM/F12 supplemented with 0.15% BSA, 50 units/mL penicillin, 50 μg/mL streptomycin, 1% ITS-X, 1% KSR, 0.2 mM L-ascorbic acid, 2.5 μM Y-27632, 25 ng/mL EGF, 20 ng/mL BMP4, 50 ng/mL bFGF, 0.1 μg/mL heparin, 2 μM CHIR99021, and 2 μM SB202190) was added to each well of 12-well plates. The culture medium was replaced with 2 mL/well of a weak differentiation medium (W-DM; DMEM/F12 supplemented with 0.15% BSA, 50 units/mL penicillin, 50 μg/mL streptomycin, 1% ITS-X, 1% KSR, 0.2 mM L-ascorbic acid, 2.5 μM Y-27632, 25 ng/mL EGF, 20 ng/mL BMP4, 50 ng/mL bFGF, 0.1 μg/mL heparin, 0.5 μM CHIR99021, and 2 μM SB202190) on days 2 and 4. Then culture media were replaced with 2 mL/well of a strong differentiation medium (S-DM; DMEM supplemented with 10% FBS, 100 units/mL penicillin, and 100 μg/mL streptomycin or DMEM/F12 supplemented with 10% FBS, 50 units/mL penicillin, and 50 μg/mL streptomycin) on days 5 and 7. On day 8, spherical trophoblast organoids were obtained.

To demonstrate the fusion of ST cells, KSR-expression TS cells and GFP-expressing TS cells were mixed at the ratio of 1:2 (KSR:GFP), seeded at 1.1 ×10[5] cells/mL ×200 μL on the agarose microwell plate, and cultured in the same manner described above.

## Generation of ST barrier models

The column-type device was used to develop placenta barrier models (Fig. 3A, B). Each column was placed with a collagen membrane on top. To promote cell attachment and growth, the collagen membrane was coated with 40 μL of 0.2 mg/mL Matrigel/PBS(-) at 37 °C for 30 min, followed by washing the membrane with PBS(-) twice. TS cells were seeded at 1.0 ×10[6] cells/mL ×40 μL/column on the collagen membrane as shown in Fig. 3A. The cells were incubated for 5 h at 37 °C and 5% $CO_2$ to allow cells to attach to the membrane. Then, columns were inserted into each well of a 24-well plate, and PreM was added to each column (200 μL) and well (800 μL). Cells were cultured using the three kinds of medium according to the culture schedule illustrated in Fig. 3D. For a generation of ST/HUVEC barrier models, TS cells were seeded on Day 0 on the collagen membrane from the outside of the column insert. On day 1, HUVEC were seeded at 0.2 ×10[6] cells/mL ×200 μL/column within the column insert. TS cells were cultured along with the 3-step culture schedule with PreM, W-DM, and S-DM and HUVECs were cultured within the insert with EGM-2.

To increase the ST coverage in the column device, we modified the culture condition for ST barrier models. Human fibronectin (Promo Cell) (0.05 mg/mL) was mixed in the coating solution of 0.2 mg/mL Matrigel/PBS(-), applied to the column, and incubated at 37 °C for over 1 h. Following removal of the coating solution, TS cells were seeded at 0.8 ×10[6] cells/mL ×40 μL/column to the collagen

membrane. After preculturing for 5 h, columns were inserted into each well of a 24-well plate, and PreM was added to each column (200 μL) and well (800 μL). After 2 days, the medium was replaced with W-DM for each column (200 μL) and well (1300 μL). Following 2 days, cells were cultured with the same amount of S-DM2 containing 2 μM forskolin for 2 days. For co-cultures, GFP-HUVEC was seeded at 0.2–0.3 ×10⁶ cells/mL ×200 μL/column to the column insert 2 days after seeding of TS cells. TS cells were cultured along with the 3-step culture schedule with PreM, W-DM, and S-DM2, and HUVECs were cultured within the insert with EGM-2.

### Preparation of sections

For frozen sections, TS organoids were fixed with 4% paraformaldehyde (PFA) for 1 h and then permeabilized with 0.3% Triton X-100 for 1 h. Cells were washed with PBS(-) twice and treated with 15% sucrose/PBS(-) at 4 °C for over 5 h. Then, the cells were treated with 30% sucrose/PBS(-) at 4 °C overnight. Cell samples were embedded in Tissue-Tek OCT compound (Sakura Finetek Japan Co., Ltd.), frozen on dry ice, stored at −20 °C, and then sectioned using a cryostat (Leica CM1950 or Tissue-Tek Polar-D) at a thickness of about 12 μm. The frozen sections were mounted on glass slides, air-dried for 1 h, and stored at 4 °C until use.

For hematoxylin-eosin (H&E) sections, TS organoids were fixed with 4% paraformaldehyde (PFA) for 60 min and stored in 1% formalin neutral buffer solution (Fujifilm Wako), followed by conventional paraffin embedding and H&E staining (Advantec).

### Immunofluorescent staining

Cells on column-type devices were fixed with 4% PFA for about 20–40 min at room temperature (RT). After washing cells with PBS(-), the cells were treated with 0.3% Triton X-100/PBS(-) for 40 min at RT. Cells were washed with PBS(-) and incubated with a primary antibody at 4 °C overnight. The following primary antibodies were diluted with PBS(-) containing 0.1% Tween 20 and 2% FBS. PE-conjugated anti-SDC1 (Clone 44F9) (Miltenyi Biotec) (1:500 dilution), anti-CDH1 (#24E10, CST; Cell Signaling Technology) (1:250-1:500), anti-CDH1 (Clone SHE78-7) (#M126, Takara) (1:800), anti-CGB (#IR508, Dako) (1:10), anti-GATA3 (#5852, CST) (1:200), anti-TP63 (#ab124762, Abcam)(1:400), anti-Ki-67 (#M7240, Dako)(1:200), rabbit IgG Isotype control (#3900, CST) (1 μg/mL), and mouse IgG Isotype control (#5415, CST) (1 μg/mL). After being washed with PBS(-), cells were treated with the following secondary antibodies at RT for 1 h. Alexa Fluor 488 conjugated anti-rabbit IgG (#4412, CST) (1:400), Alexa Fluor 555 conjugated anti-mouse IgG (#4409, CST) (1:400), and Alexa Fluor 647 conjugated anti-rabbit IgG (#4414, CST) (1:400) were used. Nuclei were stained with Hoechst 33258 (Dojindo) (1:1250) or Hoechst 33342 (Dojindo) (1:1000). Cells were analyzed using a BZ-X800 or X810 All-in-one microscope (×4, ×10, or ×20 objectives) (Keyence), a Zeiss LSM 700 confocal microscope (Carl Zeiss), or TCS SP8 (Leica). For staining F-actin, frozen sections were treated with 1 μg/mL phalloidin-FITC for 60 min at RT, followed by washing with PBS(-). For staining cells of frozen sections, cells were washed with PBS(-) twice and followed by the same staining procedure as cells on the column-type devices were subjected to. Samples of ST barrier models and section samples of spherical trophoblast organoids were stained with antibodies for SDC1, CGB, or E-cadherin in more than triplicates in over three independent experiments.

### Scanning electron microscopy

Cell samples were fixed in 4% PFA for 40 min. The samples were washed overnight at 4 °C in 0.1 M phosphate buffer and post-fixed with 1% OsO4 buffered with 0.1 M phosphate buffer for 2 h. The samples were dehydrated in a graded series of ethanol and dried in a critical point drying apparatus (JCPD-5;JEOL) with liquid CO₂. The cell samples

were spatter-coated with platinum and examined by scanning electron microscope (JSM-7900F; JEOL, Tokyo, Japan).

### Transmission electron microscopy

Cell samples were fixed in 2.5% glutaraldehyde in 0.1 M phosphate buffer for 2 h, and washed with 0.1 M phosphate buffer, post-fixed in 1% OsO4 buffered with 0.1 M phosphate buffer for 2 h, dehydrated in a graded series of ethanol and embedded in Epon 812. Ultrathin sections, 70 nm, were collected on copper grids, double-stained with uranyl acetate and lead citrate. and then examined by a transmission electron microscope (JEM-1400Flash, JEOL, Japan).

### TEER measurement

TEER was measured with an epithelial voltohmmeter (EVOM)2 (World Precision Instruments) and STX2 Chopstick electrodes to evaluate barrier integrity. The TEER values of cell samples were normalized to the values of controls without cells.

### Measurement of hCG

TS cells were seeded at 1.2 ×10⁶ cells/mL ×40 μL/column onto a membrane coated with Matrigel in column-type devices. Cells were cultured with three kinds of medium [PreM (2 days), W-DM (2 days), and S-DM (3 days)] as illustrated in Fig. 3D(a). Media were collected from columns and wells on days 2, 4, 6, and 7. For controls, the same number of TS cells were seeded onto a membrane coated with 0.3 mg/mL collagen (I-AC, Koken, Japan) and cultured in TSM for 2 or 4 days. The amount of secreted hCG was determined using an enzyme-linked immunosorbent assay (ELISA) kit (#KA4005, Abnova).

### IgG and IgA incorporation analysis

ST barrier models were prepared according to the culture schedule illustrated in Fig. 3D(a). As controls, TS cells were seeded at 1.0 ×10⁶ cells/mL ×40 μL/column on the collagen membrane coated with 0.3 mg/mL collagen solution and cultured for 1 day in TSM. IgG-FITC from human serum (#F9636, Merck) and FITC-conjugated ChromPure Human IgA (#099-090-011, Jackson Immuno Research) were diluted to 80 μg/mL with an assay buffer (HBSS containing 0.15% BSA, 1.6 g/L glucose, 2 mM L-glutamine) that was prepared with a modified ex vivo perfusion buffer[73]. Cell samples were washed once with the assay buffer, and then columns were inserted into wells of 24-well plates. The assay buffer was added to each column (200 μL), and buffer containing FITC-IgG or IgA was added to each well (1300 μL). Cells were incubated at 37 °C and 5% CO₂ for 3 h. Cells were fixed with 4% PFA and washed with PBS(-).

### FITC-dextran permeability assay

ST barrier models were prepared according to the culture schedule illustrated in Fig. 4B. To prepare TS barrier models, TS cells were seeded at 0.8 ×10⁶ cells/mL ×40 μL/column on the collagen membrane coated with the mixture of human fibronectin (0.05 mg/mL) and Matrigel/PBS(-) (0.2 mg/mL) and cultured for 3 days in TSM with 0.25 μg/mL iMatrix-511. Culture mediums were changed every two days for each ST or TS model. Each barrier model was washed once with the assay buffer mentioned above. FITC-dextran (4 kDa and 70 kDa, Sigma Aldrich) was prepared at 0.4 mg/mL with the assay buffer. The fresh assay buffer was added to each column (200 μL), and buffer containing FITC-dextran was added to each well (1300 μL). Cells were incubated at 37 °C and 5% CO2 for 1.5 h. Then, FITC-dextran in each insert column was collected. FITC-dextran in the buffer samples was quantified by measuring FITC using a fluorescence microplate reader (SpectraMax Gemini XPS).

### Translocation analysis

HUVEC, ST, and ST/HUVEC barrier models were prepared. After washing cells, the assay buffer was added to each insert (200 μL), and

buffer containing 100 µg/mL antipyrine (Fujifilm Wako chemicals), 200 µM caffeine (Fujifilm Wako chemicals), or 200 µM glyphosate (Fujifilm Wako chemicals) was added to each well (1300 µL). After 30 min, the buffers were collected for HPLC analysis. For time course experiments, cells were exposed to 100 µg/mL antipyrine for 2, 30, 60, 120, 180 min.

The apparent permeability coefficient (Papp) was calculated as follows: Papp (cm/s) = $(dQ/dt)/(A \times C_0)$, where $dQ/dt$ is the rate of the substrate appearance in the column insert (nmol/s), A is the surface area of the basement collagen membrane ($cm^2$), and $C_0$ is the initial concentration of the substrate in the well (µM)[19].

## HPLC analysis

Analysis of antipyrine was performed using Agilent 1200. The Quaternary Pump G1311A was used in combination with a Variable Wavelength Detector (VWD) G1314B and TCC (Thermostatted Column Compartment) G1316A, and ALS G1329A autosampler (Agilent Technologies). The column was a Zorbax eclipse XDB C-18 (4.6 × 100 mm, 3.5 µm). A gradient analysis was performed using methanol and 0.1% formic acid, which started from 20% methanol for 7 min, followed by 28% methanol for 4 min and 75% methanol for 2 min. The flow rate was 1 mL/min and detection was 245 nm. Analyses of caffein were performed with the same instrument as the antipyrine analysis. The analysis was performed using a mixture of 85% water and 15% acetonitrile.

## LC-MS/MS

Glyphosate was analyzed using an Agilent 1200 device (Agilent Technologies International Japan) coupled to an AB Sciex TripleTOF 5600+ (AB Sciex, Framingham). Chromatographic separation was performed with hydrophilic interaction liquid chromatography (HILIC) column (InertSustain Amide, 100 mm × 3.0 mm i. d., 3 µm; GL Sciences Inc.). The mobile phase was 0.1% formic acid–acetonitrile (80:20, v/v) at 0.2 mL/min and the injection volume was 2 µL. The column was kept at 40 °C in the column oven. An AB Sciex 5600+ triple quadrupole mass spectrometer was used for mass spectrometry determination. The software AB Sciex Analyst was used for data collection. Ionization was performed in negative mode. Mass spectrometry data were collected in the multiple reaction monitoring (MRM) mode. The MRM parameters for the target analytes are shown in Supplementary Data 2. Instrument settings for ESI analysis were as follows: Heater temperature, 350 °C; curtain gas, 25 psi; nebulizer gas, 50 psi; heater gas, 50 psi; ionspray voltage, −4500 V; dwell time, 250 ms.

## Statistical analysis

Statistical analysis was performed using StatView version 5.0.1 (SAS Institute Inc.). A value of $P < 0.05$ was considered statistically significant. All tests were two-sided.

## Reporting summary

Further information on research design is available in the Nature Portfolio Reporting Summary linked to this article.

## Data availability

The data that support the findings of this study are available from the corresponding author on reasonable request. RNA sequencing data reported in this paper were obtained from our previous study[21] (Japanese Genotype-phenotype Archive (JGA) under the accession number: JGA000117 and JGA000122). Data of Figs. 3E, K, L, 4G, I, J, 5F, G, Supplementary Figs. S4, S12A, B, C, S16E, F, G, H, and I are provided as a Source Data file. Source data are provided with this paper.

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

## Acknowledgements

The authors thank Dr. Shinji Takahashi (Technical Division, School of Engineering, Tohoku University) for providing HPLC and LC-MS/MS data, Mr. Junya Takagi (Tohoku University) for helping to design microwell plate, and Dr. Yuji Nashimoto (TMDU) for valuable comments. The authors thank Dr. Yuriko Sakamaki and Dr. Ayako Mimata (the TMDU Research Core) for their technical support with confocal microscopes, SEM, and TEM. This study was supported in part by JSPS KAKENHI (Grant Numbers 23H01821, 22K18936 and 21K04852) (to T.H. and H.K.), the Mandom International Research Grants on Alternatives to Animal Experiments (to T.H.), AMED-CREST (Grant Number JP21gm1310001) (to T.A. and H.K.), the JST Adaptable and Seamless Technology Transfer Program through Target-driven R&D (Grant Numbers JPMJTM22BD) (to H.K.), the Research Center for Biomedical Engineering (to T.H. and H.K.), and TMDU priority research areas grant (to T.H.).

## Author contributions

T.H., H.O., S.S., N.K., E.K., A.O., T.A., and H.K. conceived the ideas. T.H., and H.O. conducted experiments. T.H., H.O., and A.S. analyzed data. T.H., H.O., S.S., N.K., E.K., A.O., A.S., T.A., and H.K. wrote and revised the manuscript. All authors have given approval to the final version of the manuscript.

## Competing interests

TMDU has filed a patent application covering the protocol and methods for the generation of human trophoblast organoids. T.H. and H.K. own stock as members of a recently established company, HPS Inc., and they potentially would receive compensation from the company. The remaining authors declare no competing interests.
