## [Peer Review File · Nature Communications]

Trophoblast stem cell-based organoid models of the human placental barrierReviewers' comments:

Reviewer #1 (Remarks to the Author):

In this manuscript the authors present two key findings -

1) they develop methods to generate trophoblast stem cell spheroids that are able to form syncytiotrophoblast on the outside. This is an important advance, as current 3D organoid models have an inverted morphology, meaning they are unsuitable to assess trophoblast turnover or syncytiotrophoblast uptake.

2) they develop a column system to study transport of compounds across the placental barrier in which syncytiotrophoblast form on top of a trophoblast stem cell layer. This model is used to assess antibody transport. They then improve the model by incorporating an endothelial cell layer underneath, and investigate the transfer of caffeine, antipyrine, and glyphosphate across this barrier. Development of such barrier models is important across a range of areas of placental biology, to understand nutrient transport, syncytiotrophoblast function, drug transport, and environmental exposure. Prior similar organ on a chip models have been relatively basic in comparison, employing trophoblast cell lines that have been syncytialised, and not recapitulating all the cell lines in the barrier (syncytiotrophoblast, TSC/cytotrophoblast and endothelial cells), and this model improves on those.

The work in the first half of the manuscript is highly novel, and following their prior publication in which they generated TSCs in part by advancing the media used, they have carefully manipulated culture conditions to refine syncytiotrophoblast differentiation. This is key as previously syncytiotrophoblast differentiation in organoids happened spontaneously by default, limiting how we could deliberately study this process and the factors that regulate it. However, the second half of the manuscript where the column placental barrier model is established is less cohesive, and coverage of this in the discussion is lacking.

- Introduction, pg 3, line 9 - 'chemicals in both mother and fetus fractions' - this would be better worded as 'chemicals in both the maternal and fetal compartments' (or circulations perhaps).

- Introduction, pg 3, line 17/18 - this sentence is slightly misleading in stating that the surface of the villus has two kinds of trophoblast lineage cells. It would be more accurate to say the villus is encased in a bilayer of trophoblast, with an inner cytotrophoblast layer, and an outer syncytiotrophoblast layer that is in direct contact with maternal blood'

- Introduction, pg 3 line 34 - some ST barrier models have been established (cited by the authors, eg Blundell et al) - I suspect what the authors mean here is that these are not derived from primary cells, which is a fair point. Suggest adding the word 'primary' into this sentence to make this more accurate.

- Introduction, pg 4, line 9-11 - please provide references for the statements about prior work attempting to use TS cells to form barrier models (even if a conference abstract?). It is also not clear how focussing on the process of ST turnover has addressed this issue?

- First results paragraph / Figure 1 - here the authors use both E-cadherin and F-actin. F-actin nicely shows the cell boundaries and multinucleated regions of syncytiotrophoblast are evident around the spheroid edge, so this staining is convincing. What was surprising to me is that E-cadherin staining is usually used for a similar purpose in syncytiotrophoblast differentiation cultures (to delineate cell membranes and count multinucleated regions), yet the staining shown here is spread throughout the cytoplasm in panels C and D. Apart from this syncytiotrophoblast marker expression, and morphology in panel G, is convincing (assuming negative controls have been used appropriately, see below).

- Methods, page 17 (relating to the above results) - please state the negative controls used of immunofluorescent staining. Trophoblast are notoriously autofluorescent (the syncytiotrophoblast even

more so), and the use of an irrelevant IgG control antibody is essential.

- Results, Pg 5, line 35 - 'collagen membrane to ensure'

- Results, Pg 6 , line 3 - it is not immediately obvious that the protocol described in lines 1-3 is treatment protocol a) (referred to in line 3). Adding this to the previous sentence would assist the reader.

- Figure 2 / related results - while not explicitly stated in the corresponding results section that talks of 'a single layer of ST cells on the differentiated cell layer', Figure 2 shows that only 41% to 67% of the surface is covered by syncytiotrophoblast. As the syncytiotrophoblast is thought to play a role as a primary barrier in many scenarios (for example sequestering viruses such as Covid-19, or microparticles from the circulation), I have significant concerns that this level of coverage does not represent a true syncytiotrophoblast barrier that is comparable to that observed in vivo. Indeed, (for example) it is thought that small breaches in the syncytiotrophoblast may underlie the low frequency of Covid-19 transmission to the fetus, thus it is unclear how effective a barrier that only covered 50% of the surface would be.

- Results, page 6, line 27 - the authors should make it clear here that the results showing human ST cells express very little FcRn are from TSC-derived ST - not directly isolated primary syncytiotrophoblast, or analysis of syncytiotrophoblast in sections. This could be an inconsistency between in vivo and in vitro scenarios, and should be discussed.

- Results, page 6 - Was there a correlation between the extent of IgG/IgA crossing the barrier, and the % of ST coverage? This comment would also apply to the transport of antipyrine, caffeine and glycerophosphate in the later results section.

- Results, page 6, line 36 - the authors hypothesise here that selective transport of IgG to the fetus may be regulated by the fetal endothelium rather than the syncytiotrophoblast - whilst a valid hypothesis, it is strange that they go on to improve their model by including an endothelial layer, and then do not use this to test this hypothesis but rather assess transport of other factors? Antibody transport across the ST/TSC/HUVEC model should also be assessed to complete this work, particularly if seeking publication in a journal such as Nature Comms.

- Results, page 7, line 13-17 - to what degree are higher TEER values impacted by the thickness of the combined cell layers (ie the ST/TSC/HUVEC model has a higher TEER than the ST/TSC model only) compared to the 'tightness' of adhesion between the cells? ie what does this really tell us about the integrity of the barrier to small compounds and which cell type is most responsible for forming an effective barrier?

- Discussion - The discussion is very short and talks primarily of the generation of ST layers and the different media conditions required to achieve this. Discussion of transport across this barrier model (antibodies, compounds) is missing, as is any reference to their model incorporating HUVECs into this barrier. This is critical to enable assessment of the reliability of the model they have developed - for example, how do Papp values of caffeine, antipyrine and glycerophosphate compare to levels seen in placental perfusion models in which these are more commonly assessed? What do their findings around transport tell us about placental physiology? What are the limitations of the models developed here?

- Methods, page 19 - please state the type of statistical test used in each scenario where this was undertaken. Were data tested for normal distribution?

Reviewer #2 (Remarks to the Author):

The authors have successfully developed two types of trophoblast stem (TS) cell-based models of the human placental barriers (formed by syncytiotrophoblasts), such as organoid and column-type 2D models, which can be used for evaluating the transplacental drug transfer to the fetuses. Although the successful differentiation into syncytiotrophoblasts from TS cells on cell culture dishes has already been reported, syncytiotrophoblast-like cells cultured on dishes are not suitable for evaluating the transcellular transfer of drugs. On the other hand, the column-type 2D model developed in this study using TS cells has the potential for in vitro estimating transplacental transfer of drugs. Compared with existing models of the placental barriers using choriocarcinoma cells, primary cultured cells, or pluripotent stem cells, the model developed in this study can form a substantial barrier with a minimal paracellular transfer of substances between cells. Of course, some issues are still to be solved, including that the expression and function of membrane transporters involved in transcellular drug transfer are unclear. Nonetheless, the reviewer believes that this model will undoubtedly trigger establishing a gold standard model of the placental barrier. Following comments are raised to deepen understanding of the model:

- 1) Page 5, line 12: Please explain why authors consider that differentiation to ST cell was insufficient. The reviewer failed to clearly recognize the differences between organoids in Fig. S2A, S2B, and S2C.
- 2) Page 5, line 15: It is hard to identify the smoothness of the organoids from Fig. S2. Further explanations are needed.
- 3) Page 5, line 18: Explanations for marker proteins such as SDC1, hCG, and E-cadherin should be put before explaining Fig. S2A and S2B.
- 4) Page 5, line 36: Please explain why the device can facilitate the removal of dead cells.
- 5) Page 7, line 3: Is it sure that fetal umbilical endothelial cells form tight junctions? If so, please cite the article that shows the presence of tight junctions in umbilical vessels.
- 6) Page 7, line 27: It is necessary to discuss further why the Papp values of caffeine in TS-derived models are lower than that in BeWo cells. Is it due to the lack of caffeine transporter in TS-derived models?
- 7) Page 7, line 30: Please clearly indicate that EGM-2 contains FBS. And please explain why FBS affects Papp values even though FBS is also included in S-DM.
- 8) Fig. 1, legend: "Staining with phalloidin-FITC and Hoechst" is the explanation for (F), not (D).
- 9) Fig. 4E: The reviewer would like to know why the TEER values in ST/HUVEC co-culture were significantly higher than those in ST. It looks inconsistent with the Papp values of antipyrine and caffeine.
- 10) Fig. S4: Further explanations regarding the cell types (x-axis) and gene names (FCGRT is the same with FcRn?) are needed.

Reviewer #3 (Remarks to the Author):

In this study the authors sought to establish a model, using human trophoblast stem cells (TS), that could mimic the placental barrier in order to study how drugs cross the placenta. First, they develop a protocol for the formation of trophoblast organoids in which the syncytiotrophoblast (ST) cover the outer layer with cytotrophoblast (CT) on the inside. This protocol is then used to form layers of TS and ST on a modified column-type device to study their permeability to different substances. These two layers form a barrier (as shown by TEER measurement) that show characteristics similar to the placental barrier. The authors characterize the permeability of the barrier both to proteins (such as hCG, IgG and IgA) and to diffusible chemical compounds. The model is further improved by the addition of an endothelial layer (HUVEC cells).

A model that recapitulates placental villous structure and cell composition would represent an important tool to evaluate the transfer and toxicity of compounds from the maternal blood to the fetus through the placenta. However, there are several issues with this study. The reproducibility and

technical aspects of the proposed model are not clear and important controls are missing from some experiments, making the results difficult to interpret and to evaluate whether this really could model the placental villus. The nature of the cells present and their organization within their model system also is not clear. Further work is needed on the culture conditions and experimental setup, to provide a more reliable model for placental barrier as well as a more detailed characterization of their system. Please see below for further details.

Major comments:

(1) How many TS lines were used for this work (also per experiment)? This was not specified. Given that TS cells are derived from primary samples, to account for inter-individual variability, several independent, biological replicates are necessary for all experiments.

(2) What is the efficiency of formation of organoids with a completely intact and single layer of ST (Fig. 1)? Further evidence of true multinucleated ST is needed besides Fig 1G. Please show an EM of this layer.

(3) The authors show two new methods for the culture and differentiation of TS cells: generation of apical-out organoids in the microwell plate (Fig. 1) and the formation ST barrier on a column insert (Fig. 2). As this is the first time that TS cells are seeded in these conditions, it is important to show a detailed characterization of the TS cells in these new undifferentiated conditions, verifying the maintenance of stem characteristics. Please show that they still express trophoblast stem cell and proliferation markers of CT for eg. TP63, GATA3, Ki67.

(4) A report has shown that the TS cells actually represent trophoblast cells from the cell column (CC) rather than CT (PMID: 32619492). Please stain for NOTCH1 and ITGA2 which are markers of CC (PMID: 27849611, PMID: 29540503).

(5) The outside-in organoids are not truly physiologically representing the villous structure as VT are a single epithelial layer in vivo (as shown in their Fig. 1A) and not a clump of cells (Fig 1A-F). Please rephrase throughout the manuscript.

(6) The seeding of the TS cells in the microwells is a key step for the formation of the apical-out trophoblast organoids, thus it should be mentioned in the main text (with reference to Fig. S1)

(7) From the images in Fig. 2, it is clear that the ST layer formed (even under the best conditions) is not a continuous layer and therefore it is not a proper barrier. This is also pointed out by the authors (p9 L33-35). This is a major limitation and concern as the formation of an intact barrier for a placental barrier model, is an essential requirement. Furthermore, as different lines will have different differentiation efficiency, the amount of ST formed will vary, increasing the variability between already an suboptimal system.

(8) In Fig. 3C the authors measure hCG in the well (apical side of the barrier) and in the column (basal side). Some critical controls are missing in this experiment. First, the TS-only control is only seeded for 1 day, while it would be more appropriate to keep the cells for the same number of days as the ST cells, without differentiation medium (the same applies to Fig. 2E). Second, the authors claim that the absence of hCG in the column demonstrates the formation of a barrier in the STM compared to TSM. However, in the TSM control, hCG is not present, thus it is not possible to see its diffusion in the column. To overcome this issue, hCG should be added in the wells of a TSM control seeded for at least 4 days (in which a barrier effect is visible in the STM) and measure its diffusion into the column.

(9) The experiment with the immunoglobulins is not really helpful in showing the validity of the model. All they show in Fig. 3D-G is that both IgG and IgA are uptaken by the ST. They claim in the text that "the transition rate of IgG and IgA in comparison to controls was similar". From the graph it is not

possible to make this comparison, as the Y-axis scale is different between IgG and IgA, and the transition rate is very low. I would suggest using the % over the control in order to make the point. It is also not clear what the control "Cont" stands for. Furthermore, in Fig. 3F, it is not clear why the transition rate is so low. While the authors claim that "This finding raises the possibility that not only IgG, but also IgA can cross the ST barrier", this is not supported by the data, which rather show that the Igs are retained in the ST cells and not released on the opposite side. Furthermore, as an appropriate control, the authors should compare the transition rate if IgG-FITC and IgA-FITC with a protein that is known to not being able to cross the barrier (e.g., hCG, that they showed to be retained in the wells).

(10) The authors reference through published data that FcRn, which is considered the mediator of transplacental passage of IgG, is expressed at low levels in the ST. The expression of this receptor should be checked also in the seeding and differentiating conditions that the authors use for the experiments in Fig. 3, to confirm the low expression in their model. This could be achieved by qPCR or by immunofluorescence, using cells that show the expression of the receptor as a positive control.

(11) The integrity of the HUVEC layer (Fig. 4B-D) needs to be evaluated using confocal microscopy of the cell-cell junctions, as from the images shown in the figure it is not clear whether they form a continuous layer.

(12) Fig. 4F-H: the graphs show a difference in Antipyrine and Caffeine Papp between ST and ST/HU conditions, but the graphs lack an indication of the significance of this difference. If the difference was significant, it should be experimentally addressed. The authors postulate that the difference could be due to FBS affecting the thickness of the undifferentiated cell layers in the ST/HU condition. This hypothesis could be easily proved by adding a condition with ST layer, and HUVEC medium in the column (without the HUVEC cell layer). Furthermore, given the alteration in the media composition to which the underlying TS cells are in contact, the authors should characterize the features of the TS cells in terms of stemness and integrity of the layer.

(13) Fig. 4I: The Glyphosate Papp is already very low in the control, as pointed out by the authors, possibly due to an effect of the collagen membrane. Is there any evidence that these values are similar to the chorionic connective tissue barrier? Without this information, it is not possible to understand if there could be a bias in the results of chemical screens using the proposed model. A comparison of the permeability to Glyphosate of the connective tissue and the collagen membrane would solve this issue.

(14) Fig. S5: it is not very clear what the fluorescent images should show. The images show very different background. To be comparable, the images should be acquired with the same setups and subjected to the same brightness/contrast and background removal. In this case, to evaluate the morphology of the cells, a close-up would be required. Staining of the nuclei and the cell junctions are needed to show the integrity of the layer.

Minor comments:

(1) Details are missing in the methods:

- o The derivation of the TS cells and of human placental villi are missing.
- o Some details are missing regarding the imaging setup (e.g., objectives and cameras used for the acquisition)
- o In the immunofluorescence method section, there is no mention of the organoids in the microwell plates. Please specify in the methods if the frozen sections were used for this purpose.
- o In the immunofluorescence method section, please specify how many times the stainings were performed in independent experiments.
- o The H&E staining is not mentioned in the methods
- o Method "fabrication of a column-type device for barrier models": it is not clear the reason for the

addition of a polyimide film and a silicone ring.

(2) Details are missing in the figure legends:

- o When an immunofluorescence image is shown, please always specify the acquisition method (e.g., confocal or widefield)
- o The legends should specify at which differentiation stage the cells were fixed and stained
- o The legends should specify if N refers to independent experiments or technical replicates
- o Fig. 1: the F is missing (D should be an F)
- o Fig. S4: please include a short summary about the generation and analysis of the data

(3) Fig. 1E: is it a 3D reconstruction of a series of confocal sections (same applies to S2B)? Please specify how the samples were prepared and how these images were acquired. These images are not informative, a section of the costaining hCG-SDC1 would show better their colocalization in the ST layer.

(4) Fig. S4: In the graph there isn't any standard deviation, and it is not clear how many times the experiment was repeated.

(5) The considerations about the medium that are present in the discussion should be moved in the main text (Fig. S6 and S7)

Trophoblast stem cell-based organoid models of the human placental barrier

Takeshi Hori, Hiroaki Okae, Shun Shibata, Norio Kobayashi, Eri H Kobayashi,
Akira Oike, Takahiro Arima, and Hirokazu Kaji

Point-by-point response to the reviewers' comments

Reviewer #1:

General comments:

In this manuscript the authors present two key findings –

1) they develop methods to generate trophoblast stem cell spheroids that are able to form syncytiotrophoblast on the outside. This is an important advance, as current 3D organoid models have an inverted morphology, meaning they are unsuitable to assess trophoblast turnover or syncytiotrophoblast uptake.

2) they develop a column system to study transport of compounds across the placental barrier in which syncytiotrophoblast form on top of a trophoblast stem cell layer. This model is used to assess antibody transport. They then improve the model by incorporating an endothelial cell layer underneath, and investigate the transfer of caffeine, antipyrine, and glyphosphate across this barrier. Development of such barrier models is important across a range of areas of placental biology, to understand nutrient transport, syncytiotrophoblast function, drug transport, and environmental exposure. Prior similar organ on a chip models have been relatively basic in comparison, employing trophoblast cell lines that have been syncytialised, and not recapitulating all the cell lines in the barrier (syncytiotrophoblast, TSC/cytotrophoblast and endothelial cells), and this model improves on those.

The work in the first half of the manuscript is highly novel, and following their prior publication in which they generated TSCs in part by advancing the media used, they have carefully manipulated culture conditions to refine syncytiotrophoblast differentiation. This is key as previously syncytiotrophoblast differentiation in organoids happened spontaneously by default, limiting how we could deliberately study this process and the factors that regulate it. However, the second half of the manuscript where the column placental barrier model is established is less cohesive, and coverage of this in the discussion is lacking.

Response: We greatly appreciate the reviewer's careful reading of our manuscript and the positive comments. We revised our manuscript based on

the reviewer's suggestion.

Comment #1: - Introduction, pg 3, line 9 - 'chemicals in both mother and fetus fractions' - this would be better worded as 'chemicals in both the maternal and fetal compartments' (or circulations perhaps).

Response: We thank the reviewer for the suggestion. We revised the sentence in red (page 3, lines 8-9).

Comment #2: - Introduction, pg 3, line 17/18 - this sentence is slightly misleading in stating that the surface of the villus has two kinds of trophoblast lineage cells. It would be more accurate to say the villus is encased in a bilayer of trophoblast, with an inner cytotrophoblast layer, and an outer syncytiotrophoblast layer that is in direct contact with maternal blood'

Response: We thank the reviewer for the suggestion. We revised the sentence (page 3, lines 17-19).

Comment #3: - Introduction, pg 3 line 34 - some ST barrier models have been established (cited by the authors, eg Blundell et al) - I suspect what the authors mean here is that these are not derived from primary cells, which is a fair point. Suggest adding the word 'primary' into this sentence to make this more accurate.

Response: As suggested, we added the word 'primary' in the sentence (page 3, line 34).

Comment #4: - Introduction, pg 4, line 9-11 - please provide references for the statements about prior work attempting to use TS cells to form barrier models (even if a conference abstract?). It is also not clear how focussing on the process of ST turnover has addressed this issue?

Response: As suggested, we provided a reference for the statements about the work attempting to use TS cells to form barrier models (page 4, line 10).

The previous method with forskolin to differentiate TS cells to ST cells cannot provide a placental barrier model because the ST cells do not proliferate,

fuse and shrink slightly in this differentiation condition. We added a result of an additional experiment shown in Fig. S1. To address this issue, our group has focused on the process of ST turnover in the villus. During turnover, the ST cell layer forms small knots and releases them into the maternal blood, and progenitor CT cells underlying the ST cell layer fuse to replenish ST cells. Inspired by this phenomenon, we aimed to generate trophoblast organoid barrier models in which an ST cell layer is placed over undifferentiated TS cells. In other words, we hypothesized that an ST cell layer constantly supplied with cells from the underlying undifferentiated cell layer can serve as a placental barrier model that maintains a stable barrier function without being affected much by ST turnover and morphological changes. We added the abovementioned explanations in the section of the Introduction (page 4, lines 11-13 and 17-20).

Fig. S1

Fig. S1 Differentiation of ST cells on a culture plate.

A 6-well plate was coated with 2.5 $\mu\text{g}/\text{mL}$ collagen IV for 30 min and washed with PBS(-). TS cells were then seeded at 0.5×10^5 cells/well to the wells and cultured with TS medium without iMatrix511. After 2 days, culture media were changed to ST medium with 2 μM forskolin. On Day 4, images of cells were taken. Scale bars indicate 200 μm .

Comment #5: - First results paragraph / Figure 1 - here the authors use both E-cadherin and F-actin. F-actin nicely shows the cell boundaries and multinucleated regions of syncytiotrophoblast are evident around the spheroid edge, so this staining is convincing. What was surprising to me is that E-cadherin staining is usually used for a similar purpose in syncytiotrophoblast differentiation cultures (to delineate cell membranes and count multinucleated

regions), yet the staining shown here is spread throughout the cytoplasm in panels C and D. Apart from this syncytiotrophoblast marker expression, and morphology in panel G, is convincing (assuming negative controls have been used appropriately, see below).

Response: As the reviewer pointed out, E-cadherin proteins localize in the cell membrane. When it is stained with an antibody, E-cadherin is shown clearly in the cell-cell interface in the case of 2D cultures in TS cells but not ST cells. On the other hand, in the case of a three-dimensional cell population, the cell-cell adhesion part exists not only on the sides of TS cells but also on the top and bottom of the cells. Therefore cell-cell surface can be unclear in cross-sections of 3D-cultured cells. Some reports are showing similar results of E-cadherin staining in 3D-cultured cells (Jang et al., Sci Rep. 2017 Jan 27;7:41541). Moreover, in our results, the ST cell layer was not stained significantly with two kinds of E-cadherin antibodies (Fig. 1C and 1D). Therefore, we are sure that our experiments successfully conducted E-cadherin staining. As the reviewer suggested, we conducted staining experiments with negative controls (Fig. S4A).

Fig S4A

Fig. S4A. Immunostaining analysis of trophoblast organoids.

(A) Apical-out spherical trophoblast organoids were generated from a TS cell line CT27 by the same procedure as described in Figure 1A. Immunostaining analysis with antibodies for syndecan 1 (SDC1), E-cadherin (E-cad), GATA3, normal mouse IgG (negative control), or normal rabbit IgG (negative control) was conducted for cross-sections of the organoids. The scale bar indicates 200 μm .

Comment #6: - Methods, page 17 (relating to the above results) - please state the negative controls used of immunofluorescent staining. Trophoblast are notoriously autofluorescent (the syncytiotrophoblast even more so), and the use of an irrelevant IgG control antibody is essential.

Response: As the reviewer pointed out, we conducted additional experiments for the negative controls (Fig. S4A shown above).

To further demonstrate the ST formation and its fusion, we generated two kinds of genetically-modified TS cells, i.e., GFP-expressing cells and KSB (kusabira-orange)-expressing TS cells. These TS cells show green or orange fluorescence, respectively. Therefore if the two kinds of TS cells were fused, they should be seen with yellow color under the fluorescence microscopy (Fig. 2A). As we expected, we were able to see the ST cell layer showing yellow color, demonstrating that GFP-TS cells and KSB TS cells were differentiated into ST cells and fused to make a ST cell layer. This is a direct evidence of ST fusion (Fig. 2B and C). The methods to generate these two cells are described in the Material and Methods (page 21, lines 4-37 and page 22, lines 1-4), and we explained these points in the revised manuscript (page 7, lines 13-22).

Fig. 2

Fig. 2. Analysis of the fusion of ST cells in trophoblast organoids.

(A) The procedure for generation of the trophoblast organoids using a mixture of GFP-TS cells and Kusabira-orange (KSB)-TS cells.

(B) An image of a trophoblast organoid in a microwell.

(C) Fluorescence images of a cross-section of the organoids generated from a mixture of GFP-TS cells and KSB-TS cells.

Comment #7: - Results, Pg 5, line 35 - 'collagen membrane to ensure'

Response: We have amended the phrase as suggested (page 7, line 32).

Comment #8: - Results, Pg 6 , line 3 - it is not immediately obvious that the protocol described in lines 1-3 is treatment protocol a) (referred to in line 3). Adding this to the previous sentence would assist the reader.

Response: We are sorry for the confusion. We revised the manuscript according to the reviser's advice (page 8, line 4).

Comment #9: - Figure 2 / related results - while not explicitly stated in the corresponding results section that talks of 'a single layer of ST cells on the undifferentiated cell layer', Figure 2 shows that only 41% to 67% of the surface is

covered by syncytiotrophoblast. As the syncytiotrophoblast is thought to play a role as a primary barrier in many scenarios (for example sequestering viruses such as Covid-19, or microparticles from the circulation), I have significant concerns that this level of coverage does not represent a true syncytiotrophoblast barrier that is comparable to that observed in vivo. Indeed, (for example) it is thought that small breaches in the syncytiotrophoblast may underlie the low frequency of Covid-19 transmission to the fetus, thus it is unclear how effective a barrier that only covered 50% of the surface would be.

Response: We thank the reviewer for the constructive suggestions. We agree with the reviewer's comment and have tried to increase the level of coverage by syncytiotrophoblast on the undifferentiated cell layer.

To find the condition to increase the level of coverage by ST efficiently and further demonstrate the ST differentiation, we generated a TS cell line that will have SDC1-GFP (green fluorescence protein) when the SDC1 promoter is activated. We applied split fluorescent protein systems since it was difficult to transfect a much larger DNA construct into TS cells (Fig. S5). Following consideration of some culture conditions using the SDC1-GFP TS cells, we finally succeeded in increasing the levels of coverage up to about 100% (Fig. 4A-E). The main point we changed the following:

1. We added forskolin, an activator of adenylyl cyclase, to the 3rd strong-differentiation medium (S-DM) at 2 μM to facilitate the differentiation from TS cells to ST cells. This was used as S-DM2.
2. The culture schedule was shortened to 6 days culture from 7 days culture because forskolin is a very strong inducer into ST cells and too much differentiation leads to cell death.
3. We increased the amount of medium from 800 μL to 1300 μL to equalize the water level in the inner column and outer well. This is because a slight medium flow would occur from the inner column to the outer well that may affect cell differentiation if the water level in the inner column is higher than that in the outer well as in the previous experiments.
4. We added fibronectin to the Matrigel solution that was used for coating the collagen membranes to increase the adhesion of TS cells to the collagen membranes.

We revised the manuscript in the sections of Materials and Methods (page 23, lines 1-11) and Results (page 9, lines 16-34) in the revised manuscript.

Fig. S5

Fig. S5. Split fluorescent protein systems for SDC1-GFP proteins
 (A) Insertion of a DNA fragment for 7xGFP11 into the last exon of the *SDC1* gene.
 (B) A schematic illustration of the split fluorescent protein systems with 7xGFP11 and GFP1-10.

Fig. 4

Fig. 4. Enhancement of ST coverage in the barrier models

(A) Schematic of additional factors for ST barrier formation.

(B) A culture schedule for the improved ST barrier model. Triangular marks indicate medium exchange.

(C) SDC1 expression levels visualized using SDC1-GFP TS cells in the improved culture condition.

(D) Immunostaining for SDC1 and fluorescence of SDC1-GFP in SDC1-GFP TS cells.

(E) Immunostaining for SDC1 and fluorescence of Hoechst in TS cells (CT27 line).

The scale bars indicate 200 μm (C-E).

(C-E) Images were taken using BZ-X800/810. (D and E) Images from the barrier models of day 6.

Comment # 10: - Results, page 6, line 27 - the authors should make it clear here that the results showing human ST cells express very little FcRn are from TSC-derived ST - not directly isolated primary syncytiotrophoblast, or analysis of syncytiotrophoblast in sections. This could be an inconsistency between in vivo and in vitro scenarios, and should be discussed.

Response: In Fig. S8, we showed that both directly isolated primary syncytiotrophoblast and TSC-derived ST express very little FcRn. Therefore, there is a consistency between in vivo and in vitro in our samples. Data analysis of the Human Protein Atlas public repository (proteomics.org) supported our findings that human ST cells express very little FcRn. This is quite interesting because syncytiotrophoblast has been thought to express functionally FcRn. To describe more clearly, we have amended the sentences in the main text (page 8, lines 3-5) and the legend of Fig. S8.

Comment # 11: - Results, page 6 - Was there a correlation between the extent of IgG/IgA crossing the barrier, and the % of ST coverage? This comment would also apply to the transport of antipyrine, caffeine and glyophosphate in the later results section.

Response: We thank the reviewer for the constructive suggestions. IgG/IgA transport was slower than the transport of chemicals such as caffeine in the ST

barrier models. Therefore, in the experiment of Fig S8 B and S8C, we incubated cells with IgG/IgA for a little long time, i.e. 3 hours, to collect IgG/IgA that crossed through the ST cells as much as we can. To obtain reliable data that shows a correlation between the extent of IgG/IgA crossing the barrier and the % of ST coverage, we may need to find another condition to culture ST cells for a longer time in the assay buffer or other methods to analyze IgG/IgA more sensitively. Thus, we could not perform further experiments to determine the correlation.

On the other hand, chemical permeability using antipyrine caffeine, or glyphosate was conducted for a revised ST barrier model. There was no large difference between the previous ST models and the revised ST modes in the translocation of antipyrine. The Papp values of caffeine and glyphosate were increased in the revised models compared with the previous models. Since the number of TS cells used was lesser (ca 70% of the previous cell number) and culture time was shorter (6 days) in the revised protocol, the revised ST/TS layer should be thinner than the previous one. However, we could not accurately determine the thickness of the thinner barrier models because of their vulnerability in making section analyses. We think that the difference in the Papp values between the revised barrier models and the previous models was probably due to the thickness of the barriers rather than ST coverage.

Unfortunately, based on the results of additional experiments, we concluded that our previous experiments with mass spectrometry had technical problems with sensitivity for the analysis and that we did not provide accurate values of concentrations of glyphosate in the previous samples. The technical issues include the possibility that the detector or vacuum system was defective or the condition of the analytical column was at low performance. After replacing the detector, vacuum system, and column with new ones, we performed mass spectrometry again (Fig. S11I). We found that Papp of glyphosate in the control collagen membrane is almost similar to Papp of antipyrine and caffeine.

Comment # 12: - Results, page 6, line 36 - the authors hypothesise here that selective transport of IgG to the fetus may be regulated by the fetal endothelium rather than the syncytiotrophoblast - whilst a valid hypothesis, it is strange that they go on to improve their model by including an endothelial layer, and then do not use this to test this hypothesis but rather assess transport of other factors? Antibody transport across the ST/TSC/HUVEC model should also be assessed

to complete this work, particularly if seeking publication in a journal such as Nature Comss.

Response: We thank and agree with the valuable comment. The present study is the first study to generate apical-out spherical trophoblast organoids and barrier models highly covered with fused-ST cells. So we would like to focus on the demonstration of the above-mentioned organoids and barrier models in this manuscript. We removed Fig. 3D and F in the previous manuscript and moved Fig. 3E and G into the supplemental figure S8B and C, respectively. To completely demonstrate the IgG/IgA transport to the fetus compartment, in addition to the reviewer's mentioned experiment, the following experiments should be done. 1) analysis of the uptake and excretion rates of each IgG/IgA to ST cells. 2) Accurate quantification of IgG/IgA that is translocated to the fetal compartment using mass spectrometry of IgG/IgA without labeling any fluorescent chemicals such as FITC. 3) Determination of localization of IgG/IgA by immunostaining for human placental section samples using anti-IgG/anti-IgA antibodies. 4) Mass-spectrometry to reveal receptors for IgG/IgA in syncytiotrophoblast in vivo. We would like to perform these experiments in future projects.

Comment # 13: - Results, page 7, line 13-17 - to what degree are higher TEER values impacted by the thickness of the combined cell layers (ie the ST/TSC/HUVEC model has a higher TEER than the ST/TSC model only) compared to the 'tightness' of adhesion between the cells? ie what does this really tell us about the integrity of the barrier to small compounds and which cell type is most responsible for forming an effective barrier?

Response: The TEER value varies depending on factors such as the pathways in which electricity flows, i.g., intracellular, intercellular, and gaps between cells and substrates. It is thought that the thicker the cell layer in total, the higher the TEER value, and the more the ST is fused, the higher the TEER value. Since the placental villi in vivo are extremely small, it is difficult to measure TEER values using the actual villi. The purpose of TEER measurements in this study is not to compare in vivo villi and in vitro models but mainly to confirm whether cell sheets with resistance have actually been produced. This measurement is important because if there are surface regions in the collagen membrane

without being covered with cells, TEER values should be shown as they are very low. Small compounds cross the placental barrier depending on the nature of the chemicals. Some chemicals cross the barrier through, for example, passive diffusion, facilitated diffusion, or active transport, and this translocation manner is different from how electricity crosses the cell layers.

Our study demonstrated that the co-culture of ST/TSC and HUVEC show higher TEER values compared with the ST/TSC only. In the revised models (ST/TSC/HUVEC v.s. ST/TSC only), TEER increased to 2.38-fold on day 6 (Fig. 5F). In the previous models (ST/TSC/HUVEC v.s. ST/TSC only), TEER increased to 2.29-fold on day 6, and 1.40-fold on day 7 (Fig. S11E). Although the mechanism of the synergistic effects is not clear, culture media or factors produced by HUVEC probably affected the tightness of the ST/TSC barrier. Moreover, some factors in the ST/TSC culture media may affect the integrity of HUVEC layers shown in Fig. S10. We have amended the manuscript by explaining these things (page 10, lines 17-19).

Comment # 14: - Discussion - The discussion is very short and talks primarily of the generation of ST layers and the different media conditions required to achieve this. Discussion of transport across this barrier model (antibodies, compounds) is missing, as is any reference to their model incorporating HUVECs into this barrier. This is critical to enable assessment of the reliability of the model they have developed - for example, how do Papp values of caffeine, antipyrine and glyphosphate compare to levels seen in placental perfusion models in which these are more commonly assessed? What do their findings around transport tell us about placental physiology? What are the limitations of the models developed here?

Response: As the reviewer suggested, we have amended the manuscript by adding information about the barrier model compared to ex vivo perfusion systems (page 13, lines 3-37 and page 14, lines 1-8). We also discussed spherical trophoblast organoids and their applications in the revised manuscript (page 12, lines 18-37 and page 13, lines 1-2).

Comment # 15: - Methods, page 19 - please state the type of statistical test used in each scenario where this was undertaken. Were data tested for normal distribution?

Response: We thank the reviewer's advice. As the reviewer suggested, we added detailed statements about statistical tests in each figure legend. Statistical analysis was tested under the assumption that the data are normally distributed because our data were obtained from experiments with cell lines but not animals or patients, and this assumption is generally accepted in the field of cell biology.

Reviewer #2 (Remarks to the Author):

General comments:

The authors have successfully developed two types of trophoblast stem (TS) cell-based models of the human placental barriers (formed by syncytiotrophoblasts), such as organoid and column-type 2D models, which can be used for evaluating the transplacental drug transfer to the fetuses. Although the successful differentiation into syncytiotrophoblasts from TS cells on cell culture dishes has already been reported, syncytiotrophoblast-like cells cultured on dishes are not suitable for evaluating the transcellular transfer of drugs. On the other hand, the column-type 2D model developed in this study using TS cells has the potential for in vitro estimating transplacental transfer of drugs. Compared with existing models of the placental barriers using choriocarcinoma cells, primary cultured cells, or pluripotent stem cells, the model developed in this study can form a substantial barrier with a minimal paracellular transfer of substances between cells. Of course, some issues are still to be solved, including that the expression and function of membrane transporters involved in transcellular drug transfer are unclear. Nonetheless, the reviewer believes that this model will undoubtedly trigger establishing a gold standard model of the placental barrier. Following comments are raised to deepen understanding of the model:

Response: We greatly appreciate the reviewer's very positive comments about our trophoblast models and understanding of the potential of our models for in vitro estimating transplacental transfer of drugs. We revised our manuscript

based on the reviewer's suggestion.

Comment #1: Page 5, line 12: Please explain why authors consider that differentiation to ST cell was insufficient. The reviewer failed to clearly recognize the differences between organoids in Fig. S2A, S2B, and S2C.

Response: We are sorry for the lack of information. The culture condition to make trophoblast organoids with the pre-culture medium (PreM) and the week differentiation medium (W-DM) did not enable us to produce ST-covered trophoblast organoids efficiently. In other words, some organoids were not covered with ST cells totally, as shown in Fig. S3A(right figure) and S3B(c) which were additional figures in the revised manuscript, while the culture methods with three kinds of media ensured significant high production of ST-covered spherical organoids (Fig. 1). On the other hand, as we show with additional figures of Fig. S3C(b), the culture condition with the pre-culture medium (PreM) and strong differentiation medium (S-DM) was able to produce organoids totally covered with ST cells, but the surface of organoids are not smooth, which did not resemble the surface of the human placental villus. We added these explanations in the revised manuscript (page 6, lines 18-19 and 23-24)

Fig. S3

Fig. S3. Treatment of TS cells with two kinds of culture medium.

(A-C) TS cells were seeded into agarose microwell plates and cultured according to each culture schedule. Scale bars indicate 200 μm . Triangular marks indicate medium exchange.

(A) Immunostaining of SDC1 and E-cadherin and staining of nuclei with Hoechst. Frozen sections samples were prepared and imaged using BZ-X800/810.

(B) A representative image of trophoblast organoids (a), immunostaining of hCG and SDC1 (b), and the surface region highlighted in yellow showing lower expression of both hCG and SDC1 proteins (c). (b and c) Images were obtained by confocal microscopy.

(C) Immunostaining of SDC1 and E-cadherin and staining of nuclei with Hoechst(a). Frozen sections samples were prepared and imaged using BZ-

X800/810. Images of spherical trophoblast organoids were taken using a phase contrast microscope (b)(CKX53, Olympus).

Comment #2: Page 5, line 15: It is hard to identify the smoothness of the organoids from Fig. S2. Further explanations are needed.

Response: We agree with the reviewer and made Fig. S3C(b). The organoids shown in Fig. S3C(b) are apparently different from organoids in Fig. 1B and S3B(a) in the smoothness of the surface of organoids. Fig. S3B(a) and S3C(b) are shown above.

Comment #3: Page 5, line 18: Explanations for marker proteins such as SDC1, hCG, and E-cadherin should be put before explaining Fig. S2A and S2B.

Response: We thank you for the valuable comment. To explain the marker proteins of SDC1, hCG, and E-cadherin, we added the following sentences before the explanation of the previous Fig. S2A and S2B. " To identify the ST cells in the cell construct, expression of syndecan-1 (SDC1) and the beta-subunit of human chorionic gonadotropin (hCG) were analyzed. SDC1, a cell surface heparan sulfate proteoglycan, is apically localized in the placental villi and is not detected in other cells in the placenta (Jokimaa et al., 1998). hCG comprises α - and β -subunits and has multiple functions such as hormone production and differentiation of ST cells (Cole, 2010). SDC1 and the beta-subunit of hCG (CGB) can be used as ST maker proteins." (page 6, lines 11-17)
[1] V. Jokimaa, et al., Placenta 19 (2-3), 157-163 (1998).
[2] L. A. Cole, Reprod Biol Endocrinol 8, 102 (2010).

Comment #4: Page 5, line 36: Please explain why the device can facilitate the removal of dead cells.

Response: If some cells die within insert columns for cell cultures, the dead cells would be contained within the column as long as a medium change in the column is not conducted. To solve this problem, we cultured cells on the outside surface of the insert column. By doing so, dead cells fell to the bottom of culture wells by gravity. In the revised manuscript, we clarified this point with an additional explanation (page 7. lines 34-35).

This structure of the column device has other advantages. Cells can receive more nutrients and oxygen from the media because the well contains 800 μL (or 1300 μL in the revised protocol) medium, but the insert column has only 200 μL medium. We can see cells clearly from the bottom without struggling to see cells with the collagen membrane. Moreover, when we use the chopstick type of electrode to measure TEER values of the ST barriers, we can avoid scratching the cell layer with the electrode.

Comment #5: Page 7, line 3: Is it sure that fetal umbilical endothelial cells form tight junctions? If so, please cite the article that shows the presence of tight junctions in umbilical vessels.

Response: Some reports show that fetal umbilical endothelial cells form tight junctions. Leach et al. found that human-term placental capillaries have junctions consisting of wide zones of tight junctional regions discontinuously, which closely resemble other non-brain capillaries [1-3]. Newly formed vessels in the first trimester also possess tight junctions like in term placenta [4]. We used HUVECs (human umbilical vein endothelial cells) in the present study, which have been shown to express tight junction-related proteins in many reports [5-7]. As the reviewer suggested, we cited some of those articles in the revised manuscript (page 10. lines 4-5).

[1] L. Leach, et al., Cell Tissue Res 268 (3), 447-452 (1992).

[2] L. Leach, et al., Microsc Res Tech 38 (1-2), 137-144 (1997).

[3] L. Leach, et al., J Vasc Res 39 (3), 246-259 (2002).

[4] D. Elad, et al., Traffic 15 (1), 122-126 (2014).

[5] C. M. Chen, et al., Int J Mol Sci 21 (10) (2020).

[6] Y. Shi, et al., Int J Physiol Pathophysiol Pharmacol 12 (2), 70-78 (2020).

[7] R. Bhattacharya, et al., J Mol Signal 3, 14 (2008).

Comment #6: Page 7, line 27: It is necessary to discuss further why the Papp values of caffeine in TS-derived models are lower than that in BeWo cells. Is it due to the lack of caffeine transporter in TS-derived models?

Response: We thank the reviewer for the valuable suggestions. Caffeine can cross the placenta by passive diffusion [1]. Therefore, it penetrates the cell membrane of the placental barrier to reach the fetus. Barrier models with BeWo

cells usually consist of a monolayer of cells and therefore caffeine may penetrate the barrier easily. In contrast, our barrier models are constructed with ST cells having scaffolds of the undifferentiated cells on a collagen membrane layer. The differences in thickness probably provided the difference of Papp in both barrier models. We described these points in the revised manuscript (page 10, lines 34-36).

[1] T. Mose, et al., J Toxicol Environ Health A 71 (15), 984-991 (2008).

Comment #7: Page 7, line 30: Please clearly indicate that EGM-2 contains FBS. And please explain why FBS affects Papp values even though FBS is also included in S-DM.

Response: As the reviewer suggested, we clearly indicated that EGM-2 contains FBS at 2%. S-DM and a newly prepared S-DM2 contain FBS at 10%. FBS can induce the differentiation of TS cells to ST cells, and it affects Papp values. In the present study, we found that the differentiation schedule is important to make a robust ST barrier model, and FBS effects are better to be used at the third/final stage of the schedule. Therefore, we would like to optimize the timing and amount of HUVECs for addition in future experiments. We described these points in the revised manuscript (page 11, lines 4-8).

Comment #8: Fig. 1, legend: "Staining with phalloidin-FITC and Hoechst" is the explanation for (F), not (D).

Response: We thank the reviewer's indication. We reorganized figures and amended figure legends in the revised manuscript.

Comment #9: Fig. 4E: The reviewer would like to know why the TEER values in ST/HUVEC co-culture were significantly higher than those in ST. It looks inconsistent with the Papp values of antipyrine and caffeine.

Response: Because of their different natures, electricity and chemicals flow in different manners in the placental barrier models. Electricity flows through for example intracellular, intercellular, and gaps between cells and substrates. The electrical resistance value, i.e. TEER value, varies depending on how much electricity flowed through those pathways. It is thought that the thicker the cell

layer in total, the higher the TEER value, and the more the ST is fused, the higher the TEER value. The purpose of TEER measurements in this study is mainly to confirm whether cell membranes with resistance have actually been produced. This measurement is important because if there are surface regions in the basal membrane without being covered with cells, TEER values should be shown as they are very low. Small compounds cross the placental barrier depending on the nature of the chemicals.

On the other hand, small compounds cross the placental barrier depending on the nature of the chemicals. Some chemicals cross the barrier through, for example, passive diffusion, facilitated diffusion, or active transport, and this translocation manner is different from how electricity crosses the cell layers. Both antipyrine and caffeine are known to cross the placental barrier by passive diffusion, i.e., those chemicals mainly crossed intracellularly.

Our study demonstrated that the co-culture of ST/TSC and HUVEC show higher TEER values compared with the ST/TSC only. Although the mechanism of the synergistic effects is unclear, culture media or factors that are produced by HUVEC perhaps affected the tightness of the ST/TSC barrier. Moreover, some factors in the ST/TSC culture media may affect the shapes of HUVEC, which are shown in Fig. S10. We have amended the manuscript by explaining these things (page 7, line 36, page 8, lines 1-2, and page 11, lines 20-28).

Comment #10: Fig. S4: Further explanations regarding the cell types (x-axis) and gene names (FCGRT is the same with FcRn?) are needed.

Response: We added the following information regarding the cell types and gene names in the legend of Fig. S8A, as suggested. CT cells and syncytial sprout ST cells were collected from fresh 1st-trimester placental tissues. The number shown after the name of CT or ST (i.e., CT_1, CT_2, and CT_3, and ST_1, ST_2, and ST_3) mean that samples were isolated from the three placentas. FCGRT is the gene name for FcRn.

Reviewer #3 (Remarks to the Author):

General comments:

In this study the authors sought to establish a model, using human trophoblast

stem cells (TS), that could mimic the placental barrier in order to study how drugs cross the placenta. First, they develop a protocol for the formation of trophoblast organoids in which the syncytiotrophoblast (ST) cover the outer layer with cytotrophoblast (CT) on the inside. This protocol is then used to form layers of TS and ST on a modified column-type device to study their permeability to different substances. These two layers form a barrier (as shown by TEER measurement) that show characteristics similar to the placental barrier. The authors characterize the permeability of the barrier both to proteins (such as hCG, IgG and IgA) and to diffusible chemical compounds. The model is further improved by the addition of an endothelial layer (HUVEC cells).

A model that recapitulates placental villous structure and cell composition would represent an important tool to evaluate the transfer and toxicity of compounds from the maternal blood to the fetus through the placenta. However, there are several issues with this study. The reproducibility and technical aspects of the proposed model are not clear and important controls are missing from some experiments, making the results difficult to interpret and to evaluate whether this really could model the placental villus. The nature of the cells present and their organization within their model system also is not clear. Further work is needed on the culture conditions and experimental setup, to provide a more reliable model for placental barrier as well as a more detailed characterization of their system. Please see below for further details.

Response: We thank the reviewer for understanding the usefulness of our trophoblast organoids in the study of their permeability to different substances. We agree with the reviewer and understand the concerns about our ST barrier models. Fortunately, we believe that we were able to create ST barrier models which no longer have the issues the reviewer pointed out, although it took us a very long time. In additional experiments, we generated GFP-expressing TS cells and KSB (kusabira-orange)-expressing TS cells to demonstrate the differentiation of and the fusion of ST cells in the trophoblast organoids. Moreover, using the SDC1-GFP TS cells we made, we have sought to find conditions to make ST barrier models that entirely covered the collagen membrane of the column and finally found conditions to generate robust ST barrier models. Below, we answered the reviewer's comments one by one, including these points.

Major comments:

Comment #1: How many TS lines were used for this work (also per experiment)? This was not specified. Given that TS cells are derived from primary samples, to account for inter-individual variability, several independent, biological replicates are necessary for all experiments.

Response:

We thank the reviewer for the constructive suggestions. We used a trophoblast stem cell line named CT27 derived from cytotrophoblasts from a human placenta.

As we agree with the reviewer, we conducted additional experiments using two cell lines, CT29 and CT30, each derived from different human placentas. CT29 and CT30 were obtained in our previous study (Okae et al, Cell stem cell, 2018). In the revised manuscript, we were able to demonstrate that CT29 and CT30 also formed apical-out trophoblast organoids that were entirely covered with ST cells (Fig. S4B). Furthermore, we demonstrated that CT29 and CT30 form ST barrier models as well as CT27, which were determined by immunostaining with an antibody for SDC1 (Fig. S9B). Microvilli were observed also on the surface of the barrier models created from CT29 and CT30 (Fig. S9C). All cell lines in the present study formed organoids and barrier models, suggesting that our protocol can be applied at high reproducibility. We described these points in the revised manuscript (page 6, lines 35-37, and page 9, lines 32-34).

Fig. S4B

Fig. S4B. Immunostaining analysis of trophoblast organoids.

(B) Expression of SDC1 and E-cadherin in other cell lines, CT29 and CT30, of trophoblast stem cells. Two different trophoblast stem cell lines, CT29 and CT30, were immunostained as well as the CT27 cell line that was mostly used in the present study. The scale bar indicates 200 μm .

Fig. S9

Fig. S9. ST barrier models of each TS cell line (CT27, CT29, or CT30)
(A) Phase-contrast microscopic images of ST barrier models were created according to the 6 days protocol with S-DM2 in Materials and Methods.
(B) Immunostaining analysis with antibodies for syndecan 1 (SDC1) and Hoechst was conducted for the barrier models and images were taken using a fluorescent microscope BZ-X810 (Keyence). The scale bar indicates 200 μm .
(C) The surfaces of ST barrier models were analyzed using a scanning electron

microscope JSM-7900F (JEOL).

The scale bars indicate 200 μm (A and B), 10 μm (C, x3000), or 1 μm (C, x10000).

Comment #2: What is the efficiency of formation of organoids with a completely intact and single layer of ST (Fig. 1)? Further evidence of true multinucleated ST is needed besides Fig 1G. Please show an EM of this layer.

Response: Figure 1 shows that 10 of 10 organoids have a layer of ST cells entirely on the surface of the organoids. In the present study, the efficiency of the formation of organoids with the ST layer was almost 100% with the protocol shown in Fig. 1A. However, since this is the analysis of sections, parts of the surface in some organoids were technically damaged. We would like to determine the accurate formation rate of organoids with a completely intact and single layer of ST by solving this technical issue in future studies.

As the reviewer suggested, we conducted additional experiments to provide other evidence of true multinucleated ST in addition to the results of phalloidin-FITC (Fig. 2D) and hematoxylin and eosin (Fig. 2E). To do so, we generated two kinds of genetically-modified TS cells, i.e., GFP-expressing TS cells and KSB (kusabira-orange)-expressing TS cells. These TS cells show green or orange fluorescence, respectively. Therefore if the two kinds of cells were fused, they should be seen with yellow color under the fluorescence microscope (Fig. 2A). As we expected, we were able to see the yellow line along with the surface of the organoids. This result is direct evidence of ST fusion (Fig. 2B and C). The methods to generate these two cells were described in the Material and Methods, and we explained these points in the revised manuscript (page 7, lines 13-22).

As the reviewer suggested, we additionally analyzed the surface of the organoids by electron microscopy to further demonstrate the existence of ST cells on the organoid's surface and examine the existence of the microvilli. Fig. 1I clearly shows the microvilli at high density on the surface. This was similar to the microvilli on the in vivo placental villus (G. Burton, Scanning microscopy 4 (2), 29. 1990), strongly suggesting that the ST cells exist on the organoid's surface. We described these points in the revised manuscript (page 7, lines 7-9).

Fig. 2

Fig. 2. Analysis of the fusion of ST cells in apical-out spherical trophoblast organoids.

(A) The procedure for generation of the trophoblast organoids using a mixture of GFP-TS cells and kusabira-orange (KSB)-TS cells.

(B) An image of a trophoblast organoid in a microwell.

(C) Fluorescence images of a cross-section of the organoids generated from a mixture of GFP-TS cells and KSB-TS cells.

Fig. 1I

Fig. 1I. Generation of apical-out spherical trophoblast organoids

(I) Scanning electron microscopy images of a trophoblast organoid. The scale bars indicate 200 μm (B-I), 100 μm (I, x250), 10 μm (I, x850 and x3000), or 1 μm (I, x10000).

Comment #3: The authors show two new methods for the culture and differentiation of TS cells: generation of apical-out organoids in the microwell

plate (Fig. 1) and the formation ST barrier on a column insert (Fig. 2). As this is the first time that TS cells are seeded in these conditions, it is important to show a detailed characterization of the TS cells in these new undifferentiated conditions, verifying the maintenance of stem characteristics. Please show that they still express trophoblast stem cell and proliferation markers of CT for eg. TP63, GATA3, Ki67.

Response: As suggested, we additionally conducted immunostaining with normal IgG controls to characterize the undifferentiated cells in the trophoblast organoids. In addition to the staining of E-cadherin, a marker protein for undifferentiated cells, we used an antibody for staining GATA3 to characterize the cells of the organoids. In the human villus, GATA3 is expressed CT and some ST cells in the first-trimester placentas, and positive cells especially are significantly decreased in the third-trimester placentas (Mirkovic et al., 2015). GATA3 was positive in both ST and undifferentiated cells in our organoids (Fig. S4A), which characterize ST cells covering the organoids as more similar to those in the first-trimester placentas rather than those in the third-trimester placentas. We added this information in the revised manuscript (page 6, lanes 31-36).

[1] J. Mirkovic, et al., *Histopathology* 67 (5), 636-644 (2015).

Fig. S4A

Fig. S4. Immunostaining analysis of trophoblast organoids.

(A) Apical-out spherical trophoblast organoids were created with CT27 by the same procedure as described in Figure 1A. Immunostaining analysis with antibodies for syndecan 1 (SDC1), E-cadherin (E-cad), GATA3, normal mouse IgG (negative control), or normal rabbit IgG (negative control) was conducted for cross-sections of the organoids. The scale bar indicates 200 μ m.

Comment #4: A report has shown that the TS cells actually represent trophoblast cells from the cell column (CC) rather than CT (PMID: 32619492). Please stain for NOTCH1 and ITGA2 which are markers of CC (PMID: 27849611, PMID: 29540503).

Response: We thank the reviewer for the valuable comment. We stained our section samples with an antibody for NOTCH1 but the antibody and our staining

protocol did not work for the samples. Recently, the developmental stage of human TS cells has been reported to resemble that of CT which will arise immediately after implantation [1,2]. In the present study, we could not determine whether human TS cells resemble CC.

[1] G. Castel, et al., Cell Rep 33 (8), 108419 (2020).

[2] Y. Chen, et al., Development 149 (13) (2022).

Comment #5: The outside-in organoids are not truly physiologically representing the villous structure as VT are a single epithelial layer in vivo (as shown in their Fig. 1A) and not a clump of cells (Fig 1A-F). Please rephrase throughout the manuscript.

Response: As we agree with the reviewer, we rephrased the trophoblast organoids as they were derived by the culture method shown in Fig. 1A to “spherical trophoblast organoids” throughout the manuscript.

Comment #6: The seeding of the TS cells in the microwells is a key step for the formation of the apical-out trophoblast organoids, thus it should be mentioned in the main text (with reference to Fig. S1).

Response: As suggested, we mentioned the seeding of TS cells in microwells in the introduction. The following sentences were added (page 4, lines 24-28).
“In this study, based on the hypothesis that TS cells recognize ECM as the basal scaffold and do not form an ST layer outside of cell constructs, we seeded cells to agarose plates having micro-scale size wells (Fig. S2) instead of seeding cells in ECM. The microwell plates of agarose was used to avoid attaching of cells to the substrate and enhancing cell-cell adhesion under the efficient supply of nutrients and oxygen through agarose.”

Comment #7: From the images in Fig. 2, it is clear that the ST layer formed (even under the best conditions) is not a continuous layer and therefore it is not a proper barrier. This is also pointed out by the authors (p9 L33-35). This is a major limitation and concern as the formation of an intact barrier for a placental barrier model, is an essential requirement. Furthermore, as different lines will have different differentiation efficiency, the amount of ST formed will vary, increasing the variability between already an suboptimal system.

Response: We agree with the reviewer's comment that the formation of an intact ST barrier as a placental barrier model is an essential requirement. We have tried to increase the level of ST cell coverage on the undifferentiated cell layer. To find the condition to increase the level of coverage by ST efficiently and further demonstrate the ST differentiation, we generated a TS cell line that will have SDC1-GFP (green fluorescence protein) when the *SDC1* promoter is activated. We applied split fluorescent protein systems since it was difficult to transfect a much larger DNA construct into TS cells (Fig. S5). Following consideration of some culture conditions using the SDC1-GFP TS cells, we finally succeeded in increasing the levels of coverage up to about 100% (Fig. 4A-E). The main point we changed were the following:

1. We added forskolin, an activator of adenylyl cyclase, to the 3rd strong-differentiation medium (S-DM) at 2 μM to facilitate the differentiation from TS cells to ST cells. This medium was named S-DM2.
2. The culture schedule was shortened to 6 days from 7 days because forskolin is a very strong inducer into ST cells and too much differentiation leads to cell death.
3. We increased the amount of medium from 800 μL to 1300 μL to equalize the water level in the inner column and outer well. This is because a slight medium flow would occur from the inner column to the outer well that may affect cell differentiation if the water level in the inner column is higher than that in the outer well as in the previous experiments.
4. We added fibronectin to the Matrigel solution that was used for coating the collagen membranes to increase the adhesion of TS cells to the collagen membranes.

Furthermore, we demonstrated that CT29 and CT30 form ST barrier models as well as CT27 (Fig. S9A-B), as already mentioned in *Response* for *Comment #1*.

We revised the manuscript in the sections of Materials and Methods (page 21, lines 4-32, and page 23, lines 1-11) and Results (page 9, lines 16-34) in the revised manuscript.

Fig. S5

Fig. S5. Split fluorescent protein systems for SDC1-GFP proteins
 (C) Insertion of a DNA fragment for 7xGFP11 into the last exon of the *SDC1* gene.
 (D) A schematic illustration of the split fluorescent protein systems with 7xGFP11 and GFP1-10.

Fig. 4

Fig. 4. Enhancement of ST coverage in the barrier models
 (A) Schematic of additional factors for ST barrier formation.
 (B) A culture schedule for the improved ST barrier model.
 (C) SDC1 expression levels visualized using SDC1-GFP TS cells in the improved culture condition.
 (D) Immunostaining of SDC1 and fluorescence of SDC1-GFP and Hoechst in SDC1-GFP TS cells.
 (E) Immunostaining of SDC1 and fluorescence of Hoechst in TS cells (CT27 line).
 The scale bars indicate 200 μ m (C-E). Triangular marks indicate medium exchange.

Comment #8: In Fig. 3C the authors measure hCG in the well (apical side of the barrier) and in the column (basal side). Some critical controls are missing in this experiment. First, the TS-only control is only seeded for 1 day, while it would be more appropriate to keep the cells for the same number of days as the ST cells, without differentiation medium (the same applies to Fig. 2E). Second, the authors claim that the absence of hCG in the column demonstrates the formation of a barrier in the STM compared to TSM. However, in the TSM control, hCG is not present, thus it is not possible to see its diffusion in the column. To overcome this issue, hCG should be added in the wells of a TSM control seeded for at least 4 days (in which a barrier effect is visible in the STM) and measure its diffusion into the column.

Response: We thank the reviewer's comment. The purpose of this experiment shown in the previous Fig. 3C (Fig. 3L in the revised manuscript) is to reveal the timing of hCG secretion and the amount of the secreted hCG from ST cells in the barrier models in comparing our previous report showing those in 2D-cultured and 3D-cultured ST cells and to demonstrate the functional levels of the ST cells.

As the reviewer suggested, we additionally conducted ELISA with control samples of TS (Day 0-2) and TS (Day 2-4). Since some TS cells can spontaneously differentiate into ST cells in TSM after the cells become a high density, we detected hCG in samples of TS (Day 2-4) and the levels were lower than ST samples (Day 2-4) (Fig. 3L). Fig. 3L shows the amount of hCG in columns or wells (mIU/h), and the amount of culture medium in the wells was 800 μ L and that in the columns was 200 μ L. Therefore, if concentrations of hCG were exactly the same in the wells and columns, the amount of hCG in wells should be 4-fold higher than that in columns. In samples of TS (Day 2-4), the amount of hCG in the wells was 5.11-fold higher than that in the columns (Fig. 3L), i.e. the differences were 1.28-fold in concentrations, indicating that the TS cell sheet model barely functions as a barrier for hCG. We think that there are two possible reasons for the detection of hCG in columns in ST samples. One is that hCG was secreted from ST cells from the basal to the column. The other is a way that hCG secreted to the apical side comes back to the apical side and translocates into the basal side. We added sentences about these things in the revised manuscript (page 8, lines 27-35).

Fig. 3L

Fig.3 (L) Levels of secreted hCG (N = 3). * P < 0.05, Tukey's multiple comparison test. Data are shown as mean \pm SE.

Comment #9: The experiment with the immunoglobulins is not really helpful in showing the validity of the model. All they show in Fig. 3D-G is that both IgG and IgA are uptaken by the ST. They claim in the text that “the transition rate of IgG and IgA in comparison to controls was similar”. From the graph it is not possible to make this comparison, as the Y-axis scale is different between IgG and IgA, and the transition rate is very low. I would suggest using the % over the control in order to make the point. It is also not clear what the control “Cont” stands for. Furthermore, in Fig. 3F, it is not clear why the transition rate is so low. While the authors claim that “This finding raises the possibility that not only IgG, but also IgA can cross the ST barrier”, this is not supported by the data, which rather show that the Igs are retained in the ST cells and not released on the opposite side. Furthermore, as an appropriate control, the authors should compare the transition rate if IgG-FITC and IgA-FITC with a protein that is known to not being able to cross the barrier (e.g., hCG, that they showed to be retained in the wells).

Response: As we agree with the reviewer's comment that the experiment with the immunoglobulins is not really helpful in showing the validity of the models, we decided to focus on our main advances: i.e. this study is the first study to generate apical-out spherical trophoblast organoids and ST barrier models fully covered with fused-ST cells. We removed Fig. 3F and 3D in the previous

manuscript about the transition rate of IgG/IgA and moved Fig 3E and 3G about IgG/IgA uptaken by the ST into Fig. S8B and S8A, respectively. As suggested, to completely demonstrate the IgG/IgA transport to the fetus compartment the following experiments should be done. 1) Analysis of the uptake and excretion rates of each IgG/IgA to ST cells. 2) Accurate quantification of IgG/IgA that is translocated to the fetal compartment using mass spectrometry without labeling IgG/IgA with any fluorescent chemicals such as FITC. 3) Determination of localization of IgG/IgA by immunostaining for human placental section samples using anti-IgG/anti-IgA antibodies. 4) Mass-spectrometry to reveal receptors for IgG/IgA in syncytiotrophoblast in vivo. We would like to perform these experiments in future projects.

Comment #10: The authors reference through published data that FcRn, which is considered the mediator of transplacental passage of IgG, is expressed at low levels in the ST. The expression of this receptor should be checked also in the seeding and differentiating conditions that the authors use for the experiments in Fig. 3, to confirm the low expression in their model. This could be achieved by qPCR or by immunofluorescence, using cells that show the expression of the receptor as a positive control.

Response: We thank the reviewer's suggestion. As mentioned in *Response for Comment#9*, we removed Fig. 3D and F in the revised manuscript and would like to further investigate the study on IgG/IgA in future studies to focus on the main advances.

Comment #11: The integrity of the HUVEC layer (Fig. 4B-D) needs to be evaluated using confocal microscopy of the cell-cell junctions, as from the images shown in the figure it is not clear whether they form a continuous layer.

Response: We thank the reviewer's valuable comment. For the co-culture models of ST and HUVEC, we additionally analyzed the cell-cell junctions using confocal microscopy and an antibody for VE-cadherin, but we could not clearly analyze it because of adjacent the trophoblastic layers interfering with the analysis of the top HUVEC layer in the column.

We believe that the co-culture of the ST barrier layer with HUVEC strengthens the ST barrier because additional HUVEC increased TEER values.

However, a part of the HUVEC layer appeared not to form a continuous layer, which was thought by results in Fig. 5B and S11B. To address these limitations of this study and to create a courteous complete layer of HUVEC along with the ST barrier, optimization of the seeding number and timing of HUVEC need to be conducted in future studies. We have described this limitation of our study in the revised manuscript (page 14, lines 3-8).

Comment #12: Fig. 4F-H: the graphs show a difference in Antipyrine and Caffeine Papp between ST and ST/HU conditions, but the graphs lack an indication of the significance of this difference. If the difference was significant, it should be experimentally addressed. The authors postulate that the difference could be due to FBS affecting the thickness of the undifferentiated cell layers in the ST/HU condition. This hypothesis could be easily proved by adding a condition with ST layer, and HUVEC medium in the column (without the HUVEC cell layer). Furthermore, given the alteration in the media composition to which the underlying TS cells are in contact, the authors should characterize the features of the TS cells in terms of stemness and integrity of the layer.

Response: As one of the major revisions for this manuscript, we succeeded in creating ST barrier models that are entirely covered with ST cells. In the results of the revised ST and ST/HUVEC models, the difference between both models was a little. The integrity of the layer was determined by TEER measurement (Fig. 5F).

Since the number of TS cells used was lesser (ca 70% of the previous cell number) and culture time was shorter (6 days) in the revised protocol, the revised ST/TS layer should be thinner than the previous one. However, we could not accurately determine the thickness of the thinner barrier models because of their vulnerability in making section analyses. We think that the difference in the Papp values between the revised barrier models and the previous models was probably due to the thickness of the barriers.

Since FBS was used as an ST differentiation inducer in this experiment, FBS in HUVEC medium may affect the thickness of the undifferentiated cell layers, but it may not be the only factor affecting the thickness. For example, proteins secreted from HUVEC may affect ST cell differentiation and barrier integrity. We have amended the manuscript with an explanation of these points instead of conducting the experiment that we culture ST cells in EGM-2 media

(page 10, lines 4-8)..

Comment #13: Fig. 4I: The Glyphosate Papp is already very low in the control, as pointed out by the authors, possibly due to an effect of the collagen membrane. Is there any evidence that these values are similar to the chorionic connective tissue barrier? Without this information, it is not possible to understand if there could be a bias in the results of chemical screens using the proposed model. A comparison of the permeability to Glyphosate of the connective tissue and the collagen membrane would solve this issue.

Response: We thank the reviewer's comment. We used columns consisting of Type I collagen membrane since it is known that Type I collagen was the basic structural unit of the human term placenta [1]. This collagen membrane is similar to in vivo connective tissues of the placental villus more than artificial polycarbonate or polyester porous membranes.

Unfortunately, in the Papp values of glyphosate, based on the results of additional experiments, we concluded that our previous experiments with mass spectrometry had technical problems with sensitivity for the analysis and that we did not provide accurate values of concentrations of glyphosate in samples. The technical issues include the possibility that the detector or vacuum system was defective or the condition of the analytical column was at low performance. After replacing the detector, vacuum system, and column with new ones, we performed mass spectrometry again (Fig. S10I). We found that Papp of glyphosate in the control collagen membrane is almost similar to Papp of antipyrine and caffeine. This should be due to that the hydrated collagen membrane has large pores that small molecules can pass easily.

In this additional experiment, Papp values of glyphosate in the past models of ST or ST/HUVEC were below the limit of detection (0.41×10^{-6} cm/s). As additional experimentation, we performed mass spectrometry for Papp values of glyphosate in the revised ST and ST/HUVEC models and confirmed that Papp values of glyphosate were lower than those of antipyrine and caffeine also in the revised models (Fig. 5G). The Papp values of glyphosate were about one-third of the Papp values of caffeine, which is a similar tendency in the translocation of glyphosate ex vivo perfusion experiments.

[1] Amenta PS, Gay S, Vaheiri A, Martinez-Hernandez A. The extracellular matrix

is an integrated unit: ultrastructural localization of collagen types I, III, IV, V, VI, fibronectin, and laminin in human term placenta. *Coll Relat Res.* 1986 Jun;6(2):125-52. doi: 10.1016/s0174-173x(86)80021-8. PMID: 3731745.

Fig 5G

Fig. 5. ST/HUVEC co-culture barrier models and translocation of drugs. (G) The permeability of reference compounds in control (Cont, collagen membrane), HUVEC (HU), ST, and ST/HUVEC models (N = 4). * $P < 0.05$, Dunnett's test. Data are shown as mean \pm SE.

Fig. S11I

Fig. S11I. ST/HUVEC co-culture barrier models and translocation of drugs. (I) The permeability of reference compounds in control (Cont, collagen membrane), HUVEC (HU), ST, and ST/HUVEC models [N = 4 (except for HU and ST in the analysis of caffeine, N = 3)]. * $P < 0.05$, Dunnett's test. Data are shown as mean \pm SE.

Comment #14: Fig. S5: it is not very clear what the fluorescent images should

show. The images show very different background. To be comparable, the images should be acquired with the same setups and subjected to the same brightness/contrast and background removal. In this case, to evaluate the morphology of the cells, a close-up would be required. Staining of the nuclei and the cell junctions are needed to show the integrity of the layer.

Response: The images shown in Fig. S5 (Fig. S11 in the revised manuscript) were acquired with the same setups and subjected to the same brightness/contrast and background removal. Morphological differences in HUVECs cultured in different media affected the brightness and background in each image. As the reviewer suggested, we provided close-up images for each figure in the revised version.

Fig. S11

Fig. S11. Morphological change of GFP-HUVECs

(a) GFP-HUVECs were cultured in EGM-2 for 2 days.

(b) GFP-HUVECs were cultured in PreM for 1 day, followed by growth in W-DM for 1 day.

(c) TS cells were seeded onto the bottom surface of the collagen membrane and cultured in PreM for 2 days and then W-DM for 1 day. On day 1, GFP-HUVECs were seeded in the column insert and cultured in EGM-2 for 2 days.

EGM-2, endothelial cell growth medium 2 (Lonza); PreM, pre-culture medium; W-DM, weak differentiation medium. The scale bar indicates 200 μ m.

Minor comments:

Minor comment #1: Details are missing in the methods:

- o The derivation of the TS cells and of human placental villi are missing.
- o Some details are missing regarding the imaging setup (e.g., objectives and cameras used for the acquisition)
- o In the immunofluorescence method section, there is no mention of the organoids in the microwell plates. Please specify in the methods if the frozen sections were used for this purpose.
- o In the immunofluorescence method section, please specify how many times the stainings were performed in independent experiments.
- o The H&E staining is not mentioned in the methods
- o Method “fabrication of a column-type device for barrier models”: it is not clear the reason for the addition of a polyimide film and a silicone ring.

Response:

Thank you for pointing these out. We have answered the below one by one.

- o The derivation of the TS cells and of human placental villi are missing.

Response:

We have added a description of the derivation of the TS cells and of human placental villi (page 20, lines 29-30).

- o Some details are missing regarding the imaging setup (e.g., objectives and cameras used for the acquisition)

Response:

We have added information about microscope used in each figure legend and added information on objectives in the revised manuscript (page 24, line 1).

- o In the immunofluorescence method section, there is no mention of the organoids in the microwell plates. Please specify in the methods if the frozen sections were used for this purpose.

Response:

We have added the following sentences in the immunofluorescence method section (page 24, lines 3-5). “For staining cells of frozen sections, cells were washed with PBS(-) twice and followed by the same staining procedure as cells on the column-type devices were subjected to.”

o In the immunofluorescence method section, please specify how many times the stainings were performed in independent experiments.

Response:

We have specified times the staining in the immunofluorescence method section (page 24, lines 5-7). “Samples of ST barrier models and section samples of spherical trophoblast organoids were stained with antibodies for SDC1, hCG, or E-cadherin in more than triplicates in over three independent experiments.”

o The H&E staining is not mentioned in the methods

Response:

We have added details on the H&E staining in the methods (page 23, lines 21-23). “For hematoxylin-eosin (H&E) sections, TS organoids were fixed with 4% paraformaldehyde (PFA) for 60 min and stored in 1% formalin neutral buffer solution (Fujifilm Wako), followed by conventional paraffin embedding and H&E staining (Advantec).”

o Method “fabrication of a column-type device for barrier models”: it is not clear the reason for the addition of a polyimide film and a silicone ring.

Response:

While creating ST cell barrier models, we thought fabricating a larger ST barrier model would be more difficult, so we reduced the surface area. A silicone ring was set to place a cell suspension on the collagen membrane. We have amended the manuscript (page 20, line 23).

Minor comment #2: Details are missing in the figure legends:

o When an immunofluorescence image is shown, please always specify the acquisition method (e.g., confocal or widefield)

o The legends should specify at which differentiation stage the cells were fixed and stained

o The legends should specify if N refers to independent experiments or technical replicates

o Fig. 1: the F is missing (D should be an F)

o Fig. S4: please include a short summary about the generation and analysis of

the data

Response:

We thank the reviewer for pointing these out. We have answered the below one by one.

o When an immunofluorescence image is shown, please always specify the acquisition method (e.g., confocal or widefield)

Response:

In each figure legend, we have specified the kinds of microscope used for the acquisition of each figure or specified acquisition method.

o The legends should specify at which differentiation stage the cells were fixed and stained

Response:

We have specified differentiation stages (days of cultures) in figure legends.

o The legends should specify if N refers to independent experiments or technical replicates

Response:

We have specified that N refers to replicates (different samples) in the legend of Fig. 3.

o Fig. 1: the F is missing (D should be an F)

Response:

We thank the reviewer for pointing this out. We reorganized figures at this time of revision.

o Fig. S4: please include a short summary about the generation and analysis of the data

Response:

We have included include a short summary of the generation and analysis of the data in the legend of Figure S8 (previous Fig. S4).

Minor comment #3: Fig. 1E: is it a 3D reconstruction of a series of confocal sections (same applies to S2B)? Please specify how the samples were prepared and how these images were acquired. These images are not informative, a section of the costaining hCG-SDC1 would show better their colocalization in the ST layer.

Response:

Yes, it is a 3D reconstruction of a series of confocal sections (the same applies to the previous S2B). We have specified the sample preparation procedure and acquisition method of images in the legend of Fig. 1.

As the reviewer suggested, we conducted additional experiments to stain both hCG and SDC1 in the same sections to show better their colocalization in the ST layer (Fig. 1E) (page 6, line 28).

Fig. 1E

Fig. 1. Generation of apical-out trophoblast organoids (C-E, and H) Immunostaining of syndecan 1 (SDC1), the beta subunit of the human chorionic gonadotropin (CGB), or E-cadherin (E-cad) in cross-sections of spherical trophoblast organoids. Images were obtained using BZ-X800/810 (Keyence).

The scale bars indicate 200 μ m (B-I).

(C-F, H-J) Images from the organoids of day 8.

Minor comment #4: Fig. S4: In the graph there isn't any standard deviation,

and it is not clear how many times the experiment was repeated.

Response:

We thank the reviewer for pointing this out. The number shown after the name of CT or ST (i.e., CT_1, CT_2, and CT_3, and ST_1, ST_2, and ST_3) mean that samples were isolated from the three placentas. In other words, analysis for CT or ST was conducted in triplicate. We added the abovementioned information in the legend of Fig. S8.

Minor comment #5: The considerations about the medium that are present in the discussion should be moved in the main text (Fig. S6 and S7)

Response:

As the reviewer suggested, we have described the considerations about the medium in the Results sections in the main text (page 8, lines 12-17).

REVIEWER COMMENTS

Reviewer #1 (Remarks to the Author):

Overall, the authors have undertaken significant work to a) further characterise and demonstrate syncytiotrophoblast differentiation in their organoid/spheroid model, and b) improved the integrity/coverage of the ST in their barrier model. They have undertaken all corrections around wording ambiguity suggested on my part, and in particular improved the discussion of the manuscript. I have several remaining comments:

- Introduction, line 33 - Since the original review of this paper additional barrier models using primary trophoblast have been published in the literature (e.g. doi: 10.1111/cpr.13469.) - it is no longer appropriate to say that no STB barrier models from primary cells have been established so far. The publication referred to above (and possibly other recent publications) also need to be considered in the statements on page 4 lines 8-10.

- Whilst the additional experiments with green and red expressing TS lines, and fluorescent reporting of expression of SDC1, provide further evidence of cell fusion (and I am convinced that this is infact occurring), they do not replace the use of appropriate negative controls in immunohistochemistry experiments as requested and so I was somewhat confused as to why these experiments were used as a response to this comment. Appropriate negative controls are now shown in additional IHC experiments (S4A) and details added to the methods. However, the comment "we conducted additional experiments for the negative controls" makes me concerned that these controls were not done at the same time as the experiments presented in Fig 1, which means they are not true controls for staining or imaging/autofluorescence purposes.

- Page 9 (line 24) - minor (not miner)

- Page 11, line 14 - I am confused here that the authors talk of developing papp models "for the ST models with S-DM but not S-DM2" but then go on to say that "Although Papp values for antipyrine and caffeine were not much different 17 from the ST models with S-DM2 (Fig. S11F-H)" - so this comparison was infact made? By S-DM2 are the referring to the improved ST model with S-DM2 and forskolin? I suspect the later is true as in the response to reviewers the authors talk of assessing the permeability of caffeine and glyphosphate using the revised ST barrier model, but this direct comparison between the two systems and experiments done in each did not come across clearly in the results text as it is written and this could use revising (it was much more clearly explained in the response to reviewers).

- One of the key advantages of traditional organoid models (over spheroid models) is that they can be disassociated and passaged. Could the authors comment on whether this is possible in their spherical organoid model?

- The flow of the manuscript would be improved by the authors ensuring they brought out the key points and comparisons clearly in the results, and used consistent terminology around conditions throughout. They did a good job of this in the response to reviewers, but presentation of data in the manuscript itself was at times more difficult to follow.

Reviewer #3 (Remarks to the Author):

In this revised version of their manuscript, Hori and colleagues have included additional data and reproduced results with independent trophoblast stem cell lines, which has substantially improved their study. However, there are still shortcomings in their work that are not resolved, thus some of the

initial concerns remain. These are summarized under three points:

1. Definitive proof of the formation of an intact syncytial barrier in their models is lacking. The key point of this work is the generation of new trophoblast organoid models that present an accessible, intact placental (syncytial) barrier. Thus, it is essential that this is clearly demonstrated. These methods will be of great interest for researchers that require an intact barrier, for example studies on infection, and thus will impact their work if this is not the case. Further experiments are needed and the issues are listed below.

-Fig. 3F and Fig 4D, E: please provide more details on how the percentage of the area covered by SDC1-expressing cells. Fig4E does not show full coverage by SDC1+ cells but it is stated ca. 100%. An intact barrier means it is either 100% or not so an approximation is not sufficient. This also applies to other images in which 'ca. 100%' is stated (Fig. 5B, Fig. S9B). Furthermore, throughout the manuscript the images clearly show that there are patches of SDC1+ cells and not an intact layer (Fig S11B).

-Fig. 3K: TEER values are used as a measure of the integrity of the ST layer. However, an increase in these values does not provide evidence of an intact layer, as the presence of any additional cells that would increase resistance. Furthermore:

- P8L24: it is not clear what is intended by 'high' in this context. The values are higher than in TSM, but is it considered 'high' compared to other cell barriers?
- Fig. 3K: please include the TEER measurements in the same graph in order to be able to directly compare the values with the previous model and to assess if there is an improvement
- Fig. 5F: the TEER of ST is much lower compared to the previous values in Fig. 3K for the same condition (ST), why is this so?
- Fig. S11E: Another example of TEER values being very different to the other measurements found in the other graphs. How reproducible are these conditions? If the layer is intact, then shouldn't the values be consistent?

-Fig. 3L: the data should be normalized to the amount of media for each condition

-Fig. 4: Fig. 4C at day 6 look very different from figure D in term of SDC-1-GFP localization and coverage. As stated in the text, both pictures have been taken at day 6..can the author comment on that? Plus, figure D there is not colocalization between SDC-1 and SDC-1-GFP. Generally, they claim there is a 100% coverage, but the images do not support what they claim and a functional assay is required (dextran).

-There are no functional experiments that demonstrate the presence of an intact syncytium. The IgG/IgA experiment was not conclusive and was simply moved to into supplementary Fig S8. Are there alternative experiments that could be done, for instance looking at whether fluorescently-labelled small molecule (FITC-dextran) crosses, which is a commonly used assay to assess a cellular barrier?

2. Insufficient characterization of the trophoblast cell states captured in the organoid model. In the generation of new models, a key aspect is to characterize the cell types present in the model and the cellular organization within the system to have a clear understanding of which part of the tissue is recapitulated and potential limitations.

-In the rebuttal P26#4, the authors reply that they could not get NOTCH1 staining to work. This is not a satisfactory reply, as there are specific antibodies for immunofluorescence for NOTCH1 (for example see PMID: 27849611) as well as ITGA2 (PMID:27849611, PMID: 29540503). If staining does not work, then as an alternative approach the expression of these markers can also be assessed by FACS. Furthermore, the references cited in their reply refer to human ESC derived trophoblast cells and which are different to the blastocyst or 1st trimester placenta derived TSC used in this work. The latter

have been assessed clearly to be ITGA2-positive and therefore representing cells of the cell column in two independent studies (PMID:27849611, PMID: 29540503). It is important to clarify which cell types are present in the model, please assess either NOTCH1 or ITGA2 expression.

-Fig 1C, D, G, H and Fig. S4B: SDC-1 and hCG staining can be clearly seen both inside and in the outer layer of the organoids. This suggests that syncytium is also forming also on the inside of the organoid which is not physiological. The authors should comment on this.

-Fig 1I: TEM is needed to show multinucleated region in the STB, only presence of microvilli is shown.

-Fig. S4: the quality of the images and/or staining is low making it difficult to assess them. For example, E-cadherin is not clearly on the cell surface and GATA3 should be nuclear. Please include better quality images in which the localization of these markers can be assessed more clearly and also the inclusion of a proliferative marker, Ki67 would be helpful.

-P13L23-25, Fig.S7D: not possible to assess if there is a release of ST cell aggregates, there is no characterization. Fig. S1: please show cells before and after ST differentiation to compare.

3.The endothelial compartment is not properly formed. The inclusion of an endothelial component (HUVEC) to increase the complexity of their model is nice, however the data shown does not support an improvement for the following reasons:

-The presence of the three layers (HUVEC, CT and ST) has not been demonstrated. Fig. 5D shows only ST (SDC1+) and HUVEC (GFP). According to Fig 5H, the barrier model should have the three layers. Please include a staining with a CT marker for example, E-Cadherin.

-Fig. 5B and D: HUVEC line is able to form endothelial tubes but in these images, it seems that it does not even form a continuous layer and the cells are present but sparse. For the inclusion of these cells to add value to the model, it would require a formation of a layer. Could a different medium be used that is in direct contact with the HUVEC only to promote this?

-Fig.5A and B: what is the medium used in B? P10L12 the use of EGM-2 is mentioned but it is not mentioned in 5A.

-Fig. 5C: P11L4 'co-culture with HUVEC slightly increased the Papp values of antipyrine and caffeine' is not supported by the data, as a significant difference between ST and ST/HU is not evident (please include p-value).

Minor comments

-Consistency with nomenclature: in Fig 1 organoids are referred to as spheroids and also spherical organoids throughout.

-Abstract, L13: 'revealed that and that'

-Abstract L10-11, due to the lack of experiments that do not demonstrate the formation of an intact syncytium, the authors should be more cautious of their claims. This is valid throughout the manuscript.

-P4L12: 'shrinkage' is not clear what the authors mean

-P5L2: human placental villous

-P6L31: GATA 3 is expressed 'in' CT

-Fig.3: G and H, please add in the figure legend, the protocol/culture conditions that were used for these images for clarity.

-Fig S8B legend: 'intensicy' instead of 'intensity'

-P12L20: ST cells of the villi do not attach to decudua

-P12L21: 'anchoring villous' - 'villus'

-P14L21: repetition of 'pathology'

Trophoblast stem cell-based organoid models of the human placental barrier

Takeshi Hori, Hiroaki Okae, Shun Shibata, Norio Kobayashi, Eri H Kobayashi,
Akira Oike, Asato Sekiya, Takahiro Arima, and Hirokazu Kaji

Point-by-point response to the reviewers' comments

Reviewer #1:

General comments:

Overall, the authors have undertaken significant work to a) further characterise and demonstrate syncytiotrophoblast differentiation in their organoid/spheroid model, and b) improved the integrity/coverage of the ST in their barrier model. They have undertaken all corrections around wording ambiguity suggested on my part, and in particular improved the discussion of the manuscript. I have several remaining comments:

Response:

We are very grateful to the reviewer for carefully reading our manuscript. We have revised the manuscript based on the reviewers' constructive suggestions.

Comment #1:

- Introduction, line 33 - Since the original review of this paper additional barrier models using primary trophoblast have been published in the literature (e.g. doi: 10.1111/cpr.13469.) - it is no longer appropriate to say that no STB barrier models from primary cells have been established so far. The publication referred to above (and possibly other recent publications) also need to be considered in the statements on page 4 lines 8-10.

Response:

We appreciate the reviewer's suggestion. In the recently published paper (doi: 10.1111/cpr.13469), the researchers used trophoblast stem cells derived from human blastocysts using the method we previously reported (Okae, et al., Cell Stem Cell 22 (1), 50-63 e56, 2018). They did not use primary trophoblasts in the study. They have cultured the TS cells in a microfluidic device as we previously reported in international meetings [1-2]. We analyzed the ST coverage against the basal substrate for their barrier model shown in Fig S1A of their manuscript in the same manner and found that it is 49.9% coverage. Thus, our barrier models having about 100% ST coverage in the revised manuscript have great

advantages in assessing transplacental passages of materials. According to the reviewer's suggestion, we have included the paper [3] in the revised manuscript and added an explanation about differentiation ratios into ST cells (page 4, lines 13-15).

[1] Hori T, Okae H, Kobayashi N, Arima T, Kaji H. Recapitulating the Human Placental Barrier with Trophoblast Stem Cells and a Microfluidic Device. The society for biomaterials (SFB) 2022 Annual Meeting, 2022

<https://biomaterials.org/sites/sfb/files/docs/2022/v123.pdf>

[2] Hori T, Okae H, Kobayashi N, Arima T, Kaji H. A trophoblast stem cell-based model of the human placental barrier., MicroTAS 2021, 12th Oct 2021.

[3] R. Cao, et al., Cell Prolif 56 (5), e13469 (2023).

Comment #2:

- Whilst the additional experiments with green and red expressing TS lines, and fluorescent reporting of expression of SDC1, provide further evidence of cell fusion (and I am convinced that this is infact occurring), they do not replace the use of appropriate negative controls in immunohistochemistry experiments as requested and so I was somewhat confused as to why these experiments were used as a response to this comment. Appropriate negative controls are now shown in additional IHC experiments (S4A) and details added to the methods. However, the comment "we conducted additional experiments for the negative controls" makes me concerned that these controls were not done at the same time as the experiments presented in Fig 1, which means they are not true controls for staining or imaging/autofluorescence purposes.

Response:

We are sorry for confusing you by giving a different answer when you asked about negative control of staining for ST cells in Comment #6 in the previous comments. We agree with the reviewer's comment that trophoblast are notoriously autofluorescent and the use of an irrelevant IgG control antibody is essential.

As the reviewer's suggestion, we correctly conducted immunostaining on the same day (the same time) for SDC1, hCG, GATA3, irrelevant IgG control s (normal mouse IgG or normal rabbit IgG) in the previous Fig 1E and Fig. S4. Staining for E-cadherin in the previous Fig. S4 was individually conducted on another day and analyzed at the same procedure and conditions in staining and

fluorescence microscopy, i.e., the same time length of staining and light exposure in observation.

To conduct immunostaining with an IgG control for E-cadherin, we have conducted additional experiments and confirmed specific staining of E-cadherin in the section samples of the spherical trophoblast organoids. The results of E-cadherin are shown in Fig. S5A in the revised manuscript. The images shown in Fig. R1 were prepared for review only because similar results of negative controls are already shown in Fig. S5A.

We agree that showing the results of ST fusion was not an appropriate way to answer your previous comment about negative controls in immunostaining. To respond to another reviewer's concern about the existence of ST cells and the fusion of ST cells, we conducted additional experiments using GFP-expressing TS cells and KSB-expressing TS cells. We thought that the result we successfully obtained is further evidence of the existence of ST cells on the surface of spherical organoids and supports the conclusion that comes from immunostaining with the anti-SDC1 antibody. Thus, we used these results as a response to your previous comment.

Fig. R1 (review only)

Fig. R1. Immunofluorescence staining images of E-cadherin.

(A) Apical-out spherical trophoblast organoids were generated from a TS cell line CT27 by the same procedure as described in Figure 1A. Immunostaining analysis with antibodies for E-cadherin or normal rabbit IgG (negative control) was conducted for cross-sections of the organoids. The scale bars indicate 200 μm .

Fig. S5

Fig. S5. Immunostaining analysis of trophoblast organoids.

(A) Apical-out spherical trophoblast organoids were generated from a TS cell line CT27 by the same procedure as described in Figure 1A. Immunostaining analysis with antibodies for syndecan 1 (SDC1), E-cadherin, TP63, Ki67, GATA3, normal mouse IgG (negative control), or normal rabbit IgG (negative control) was conducted for cross-sections of the organoids. The scale bars indicate 200 μm or 100 μm (magnified images).

Comment #3:

- Page 9 (line 24) - minor (not miner)

Response:

We are sorry for the typo. We revised it.

Comment #4:

- Page 11, line 14 - I am confused here that the authors talk of developing papp models "for the ST models with S-DM but not S-DM2" but then go on to say that "Although Papp values for antipyrine and caffeine were not much different 17 from the ST models with S-DM2 (Fig. S11F-H)" - so this comparison was infact made? By S-DM2 are the referring to the improved ST model with S-DM2 and forskolin? I suspect the later is true as in the response to reviewers the authors talk of assessing the permeability of caffeine and glyphosphate using the revised ST barrier model, but this direct comparison between the two systems and experiments done in each did not come across clearly in the results text as it is written and this could use revising (it was much more clearly explained in the response to reviewers).

Response:

We are sorry for causing the confusion. Yes, this comparison was, in fact, made. We can directly compare Papp results between the two systems. In the paragraph that the reviewer pointed out, we mentioned the results of Papp for the ST models with S-DM medium, although we mentioned the results of Papp values for the ST models with S-DM2 right before the paragraph. We are sorry for the sentence that can cause misleading. We have amended the sentence in the revised manuscript (page 11, line 14, and page 12, lines 1-3).

S-DM2 is the modified S-DM containing forskolin (page 9, line 32). We have added this information also in the Materials and Methods section in the revised manuscript (page 19, line 8).

Comment #5:

- One of the key advantages of traditional organoid models (over spheroid models) is that they can be disassociated and passaged. Could the authors comment on whether this is possible in their spherical organoid model?

Response:

We thank the author's constructive comment. The passage of our spherical organoids is interesting, and we have conducted additional experiments. We found that our organoids could not be passaged directly from on the gel plate to the gel plate (Fig. S6A). During the process of digestion and disassociation, some cells were damaged. The damaged cells were not

removed from the microwells in our agarose microwell plates, probably affecting the formation of the spherical organoids. Instead, we were able to re-culture the TS cells, which were obtained from the organoids digested and disassociated, in 6-well plates (Fig. S6A and B). The re-cultured TS cells had the ability to form spherical organoids (Fig. S6C and D). Thus, our organoids can be passaged after re-culturing. We added this information in the revised manuscript (page 7, lines 5-9).

On the other hand, in regard to cell maintenance (passage) in 2D cultures, TS cells have a significant advantage over the primary trophoblasts that were used for traditional organoid models, i.e., TS cells can be passaged over 80 times, but the primary cells cannot. Therefore, we believe that our protocol is useful to generate the trophoblast organoids easily and reproducibly.

Fig. S6

Fig. S6. Passage of apical-out spherical trophoblast organoids

(A) The spherical trophoblast organoids were generated from CT27 by the same procedure as described in Fig. 1A and then digested with TrypL Express Enzyme (1X) for 40 min. After pipetting, cells were filtrated with 20 μ m pluriStrainer (Cat# 43-50020-01, pluriSelect) and re-seeded to the agarose microwell plates or wells of 6-well plates. Cells in the microwell plates were cultured with the medium for generation of apical-out spherical trophoblast organoids. Cells in wells were cultured with TS medium.

(B) The shapes of the re-seeded cells were compared with the normally cultured TS cell lines (CT27, CT29, and CT30).

(C) The re-seeded cells on the wells were harvested and seeded to agarose microwell plates and cultured to generate the spherical organoids.

(D) Frozen sections samples were prepared, stained with given antibodies, and imaged.

Images of spherical trophoblast organoids were taken using a phase contrast microscope CKX53 (A-C) or BZ-X800/810 (D). The scale bars indicate 200 μm .

Comment #6:

- The flow of the manuscript would be improved by the authors ensuring they brought out the key points and comparisons clearly in the results, and used consistent terminology around conditions throughout. They did a good job of this in the response to reviewers, but presentation of data in the manuscript itself was at times more difficult to follow.

Response:

We thank the reviewer for the positive comments and suggestions. We reconfirmed the flow of the manuscript and made it clearer by adding some sentences for readers.

Reviewer #3:

General comments:

In this revised version of their manuscript, Hori and colleagues have included additional data and reproduced results with independent trophoblast stem cell lines, which has substantially improved their study. However, there are still shortcomings in their work that are not resolved, thus some of the initial concerns remain. These are summarized under three points:

Response:

We thank the positive comment about the substantial improvement of our manuscript. Fortunately, we have fully addressed your suggestions in this revised manuscript. The quality of the manuscript has significantly improved thanks to your suggestions. We sincerely appreciate the reviewer's constructive suggestions.

Comment #1-1:

1. Definitive proof of the formation of an intact syncytial barrier in their models is

lacking. The key point of this work is the generation of new trophoblast organoid models that present an accessible, intact placental (syncytial) barrier. Thus, it is essential that this is clearly demonstrated. These methods will be of great interest for researchers that require an intact barrier, for example studies on infection, and thus will impact their work if this is not the case. Further experiments are needed and the issues are listed below.

-Fig. 3F and Fig 4D, E: please provide more details on how the percentage of the area covered by SDC1-expressing cells. Fig4E does not show full coverage by SDC1+ cells but it is stated ca. 100%. An intact barrier means it is either 100% or not so an approximation is not sufficient. This also applies to other images in which 'ca. 100%' is stated (Fig. 5B, Fig. S9B). Furthermore, throughout the manuscript the images clearly show that there are patches of SDC1+ cells and not an intact layer (Fig S11B).

Response:

We thank the reviewer for constructive suggestions. We analyzed the percentage of the area covered by SDC1-expressing cells using ImageJ software with the specific parameter and showed the method in Fig. R2. As the reviewer suggested, we provided more details on ST coverage in Table R1 and removed "ca." from all figures about ST coverage, i.e., Fig. 4E, Fig. 5B, and Fig. S13B.

As the reviewer pointed out, 100% ST coverage, which the reviewer calls an intact barrier, has not yet been achieved, although our 99% ST barrier models have significantly improved the ST coverage compared to other models reported.

With additional experiments, we found that we were able to generate 100% ST coverage models by reducing the circle area (collagen membrane area) and reducing the possibility of seeing the uncovered area, but reproducibility was not excellent (Fig. R3). Although further investigation is required to make 100% ST coverage models in high reproducibility, we believe that with our current ST barrier models, researchers can investigate studies on infection and material transfer by comparing the undifferentiated models such as TS barrier models.

Fig. R2 (review only)

Fig. R2. Analysis of ST coverage (example images).

ST coverages were calculated using ImageJ version 1.47t (National Institutes of Health). Briefly, RGB images were split into individual grayscale channels by Image/color/Split Channels. Then, the circle area (target area) was selected. Then, the ST area was colored with a specific parameter by “Image/Adjust/Threshold”. In our case, we set the parameter as shown in Fig. R2 (right figure). Click “Limit to threshold” button in Analyze/Set Measurements. Then, the ST area was analyzed by “Analyze/Measure”. To calculate the percentage, the whole area of the circles was analyzed after unchecking “Limit to threshold”.

Table R1 (review only)

Table R1 (review only). Details of ST coverage (%)

	Arbitrary unit		
	ST area	Whole area	%
Fig. 3F (a)	35.773	53.066	67.41227905
Fig. 3F (b)	28.192	53.895	52.30911958
Fig. 3F (c)	26.617	53.672	49.59196602
Fig. 3F (d)	21.835	53.76	40.6156994
Fig. 4E	55.751	55.754	99.99461922
Fig. 5B	53.624	53.844	99.59141223
Fig. S13B CT27	52.813	52.91	99.81666982
Fig. S13B CT29	52.779	52.804	99.9526551
Fig. S13B CT30	54.778	54.794	99.97079972
Fig. R3 A	6.539	6.539	100
Fig. R3 A	7.062	7.068	99.91511036

Fig. R3 A	6.612	6.636	99.63833635
Fig. R3 A	6.457	6.493	99.44555675

Fig. R3 (review only)

Fig. R3. ST barrier models having the small area of the collagen membrane.

(A) Immunofluorescence staining images of SDC1. Column-type devices that the diameter of the collagen membrane is 1 mm, were fabricated. TS cells were seeded on the devices and cultured according to the 6-day culture protocol for the ST barrier models. On day 6, cells were fixed with 4% PFA and treated with 0.3% Triton X-100/PBS(-). After washing, cell samples were subjected to immunostaining with antibodies for syndecan 1 (SDC1). The scale bars indicate 200 μ m. Images were taken using BZ-X800/810. Percentage of the area covered with SDC1-positive cells in the circular collagen membrane in each sample was determined using ImageJ version 1.47t (NIH).

(B) The ST coverage of the ST barrier models ($\Phi = 1$ mm). Data are shown as mean \pm SE (N=4).

Comment #1-2:

-Fig. 3K: TEER values are used as a measure of the integrity of the ST layer. However, an increase in these values does not provide evidence of an intact layer, as the presence of any additional cells that would increase resistance.

Response:

Yes, we agree with the reviewer's comment. TEER values provide information on cell layer (sheet) integrity but do not provide evidence of an intact layer. To address this point, we have conducted an additional experiment and obtained results with a 100% ST coverage sample (Fig. R3), and we found that further consideration is required to obtain 100% ST coverage models in future studies reproducibly.

Comment #1-3:

Furthermore:

·P8L24: it is not clear what is intended by 'high' in this context. The values are higher than in TSM, but is it considered 'high' compared to other cell barriers?

Response:

We thank the reviewer for pointing it out. We have changed the sentence "TEER was high in ST barrier models (Fig. 3K)." to "TEER values of ST barrier models were higher than those of TS barrier models (Fig. 3K)." (page 8, line 34). In many studies with choriocarcinoma cell lines, TEER values were measured for cells cultured on a porous membrane (porosity is about 15% at most), which differs from our collagen membrane. Therefore, a direct comparison of TEER values is not the appropriate way from this point of view. In such studies with porous membranes, TEER values of choriocarcinoma cell lines were 60 - 400 (ohms · cm²)[1, 2].

[1] Rothbauer M, Patel N, Gondola H, Siwetz M, Huppertz B, Ertl P. A comparative study of five physiological key parameters between four different human trophoblast-derived cell lines. *Sci Rep.* 2017 Jul 19;7(1):5892. PMID: 28724925

[2] Liu F, Soares MJ, Audus KL. Permeability properties of monolayers of the human trophoblast cell line BeWo. *Am J Physiol.* 1997 Nov;273(5):C1596-604. PMID: 9374645.

Comment #1-4:

· Fig. 3K: please include the TEER measurements in the same graph in order to be able to directly compare the values with the previous model and to assess if there is an improvement

Response:

As suggested by the reviewer, we conducted additional experiments to make a graph for a direct comparison of the TEER values between the previous model (1st, the 7-day culture model) and the improved model (2nd, the 6-day culture model). The results that we expected were obtained and shown in Fig. 4G. The modification of the culture protocol, which was the use of fewer cells and more differentiative S-DM2, may have affected the thickness of cell sheets in the 2nd models, resulting in lower TEER values in the improved models. We have explained this point in the revised manuscript (page 10, lines 10-15).

Fig. 4G

Fig. 4G. Enhancement of ST coverage in the barrier models (F) TEER levels for the previous model of Fig. 3D(a) (1st) and the improved model (2nd). Data are shown as mean ± SE (N=4).

Comment #1-5:

· Fig. 5F: the TEER of ST is much lower compared to the previous values in Fig. 3K for the same condition (ST), why is this so?

Response:

The TEER values in Fig. 5F and Fig. 3K are shown for the previous and improved models, respectively. As we mentioned above with additional Fig. 4G,

the thickness of cell sheets should be different between these models. Since the number of TS cells used was lesser (ca 70% of the previous cell number) and culture time was shorter (6 days) in the revised protocol, the revised ST/TS layer should be thinner than the previous one, although we could not accurately determine the thickness of the thinner barrier models because of their vulnerability in making section analyses. We think that the difference in the TEER values between the revised barrier models and the previous models was probably due to the thickness of the barriers rather than ST coverage.

Comment #1-6:

·Fig. S11E: Another example of TEER values being very different to the other measurements found in the other graphs. How reproducible are these conditions? If the layer is intact, then shouldn't the values be consistent?

Response:

We thank the reviewer for valuable comments. It may not be true that TEER values are consistent if the layer is intact. TEER values in Fig. 3K (229 ohms · cm² at day 7), Fig. 4G 1st (180 ohms · cm² at day 7), and Fig. S14E (277 ohms · cm² at day 7) were obtained from the previous models, and the values in Fig. 4G 2nd (83 ohms · cm² at day 6), Fig. 4I (63 ohms · cm² at day 6), and Fig. 5F (92 ohms · cm² at day 6) were obtained from the improved models. Variations of the values between samples in individual groups were slight, but the variations between separate experiments were relatively larger. TEER values are affected by not only the ST coverage but also cell numbers. Therefore, it could make the variation smaller by controlling the culture materials (for example, the same lot numbers of reagents and the same cell passage numbers) and culture hours (the exact same culture hours). Our results indicated that it is unnecessary to make everything, including lot numbers of reagents and culture hours, the same to make similar ST barrier models. This is a good advantage for generating the ST barrier models at other laboratories.

Comment #1-7:

-Fig. 3L: the data should be normalized to the amount of media for each condition

Response:

The data in Fig. 3L were normalized to the amount of media for each condition, and this process was written in the previous manuscript. We have amended the manuscript by adding a sentence to clarify it (page 9, line 3).

Comment #1-8:

-Fig. 4: Fig. 4C at day 6 look very different from Figure D in term of SDC-1-GFP localization and coverage. As stated in the text, both pictures have been taken at day 6..can the author comment on that? Plus, Figure D there is not colocalization between SDC-1 and SDC-1-GFP. Generally, they claim there is a 100% coverage, but the images do not support what they claim and a functional assay is required (dextran).

Response:

We thank the reviewer for the insightful suggestion. The sample for Fig. 4C and Fig. 4D was the same, and images were obtained at different levels of excitation light intensities and exposure times. The image for Fig. 4D was obtained after immunostaining, and the exposure time was shorter than that for Fig. 4C to make a clear merged image with the red-colored image (SDC1).

We agree with the reviewer's comment. Fluorescent signals of SDC1-GFP and anti-SDC1-antibody/Alexa 555 anti-mouse IgG antibody appeared to come from different cell positions. We are unsure of the reason for the difference, but we think that the difference may come from the difference of each position of GFP and anti-SDC1-antibody/Alexa 555 anti-mouse IgG antibody. The monoclonal antibody we used to detect SDC1 can recognize SDC1 in living cells, which means that this antibody recognizes an extracellular domain of SDC1. On the other hand, GFP was combined with the C-terminal domain of SDC1 (Fig. S7), which means GFP exists inside the cell membrane.

We have claimed that the ST coverage was almost 100% in the improved barrier models. As the reviewer suggested, we additionally conducted a functional assay with FITC-dextran for the improved ST barrier models and the control TS models prepared by culturing TS cells for three days. Two kinds of fluorescein isothiocyanate (FITC)-conjugated dextran (4 kDa or 70 kDa) were used to examine size-dependent paracellular transport. FITC-dextran (70 kDa) has a molecular weight similar to human serum albumin, the most abundant protein in plasma, and it is often used to check cell sheet integrity [1, 2]. Those

FITC-dextran was added into wells (the apical side of ST barrier models), and solutions in the column (the basal side) were collected after 1.5 hours. As a result, FITC-dextran (70 kDa) hardly crossed ST barrier models, indicating the function of the cellular barrier (Fig. 4H-J). We have added information on this assay in the revised manuscript (page 10, lines 15-21, and page 21, lines 13-24).

[1] R. Cao, et al., Cell Prolif 56 (5), e13469 (2023).

[2] J. Pauty, et al., Nanotheranostics 1 (1), 103-113 (2017).

Fig. 4

Fig. 4. Enhancement of ST coverage in the barrier models

(H-J) FITC-dextran permeability assay. Images of TS models and ST models (H), and TEER levels of both models (I, N=6).

(J) Percentage of transfer of FITC-dextran (4 kDa and 70 kDa) from the apical (well) to the basal (column).

Data are shown as mean \pm SE, which are percentages against initial concentrations of FITC-dextran (N=3).

* $P < 0.05$, Tukey's multiple comparison test.

Comment #1-9:

-There are no functional experiments that demonstrate the presence of an intact

syncytium. The IgG/IgA experiment was not conclusive and was simply moved into supplementary Fig S8. Are there alternative experiments that could be done, for instance looking at whether fluorescently-labelled small molecule (FITC-dextran) crosses, which is a commonly used assay to assess a cellular barrier?

Response:

As suggested, we have conducted an assay with FITC-dextran (Fig. 4H-J as shown above). We were able to demonstrate the function of the cellular barrier.

Comment #2-1:

2. Insufficient characterization of the trophoblast cell states captured in the organoid model. In the generation of new models, a key aspect is to characterize the cell types present in the model and the cellular organization within the system to have a clear understanding of which part of the tissue is recapitulated and potential limitations.

-In the rebuttal P26#4, the authors reply that they could not get NOTCH1 staining to work. This is not a satisfactory reply, as there are specific antibodies for immunofluorescence for NOTCH1 (for example see PMID: 27849611) as well as ITGA2 (PMID:27849611, PMID: 29540503). If staining does not work, then as an alternative approach the expression of these markers can also be assessed by FACS. Furthermore, the references cited in their reply refer to human ESC derived trophoblast cells and which are different to the blastocyst or 1st trimester placenta derived TSC used in this work. The latter have been assessed clearly to be ITGA2-positive and therefore representing cells of the cell column in two independent studies (PMID:27849611, PMID: 29540503). It is important to clarify which cell types are present in the model, please assess either NOTCH1 or ITGA2 expression.

Response:

We thank the reviewer for the valuable comments. As suggested by the reviewer, we have again conducted immunostaining for NOTCH1 and ITGA2 for the spherical organoids. In addition, to see gene expression levels in cells in the human placenta including VCT, VCT_CCC (the cell column), we analyzed data of a spatially resolved multi-omics single-cell atlas of the entire human

maternal–fetal interface, which were reported by another group this year [1]. Furthermore, we have analyzed our RNA-sequencing data published previously [2]. These analyses indicated that NOTCH 1 and ITGA2 are more expressed in VCT_CCC than in CT cells (Figure R4), and more expressed in TS cells (CT27, CT29, and CT30) than in CT cells (Figure R5). These results raise the possibility that TS cells include cells like VCT_CCC. Given the reviewer’s previous comment that a report has shown that the TS cells actually represent trophoblast cells from the cell column (CC) rather than CT, these things should be further elucidated for TS cells and our organoid models in future studies. We could not detect NOTCH1 and ITGA2 in immunostaining, and this might be because expression levels of these genes are not high in these cells (Fig. R5).

[1] A. Arutyunyan, et al., Nature 616 (7955), 143-151 (2023).

[2] H. Okae, et al., Cell Stem Cell 22 (1), 50-63 e56 (2018).

Fig. R4

Fig. R4. Dot plots showing log-transformed, normalized expression levels (color) and proportion of cells expressing indicated genes (dot size) for *NOTCH1*, *ITGA2*, *CDH1*, and *SDC1* genes.

scRNA-seq and snRNA-seq data for first-trimester human placentas, including cell type annotations, were obtained from previous studies (A. Arutyunyan, et al., Nature 616 (7955), 143-151 (2023)). Fetal (f), fibroblasts (F), decidual (d), epithelial (Epi), secretory (sec), luminal (lum), endothelial (Endo), maternal (m), lymphatic (l), perivascular cells (PV), stromal (S), uterine smooth muscle cells (uSMC), villous cytotrophoblast (VCT), proliferative (p), cytotrophoblast cell column (CCC), syncytiotrophoblast (SCT), extravillous cytotrophoblast (EVT), endovascular EVT (eEVT), interstitial EVT (iEVT), giant cells (GC), natural killer (NK), innate lymphocytes (ILC), macrophages (M), dendritic cells (DC), Hofbauer cells (HOFB), monocytes (MO).

Fig. R5

Fig. R5. Analysis for expression levels of *NOTCH1*, *ITGA2*, *CDH1*, and *SDC1* genes.

Gene expression analysis in primary trophoblast cells, TS cells, and TS-derived ST cells. The gene expression levels are from our previous RNA-seq data (Okae H, et al. Cell Stem Cell 22, 50-63 e56, 2018). TPM, transcripts per million. CT, cytotrophoblasts from the 1st-trimester human placenta. ST, syncytiotrophoblasts from the 1st-trimester human placenta. TS, trophoblast stem cells. CT27_ST3D or CT29_ST3D, 3D ST cells differentiated from CT27 or CT29, respectively. The number shown after the name of CT or ST (i.e., CT_1, CT_2, and CT_3, and ST_1, ST_2, and ST_3) mean that samples were isolated from the three placentas.

Comment #2-2:

-Fig 1C, D, G, H and Fig. S4B: SDC-1 and hCG staining can be clearly seen both inside and in the outer layer of the organoids. This suggests that syncytium is also forming also on the inside of the organoid which is not physiological. The authors should comment on this.

Response:

As the reviewer pointed out, some spherical organoids include ST cells inside. We have added this information in the revised manuscript (page 6, lines 29-30). We are not sure what factors cause ST cells inside. As a trend, the larger organoids appeared to have a higher possibility of having ST cells inside than the smaller organoids. Therefore, consideration to make smaller organoids may be one of the ways to reduce the inside ST cells, and we would like to perform such a consideration in our future studies. The information on the size of the spherical organoids we made has also been added to the revised manuscript (Fig. S4) (page 6, lines 26-27).

Fig. S4

Fig. S4. Size of the spherical trophoblast organoids in the 61-well agarose plate. (A) The spherical organoids were generated using microwell plates that were designed in Fig. S2 and fixed with 4% PFA on Day 8. A whole image of a microwell plate with the spherical organoids was created by taking and combining 100 images using BZ-X800/810. The scale bars indicate 2 mm. (B) A partially magnified image of Fig. S4(A). (C) Size (diameter) of the spherical organoids was measured using ImageJ version 1.47t (NIH). The organoids (N=19) positioned and focused in the central area were measured. Images were taken using BZ-X800/810. The scale bars indicate 2000 µm (A) or 1000 µm (B).

Comment #2-3:

-Fig 1I: TEM is needed to show multinucleated region in the STB, only presence

of microvilli is shown.

Response:

We thank the reviewer for the suggestion. As suggested, we have conducted TEM analysis for the spherical organoids and the improved ST barrier models and confirmed multinucleated region on the surface of the both models (Fig. 2F, 2G, and Fig. S8 for the spherical organoids) (Fig. 4F and S13D for the ST barrier models). We have added the results and methods of this analysis (page 7, lines 31-33, and page 20, lines 17-22).

Fig. 2

Fig. 2. Analysis of the fusion of ST cells in apical-out spherical trophoblast organoids.

(F) Transmission electron microscopy images of the spherical organoids.

The scale bars indicate 5 μm (F), or 10 μm (G). Images were taken using JEM-1400Flash (F and G). ST, syncytiotrophoblast. Nu, nucleus, Images from the organoids of day 8. Nu, nucleus. Mv, microvilli on the surface of the organoids.

Fig. 4

Fig. 4. Enhancement of ST coverage in the barrier models.
 (F) Transmission electron microscopy images of the barrier models. Nu, nucleus. Mv, microvilli on the surface of the organoids. The scale bars indicate 5 μm (F).

Fig. S8

Fig. S8. TEM images of the spherical trophoblast organoids of each TS cell line (CT27, CT29, or CT30)

The surfaces of the spherical organoids were analyzed using a transmission electron microscope (JEM-1400Flash, JEOL, Japan). The scale bars indicate 5 μm . Images from the organoid models of day 8. Nu, nucleus.

Fig. S13

Fig. S13. ST barrier models of each TS cell line (CT27, CT29, or CT30).

(D) The surfaces of ST barrier models were analyzed using a transmission electron microscope (JEM-1400Flash, JEOL, Japan) (D).

The scale bars indicate 5 μ m (D). Images from the barrier models of day 6. Nu, nucleus. Mv, microvilli on the surface of the organoids.

Comment #2-4:

-Fig. S4: the quality of the images and/or staining is low making it difficult to assess them. For example, E-cadherin is not clearly on the cell surface and GATA3 should be nuclear. Please include better quality images in which the localization of these markers can be assessed more clearly and also the inclusion of a proliferative marker, Ki67 would be helpful.

Response:

We thank the reviewer's suggestion. We included better quality images in Fig. S5A, and we made Fig. R6 for review only. We think that the increase in the quality of the images allowed readers to assess cells more clearly.

As suggested by the reviewer, we have stained our section samples with an anti-Ki67 antibody. Some undifferentiated cells and ST cells were stained with the antibody. Undifferentiated cells appeared to express Ki67 at higher levels than ST cells, but staining levels varied depending on individual cells (Fig. S5A and R7), which was in accordance with the results of a human placental villus in the Human Protein Atlas (Fig. R8). Furthermore, to characterize the cell types present in the model, we stained the spherical organoids with anti-TP63 antibodies. TP63 was expressed at higher levels in undifferentiated cells but to a lesser extent in ST cells (Fig. S5A and R7), consistent with the previous studies [1, 2]. We have added these things about Ki67 and TP63 in the revised manuscript (page 6, lines 30-31, and page 7, lines

1-2).

[1] M. Y. Turco, et al., *Nature* 564 (7735), 263-267 (2018).

[2] S. Haider, et al., *Stem Cell Reports* 11 (2), 537-551 (2018).

Fig. R6

Fig. R6. Immunostaining analysis of trophoblast organoids. Magnified images for E-cadherin and GATA3 from Fig. S5A.

Fig. S5

Fig. S5A. Immunostaining analysis of trophoblast organoids.

(A) Apical-out spherical trophoblast organoids were generated from a TS cell line CT27 by the same procedure as described in Figure 1A. Immunostaining analysis with antibodies for syndecan 1 (SDC1), E-cadherin, TP63, Ki67, GATA3, normal mouse IgG (negative control), or normal rabbit IgG (negative control) was conducted for cross-sections of the organoids. The scale bars indicate 200 μm or 100 μm (magnified images).

Fig. R7

Fig. R7. Immunostaining analysis of trophoblast organoids. Magnified images for Ki67 and TP63 from Fig. S5A.

Fig. R8

Fig. R8. Immunostaining for Ki67 in the human placenta

This image was obtained from the Human Protein Atlas.

<https://www.proteinatlas.org/ENSG00000148773-MKI67/tissue/placenta>

Comment #2-5:

-P13L23-25, Fig.S7D: not possible to assess if there is a release of ST cell aggregates, there is no characterization.

Response:

We thank the reviewer's valuable suggestion. We have conducted immunostaining to assess whether the released cells are ST cells. Thanks to this advice, our manuscript became clearer.

We generated the ST barrier models of 1st model (the previous model) and 2nd model (the improved model). Cell aggregates that fell on the wells from Day 6 to Day 7 were collected. In the 2nd model, cells were additionally cultured for one day with the fresh culture medium for the collection. Cell aggregates were collected, centrifuged, and stained using primary antibody (anti-SDC1) and secondary antibody (anti-mouse IgG with Alexa 555). Results of

immunostaining characterized these aggregates as ST cells (Fig. S11B). This is not direct evidence showing that released cells were ST cells, but these results raised the possibility of it.

Fig. S11B

Fig. S11. Release of cell aggregates from ST barrier models.

(B) Immunostaining of SDC1 in cell aggregates from two kinds of ST barrier models (1st and 2nd). Cell aggregates that were released between Day 6 to Day 7 were collected on Day 7. Then, the aggregates were fixed with 4% PFA and subjected to immunostaining with an anti-SDC1 antibody. Images were taken using BZ-X800/810. The scale bars indicate 100 μ m.

Comment #2-6:

Fig. S1: please show cells before and after ST differentiation to compare.

Response:

We thank the reviewer for the suggestion. We have revised images in Fig. S1 to show cells before and after ST differentiation.

Fig. S1

Fig. S1. Differentiation of ST cells on a culture plate.

A 6-well plate was coated with 10 $\mu\text{g}/\text{mL}$ collagen IV at 37°C for >1.5 min and washed with PBS(-). TS cells were then seeded at 0.5×10^5 cells/well to the wells and cultured with TS medium without iMatrix-511. After 2 days, culture media were changed to ST medium (DMEM/F12 supplemented with 0.3% BSA, 50 units/mL penicillin, 50 $\mu\text{g}/\text{mL}$ streptomycin, 1% ITS-X, 4% KSR, 2.5 μM Y27632, and 2 μM forskolin). On Day 4, the medium was changed. On day 6, cells were fixed with 4% PFA for 10 min, treated with 0.3% Triton X-100 for 5 min, and stained with an antibody for SDC1. Scale bars indicate 200 μm .

Comment #3-1:

3.The endothelial compartment is not properly formed. The inclusion of an endothelial component (HUVEC) to increase the complexity of their model is nice, however the data shown does not support an improvement for the following reasons:

-The presence of the three layers (HUVEC, CT and ST) has not been demonstrated. Fig. 5D shows only ST (SDC1+) and HUVEC (GFP). According

to Fig 5H, the barrier model should have the three layers. Please include a staining with a CT marker for example, E-Cadherin.

Response:

We thank the reviewer for the valuable suggestion. We have generated the ST barrier models and treated them with anti-SDC1 antibody/Alexa Fluor 555-conjugated secondary antibody and anti-E-cadherin antibody/Alexa Fluor 647-conjugated secondary antibody. Fig. 5D shows the three layers consisting of ST cells, undifferentiated cells, and GFP-HUVECs. As we agree with the reviewer's comment, the inclusion of HUVEC to increase the complexity of their model is nice. By including HUVECs, we can consider a HUVEC barrier and transportation of substances by HUVEC's transporters.

Fig. 5D

SDC1 E-cad GFP-HUVEC Hoechst

Fig. 5. ST/HUVEC co-culture barrier models and translocation of drugs.
(D) ST/HUVEC models in a side view by confocal microscopy. The scale bars indicate 200 μ m. Data of ST or ST/HUVEC models were obtained from the models of day 6.

Comment #3-2:

-Fig. 5B and D: HUVEC line is able to form endothelial tubes but in these images, it seems that it does not even form a continuous layer and the cells are present but sparse. For the inclusion of these cells to add value to the model, it would require a formation of a layer. Could a different medium be used that is in direct contact with the HUVEC only to promote this?

Response:

We thank the reviewer for the constructive suggestion. As the reviewer pointed out, a part of the HUVEC layer (Fig. 5B) appeared not to form a continuous

layer, and we wrote this point in the previous manuscript (page 15, lines 5-6).

As suggested by the reviewer, we have used a different medium that is in direct contact with the HUVEC only to promote the formation of the HUVEC layer (Fig. S16). For this purpose, we used a conditioned medium (CM) that contains abundant growth factors such as VEGF instead of a conventional EGM-2 medium only. CM was prepared from normal human lung fibroblasts (NHLF), which have been widely used for generating 3D HUVEC microvascular networks in the field of microphysiological systems [1-3]. The preparation procedure of CM was written in the legend of Fig. S16. As we expected, EGM2-CM induced the proliferation of GFP-HUVECs and formed a continuous layer. The significant effect of CM was not observed in ST/HUVEC models when we analyzed the fluorescence intensity of GFP-HUVECs, although there was a little trend visually that EGM+CM induced the proliferation of GFP-HUVECs in ST/HUVEC models. Further consideration for culturing HUVECs is required in future studies. We added this information in the revised manuscript (page 15, lines 5-8).

[1] S. Kim, et al., *Lab on a Chip* 13 (8), 1489-1500 (2013).

[2] Y. Nashimoto, et al., *Integrative Biology* 9 (6), 506-518 (2017).

[3] S. Bang, et al., *Advanced Functional Materials* 32 (1), 2270001 (2022).

Fig. S16

Fig. S16. Consideration of culture media for culturing GFP-HUVECs

(A) Representative two images of GFP-HUVECs in each group. ST/HUVEC barrier models were generated along with the 6-day protocol shown in Fig. 5A. TS cells were seeded to the collagen membrane on the column-type device and started to culture in PreM. On day 2, culture media in wells were replaced with W-DM, and GFP-HUVECs were added into each insert column, which contains 100 μ L of EGM-2 or CM (conditioned medium), at 0.4×10^6 cells/mL \times 100 μ L/column. On day 4, culture media in wells were changed to S-DM2, and those in each column were changed to the corresponding media. On day 6, cells were fixed with 4% PFA and imaged using a fluorescent microscope BZ-X800/810. CM was obtained from normal human lung fibroblasts (NHLF). Briefly, NHLF

were cultured with D-MEM high glucose (Fujifilm Wako) containing 10% FBS, 100 units/mL penicillin, and 100 µg/mL streptomycin in a T75 flask. After cells reached 80-90% confluency, the culture medium was replaced with EGM-2. After three days, the medium was collected and centrifuged at 1500 rpm for 3 min. The supernatant was filtrated (pore size, 0.22 µm), and the obtained medium was stored as CM at -20°C until use.

(B) Fluorescence intensity of GFP-HUVECs for each group was analyzed using ImageJ version 1.47t (N=4).

(C) TEER measurements in each barrier model (N = 4). * $P < 0.05$, Student's t -test (EGM-2 vs. EGM-2+CM).

CM, conditioned medium; EGM-2, endothelial cell growth medium 2 (Lonza); PreM, pre-culture medium; W-DM, weak differentiation medium. S-DM2, strong differentiation medium 2 containing 2 µM forskolin. The scale bar indicates 200 µm.

Comment #3-3:

-Fig.5A and B: what is the medium used in B? P10L12 the use of EGM-2 is mentioned but it is not mentioned in 5A.

Response:

We are sorry for lack of information. Samples in Fig. 5B were obtained in culture schedule shown in Fig. 5A. We have amended the legend of Fig. 5.

Comment #3-4:

-Fig. 5C: P11L4 'co-culture with HUVEC slightly increased the Papp values of antipyrine and caffeine' is not supported by the data, as a significant difference between ST and ST/HU is not evident (please include p-value).

Response:

As suggested, we have conducted Student t -test for the difference between ST and ST/HU, and p -values were $P = 0.0061$ (antipyrine) and $P = 0.0185$ (caffeine), respectively. We have added the information on statistical analysis after the sentence. "Co-culture with HUVECs slightly increased the Papp values of antipyrine and caffeine obtained in ST cell models ($P < 0.05$, Student t -test)". (page 11, lines 26-27)

Minor Comment #1:

Minor comments

-Consistency with nomenclature: in Fig 1 organoids are referred to as spheroids and also spherical organoids throughout.

Response:

We have changed "ST spheroid" to "spherical organoid" throughout.

Minor Comment #2:

-Abstract, L13: 'revealed that and that'

Response:

We have amended the manuscript. We thank the reviewer for pointing it out.

Minor Comment #3:

-Abstract L10-11, due to the lack of experiments that do not demonstrate the formation of an intact syncytium, the authors should be more cautious of their claims. This is valid throughout the manuscript.

Response:

We have changed "entirely covered" to "almost entirely covered" in the sentence and have revised other parts like that throughout the manuscript (page 2, line 10, and page 9, line 26, and page 9, line 37).

Minor Comment #4:

-P4L12: 'shrinkage' is not clear what the authors mean

Response:

When TS cells differentiate and fuse into ST cells, the total size of cells becomes smaller. The revised Fig. S1 may help to understand this phenomenon.

Minor Comment #5:

-P5L2: human placental villous

Response:

We have amended the manuscript. We thank the reviewer for pointing it out.

(page 5, line 5)

Minor Comment #6:

-P6L31: GATA 3 is expressed 'in' CT

Response:

We thank the reviewer for pointing it out (page 6, line 33).

Minor Comment #7:

-Fig.3: G and H, please add in the Figure legend, the protocol/culture conditions that were used for these images for clarity.

Response:

We are sorry for the lack of information. Fig. G is a magnified image of Fig. 3F(a), and Fig. H is a 3D image of the same sample as Fig. G. We added more information in the legend of Fig. 3 for clarity.

Minor Comment #8:

-Fig S8B legend: 'intensicy' instead of 'intensity'

Response:

We have amended the legend. We thank the reviewer for pointing it out.

Minor Comment #9:

-P12L20: ST cells of the villi do not attach to decidua

Response:

We thank the reviewer for pointing it out. We changed the phrase "attach to" to "contact with". (page 13, line 20)

Minor Comment #10:

-P12L21: 'anchoring villous' – 'villus'

Response:

We have amended the manuscript. We thank the reviewer for pointing it out. (page 13, lines 21)

Minor Comment #11:

-P14L21: repetition of 'pathology'

Response:

We removed it. We thank the reviewer for pointing it out. (page 15, lines 22)

REVIEWERS' COMMENTS

Reviewer #1 (Remarks to the Author):

I am satisfied that the authors have addressed my prior concerns with the manuscript. I support publication of this manuscript.

ADDITIONAL GUIDANCE RELATED TO REVIEWER 3'S PRIOR COMMENTS

Comment 1 (around the complete coverage of the barrier model by syncytiotrophoblast):

Here in comment #1-1 reviewer 3 is expressing a strong desire to see a model with a completely intact syncytiotrophoblast barrier. Whilst this indeed would be an ideal model, and is of the most importance in transport studies, no other studies to date have achieved this with primary cells, and so in my mind it is still a bit of a pipe dream and I don't think precludes publication of this model as a significant advance on what has gone before. However, to allow researchers to use the model and interpret results accurately in future work it is important to be transparent about the % of coverage achieved. In their response the reviewers have presented a table that quantifies the percentage of coverage of trophoblast area (undertaken in figure panels only) – in my opinion it is important to include data quantifying original images of the best developed models and report the mean % of coverage across replicates (not just from figure panels) and to include the methodology of calculation in the methods section of the paper. I am not convinced of the methodology presented in Fig R2 quantifying greyscale images (cytotrophoblast would also be grey?), and would prefer to see quantification of syndecan-1 fluorescence (as has been done in Fig R3) – indeed in the response to reviewers it is not clear which data in Table R1 was derived by which method. I have no issue with the TEER data (comment #1-2) as I think this provides important supplementing data about cell coverage and integrity of the model overall, and provides a parallel metric to that used in alternate less physiological cell line methodologies to allow comparison across the literature. The authors have addressed the other TEER comments raised by the reviewer appropriately in my opinion (#1-3 to #1-6). #1-7 is a minor comment that has been addressed by amending a sentence. Comment #1-8 and #1-9 – I don't have access to the original figures in what was sent. Additional dextran experiments are appropriate to support barrier function of the model, however I'm struggling to interpret the graph presented as it is not clear what BL TS and ST stand for in relation to the model setup described.

Comment 2 (around characterisation of trophoblast stem cell states in the organoid model)

Here reviewer 3 considers that characterisation of the TSC states in the organoid model is insufficient. Whilst I agree that there are a number of Notch1 antibodies that do work, I don't actually consider this a be all end all marker of TSCs – rather it is a marker of EVT progenitors. Infact, there is no clear marker for which the presence or absence of expression delineates TSCs from cytotrophoblasts in the human scenario – rather levels of some factors are higher or lower than others. For example, unlike the mouse Cdx2 and Tead4 are both expressed in both TSC and cytotrophoblast states. As such, I think this request from the reviewer does not actually add that much to the paper, and indeed the authors response that their analysis indicating that both Notch1 and ITGA2 are expressed in the cell column trophoblasts and cytotrophoblasts supports this. I don't think there is yet conclusive evidence that TSCs reside exclusively in a niche in the cell columns, as evidence has also been presented from several groups (published and unpublished) that they may reside intermittently at the cytotrophoblast-mesenchyme border as well. Comments #2-2 and #2-3 are relatively minor and have been addressed by the authors adequately in the text. Images provided in the response to comment #2-4 are appropriate to address the reviewers concerns. Assessment of the shed material in response to #2-5 is appropriate – although it would be nice to see H&E images to delineate cell morphology more clearly (do these have the typical tear shaped morphology of syncytial nuclear aggregates? How do the nuclei arrange themselves within the cytoplasm?). Images provided in response to #2-6 clearly show syncytialization of the cells.

Comment 3 (concerns around formation of the endothelial compartment)

Images provided in response to #3-1 are convincing of the 3 cell layers formed, and strengthen this part of the paper. Reviewer queries around the continuity of the HUVEC layer in #3-2 were valid, and the addition of different media on this side of the model a good suggestion. I don't quite understand the authors' response to this point as they seem to imply that "EGM-2-CM induced the proliferation of GFP-HUVECs and formed a continuous layer" but also state that "the significant effect was not observed in ST/HUVEC models" – I assume the former was in HUVEC cultures alone and that STB disrupted this effect? Regardless, while an intact HUVEC layer would be ideal, as presented the HUVEC layers are relatively confluent. As such I believe that as the manuscript stands the authors have clearly delineated the limitations of the current model, and so as long as this is stated clearly this would not preclude publication in my eyes. It may be appropriate to quantify the % coverage using fluorescence as has been requested for the ST layer earlier. All other comments are relatively minor and have been appropriately addressed.

Trophoblast stem cell-based organoid models of the human placental barrier

Takeshi Hori, Hiroaki Okae, Shun Shibata, Norio Kobayashi, Eri H Kobayashi,
Akira Oike, Asato Sekiya, Takahiro Arima, and Hirokazu Kaji

Point-by-point response to the reviewers' comments

Reviewer #1:

General comments:

I am satisfied that the authors have addressed my prior concerns with the manuscript. I support publication of this manuscript.

Response:

We sincerely appreciate the reviewer's constructive suggestions. The quality of the manuscript has significantly improved thanks to your suggestions.

ADDITIONAL GUIDANCE RELATED TO REVIEWER 3'S PRIOR COMMENTS

Comment 1-1 (around the complete coverage of the barrier model by syncytiotrophoblast):

Here in comment #1-1 reviewer 3 is expressing a strong desire to see a model with a completely intact syncytiotrophoblast barrier. Whilst this indeed would be an ideal model, and is of the most importance in transport studies, no other studies to date have achieved this with primary cells, and so in my mind it is still a bit of a pipe dream and I don't think precludes publication of this model as a significant advance on what has gone before. However, to allow researchers to use the model and interpret results accurately in future work it is important to be transparent about the % of coverage achieved. In their response the reviewers have presented a table that quantifies the percentage of coverage of trophoblast area (undertaken in figure panels only) – in my opinion it is important to include data quantifying original images of the best developed models and report the mean % of coverage across replicates (not just from figure panels) and to include the methodology of calculation in the methods section of the paper. I am not convinced of the methodology presented in Fig R2 quantifying greyscale images (cytotrophoblast would also be grey?), and would prefer to see quantification of syndecan-1 fluorescence (as has been done in Fig R3) – indeed in the response to reviewers it is not clear which data in Table R1 was derived by which method.

Response:

We thank the constructive comments of reviewer 1 for our response to reviewer 3 's comments. We agree that data quantifying original images of our barrier models using three kinds of TS cells (CT27, CT29, and CT30) and reporting the mean % of coverage across replicates are important. We would like to perform such experiments in our future studies.

We are sorry for confusing you about the analysis of SDC1-positive cells. As for the methodology presented in Fig R2 quantifying greyscale images, undifferentiated cytotrophoblast was not grey since we simply changed the red color (SDC1-positive cells that were stained with the anti-SDC1 antibody) into a grey color to go forward with further analytical processes. All data in Table R1 was derived by the analytical method shown in Fig. R2. The example images are from Fig. S13B CT30 in Table R1.

Comment 1-2

I have no issue with the TEER data (comment #1-2) as I think this provides important supplementing data about cell coverage and integrity of the model overall, and provides a parallel metric to that used in alternate less physiological cell line methodologies to allow comparison across the literature.

Response:

We appreciate the reviewer's positive comments.

Comment 1-3

The authors have addressed the other TEER comments raised by the reviewer appropriately in my opinion (#1-3 to #1-6).

Response:

We appreciate the reviewer's positive comments.

Comment 1-4

#1-7 is a minor comment that has been addressed by amending a sentence.

Response:

We appreciate the reviewer's positive comments.

Comment 1-5

Comment #1-8 and #1-9 – I don't have access to the original figures in what was sent. Additional dextran experiments are appropriate to support barrier function of the model, however I'm struggling to interpret the graph presented as it is not clear what BL TS and ST stand for in relation to the model setup described.

Response:

We are sorry for confusing you. In the previous manuscript, we wrote about BL, TS, and ST at the end of the legend of Fig. 4, like the following: TS, trophoblast stem cells. ST, syncytiotrophoblast. BL, blank (only collagen membrane). For readability, we have moved this information to the legend of Fig. 4(J). (page 31, line 36).

Comment 2-1 (around characterisation of trophoblast stem cell states in the organoid model)

Here reviewer 3 considers that characterisation of the TSC states in the organoid model is insufficient. Whilst I agree that there are a number of Notch1 antibodies that do work, I don't actually consider this a be all end all marker of TSCs – rather it is a marker of EVT progenitors. Infact, there is no clear marker for which the presence or absence of expression delineates TSCs from cytotrophoblasts in the human scenario – rather levels of some factors are higher or lower than others. For example, unlike the mouse Cdx2 and Tead4 are both expressed in both TSC and cytotrophoblast states. As such, I think this request from the reviewer does not actually add that much to the paper, and indeed the authors response that their analysis indicating that both Notch1 and ITGA2 are expressed in the cell column trophoblasts and cytotrophoblasts supports this. I don't think there is yet conclusive evidence that TSCs reside exclusively in a niche in the cell columns, as evidence has also been presented from several groups (published and unpublished) that they may reside intermittently at the cytotrophoblast-mesenchyme border as well.

Response:

We thank the reviewer for the insightful comments.

Comment 2-2

Comments #2-2 and #2-3 are relatively minor and have been addressed by the authors adequately in the text.

Response:

We appreciate the reviewer's positive comments.

Comment 2-3

Images provided in the response to comment #2-4 are appropriate to address the reviewers concerns.

Response:

We appreciate the reviewer's positive comments.

Comment 2-4

Assessment of the shed material in response to #2-5 is appropriate – although it would be nice to see H&E images to delineate cell morphology more clearly (do these have the typical tear shaped morphology of syncytial nuclear aggregates? How do the nuclei arrange themselves within the cytoplasm?).

Response:

We appreciate the reviewer's constructive suggestion. More detailed characterizations of the shed material are interesting. Some aggregates looked like a tear-shaped morphology under microscopy. We would like to pursue the points by analyses including section analysis of HE-stained cells in future studies.

Comment 2-5

Images provided in response to #2-6 clearly show syncytialization of the cells.

Response:

We appreciate the reviewer's positive comments.

Comment 3-1 (concerns around formation of the endothelial compartment)

Images provided in response to #3-1 are convincing of the 3 cell layers formed,

and strengthen this part of the paper.

Response:

We appreciate the reviewer's positive comments.

Comment 3-2

Reviewer queries around the continuity of the HUVEC layer in #3-2 were valid, and the addition of different media on this side of the model a good suggestion. I don't quite understand the authors' response to this point as they seem to imply that "EGM-2-CM induced the proliferation of GFP-HUVECs and formed a continuous layer" but also state that "the significant effect was not observe in ST/HUVEC models" – I assume the former was in HUVEC cultures alone and that STB disrupted this effect? Regardless, while an intact HUVEC layer would be ideal, as presented the HUVEC layers are relatively confluent. As such I believe that as the manuscript stands the authors have clearly delineated the limitations of the current model, and so as long as this is stated clearly this would not preclude publication in my eyes. It may be appropriate to quantify the % coverage using fluorescence as has been requested for the ST layer earlier.

Response:

We thank the reviewer for the positive comments. As the reviewer mentioned, the significant induction of the GFP-HUVEC proliferation by EGM-2-CM, which contains abundant growth factors from fibroblasts, was observed only in the HUVEC cultures alone, but we did not observe the significant effect in ST/HUVEC models. ST cells could provide growth factors to GFP-HUVECs, which could be why we haven't seen the significant effect of EGM-2-CM on GFP-HUVEC proliferation in ST/HUVEC models. We have amended the revised manuscript to clarify it for readers (page 15, line 8). As we agree with the reviewer's suggestion, quantifying the % coverage of GFP-HUVECs with appropriate staining would provide valuable data in our future studies.

Comment 3-3

All other comments are relatively minor and have been appropriately addressed.

Response:

We appreciate the reviewer's positive comments.